# Communication Acceleration of Local Gradient Methods via an Accelerated Primal-Dual Algorithm with Inexact Prox

**Abdurakhmon Sadiev**[*]
KAUST[†] MIPT[‡] ISP RAS[§]
abdurakhmon.sadiev@kaust.edu.sa

**Dmitry Kovalev**
KAUST
dakovalev1@gmail.com

**Peter Richtárik**
KAUST
richtarik@gmail.com

## Abstract

Inspired by a recent breakthrough of Mishchenko et al. [2022], who for the first time showed that local gradient steps can lead to provable communication acceleration, we propose an alternative algorithm which obtains the same communication acceleration as their method (ProxSkip). Our approach is very different, however: it is based on the celebrated method of Chambolle and Pock [2011], with several nontrivial modifications: i) we allow for an inexact computation of the prox operator of a certain smooth strongly convex function via a suitable gradient-based method (e.g., GD, Fast GD or FSFOM), ii) we perform a careful modification of the dual update step in order to retain linear convergence. Our general results offer the new state-of-the-art rates for the class of strongly convex-concave saddle-point problems with bilinear coupling characterized by the absence of smoothness in the dual function. When applied to federated learning, we obtain a theoretically better alternative to ProxSkip: our method requires fewer local steps ($\mathcal{O}(\kappa^{1/3})$ or $\mathcal{O}(\kappa^{1/4})$, compared to $\mathcal{O}(\kappa^{1/2})$ of ProxSkip), and performs a deterministic number of local steps instead. Like ProxSkip, our method can be applied to optimization over a connected network, and we obtain theoretical improvements here as well.

## 1 Introduction

Communication efficiency of distributed stochastic gradient descent (SGD) can be improved, often dramatically, via a simple trick: instead of synchronizing the parameters across the parallel workers after every SGD step, let the workers perform multiple optimization steps using their local loss and data only before synchronizing. This trick dates back at least to two or three decades ago [Mangasarian, 1995], and may be much older. Due to its simplicity, it has been repeatedly rediscovered [Povey et al., 2014, Moritz et al., 2016, Ma et al., 2017]. It is the basis of the famous federated averaging (FedAvg) algorithm of McMahan et al. [2016, 2017], which is the workhorse of federated learning [Konečný et al., 2016b,a]; see also the recent surveys on federated learning [Li et al., 2020a, Kairouz et al., 2019] and federated optimization [Wang et al., 2021b].

---

[*]This work was written while A. Sadiev was a research intern at KAUST during the last semester of his MS studies at the Moscow Institute of Physics and Technology, Dolgoprudny, Russia.

[†]King Abdullah University of Science and Technology, Thuwal, Saudi Arabia

[‡]Moscow Institute of Physics and Technology, Dolgoprudny, Russia

[§]Institute for System Programming of the Russian Academy of Sciences, Research Center for Trusted Artificial Intelligence, Moscow, Russia

36th Conference on Neural Information Processing Systems (NeurIPS 2022).

## 1.1 Towards provable communication acceleration via delayed parameter synchronization

Until recently, this simple trick resisted all attempts at an appropriate theoretical justification. Through collective effort of the federated learning community, the bounds of various local SGD methods were progressively getting better [Haddadpour and Mahdavi, 2019, Li et al., 2019a,b, Khaled et al., 2019a,b, 2020, Li et al., 2020b, Woodworth et al., 2020, Stich, 2020, Gorbunov et al., 2020a, Malinovsky et al., 2020, Pathak and Wainwright, 2020, Karimireddy et al., 2020, Malinovsky et al., 2021, Mishchenko et al., 2021a, Wang and Joshi, 2021, Horváth et al., 2022], and the assumptions required to achieve them weaker. A brief overview of the progress is provided in [Mishchenko et al., 2022]. However, all known theoretical rates are worse than the rate of gradient descent, which synchronizes after every gradient step. In a recent breakthrough, Mishchenko et al. [2022] developed a novel local SGD method, called ProxSkip, which performs a random number of local gradient (or stochastic gradient) steps before synchronization, and proved that it enjoys strong communication acceleration properties. In particular, while the method needs $\mathcal{O}(\kappa \log \frac{1}{\varepsilon})$ iterations, only $\mathcal{O}(\sqrt{\kappa} \log \frac{1}{\varepsilon})$ of them involve communication.

## 1.2 Problem formulation

In this paper, we consider the composite optimization problem

$$\min_{x \in \mathbb{R}^{d_x}} \ G(x) + F(Kx), \tag{1}$$

where $G : \mathbb{R}^{d_x} \to \mathbb{R}$ is a smooth and strongly convex function, $F : \mathbb{R}^{d_y} \to \mathbb{R} \cup \{+\infty\}$ is a proper, closed and convex function, and $K : \mathbb{R}^{d_x} \to \mathbb{R}^{d_y}$ is a linear map. Let us define

$$L_{xy} \stackrel{\text{def}}{=} \max \left\{ \|Kx\| : \ x \in \mathbb{R}^{d_x}, \|x\| = 1 \right\}, \tag{2}$$

where $\| \cdot \|$ refers to the standard Euclidean norm. Note that $L_{xy}^2 \geq \lambda_{\max}(KK^\top) = \lambda_{\max}(K^\top K)$.

It will be useful to formalize our assumptions at this point as we will refer to the various constants involved in them throughout the text.

**Assumption 1.** *Function $G : \mathbb{R}^{d_x} \to \mathbb{R}$ is $\mu_x$-strongly convex, i.e.,*

$$G(x') - G(x'') - \langle \nabla G(x''), x' - x'' \rangle \geq \frac{\mu_x}{2} \|x' - x''\|^2, \qquad \forall x', x'' \in \mathbb{R}^{d_x}. \tag{3}$$

**Assumption 2.** *The function $G : \mathbb{R}^{d_x} \to \mathbb{R}$ is $L_x$-smooth, i.e.*

$$\|\nabla G(x') - \nabla G(x'')\| \leq L_x \|x' - x''\|, \qquad \forall x', x'' \in \mathbb{R}^{d_x}. \tag{4}$$

**Assumption 3.** *Function $F : \mathbb{R}^{d_y} \to \mathbb{R} \cup \{+\infty\}$ is proper, closed and convex.*

**Assumption 4** (See Kovalev et al. [2021a]). *There exists a constant $\mu_{xy} > 0$ such that*

$$\mu_{xy}^2 \leq \begin{cases} \lambda_{\min}^+(KK^\top), & \textit{if } \partial F^\star(y) \in \text{range} K \ \textit{for all } \ y \in \mathbb{R}^{d_y}, \\ \lambda_{\min}(KK^\top), & \textit{otherwise.} \end{cases}$$

## 1.3 ProxSkip

The most general version of the ProxSkip method[5] of Mishchenko et al. [2022] was designed to solve problems of the form (1). In each iteration, ProxSkip evaluates the gradient of $G$ and then flips a biased coin: with probability $p$, it additionally evaluates the proximity operator of $F^\star$, and performs a matrix-vector multiplication involving $K$. The method becomes relevant to the standard optimization formulation of federated learning (FL), i.e., to the finite-sum optimization problem

$$\min_{x \in \mathbb{R}^d} \sum_{i=1}^{n} f_i(x), \tag{5}$$

through its application to its consensus reformulation

$$\min_{x_1, \ldots, x_n \in \mathbb{R}^d} \left\{ \sum_{i=1}^{n} f_i(x_i) + \psi(x_1, \ldots, x_n) \right\} = \min_{x \in \mathbb{R}^{d_x}} \ G(x) + F(x), \tag{6}$$

---

[5]This variant is called SplitSkip in their paper.

Table 1: Summary of the key complexity results obtained by our methods APDA (Algorithm 1; Theorem 1) and APDA with Inexact Prox (Algorithm 2; Theorem 2) for solving the saddle-point problem (7).

| Algorithm | No Prox $G$ | No Prox $F^\star$ | Works with $L_y = \infty$ | Linear rate with $\mu_y = 0$ | # Outer Iterations[1] | # Inner Iterations[1] |
|---|---|---|---|---|---|---|
| CP[a] | ✗ | ✗ | ✓ | ✗ | $\frac{L_{xy}}{\sqrt{\mu_x \mu_y}}$ [2] | uses prox of $G$ |
| AltGDA[b] | ✓ | ✓ | ✗ | ✓ | $\max\left\{\frac{L}{\mu_x}, \frac{L^2}{\mu_{xy}^2}\right\}$ | —[3] |
| APDG[c] | ✓ | ✓ | ✗ | ✓ | $\max\left\{\frac{\sqrt{L_x L_y}}{\mu_{xy}}, A_{xy}\right\}$ | —[3] |
| Alg 1 | ✗ | ✗ | ✓ | ✓ | $A_{xy}$ [2] | uses prox of $G$ |
| Alg 2 | ✓ | ✗ | ✓ | ✓ | $A_{xy}$ [2] | $\max\left\{\kappa_x \kappa_{xy}, \kappa_x^{1/2} \kappa_{xy}^2\right\}$ [4] |
| Alg 2 | ✓ | ✗ | ✓ | ✓ | $A_{xy}$ [2] | $\max\left\{\kappa_x^{5/6} \kappa_{xy}, \kappa_x^{1/3} \kappa_{xy}^2\right\}$ [5] |
| Alg 2 | ✓ | ✗ | ✓ | ✓ | $A_{xy}$ [2] | $\max\left\{\kappa_x^{3/4} \kappa_{xy}, \kappa_x^{1/4} \kappa_{xy}^2\right\}$ [6] |

[a] Chambolle and Pock [2011] assume that $G$ and $F^\star$ are $\mu_x$ and $\mu_y$-strongly-convex, respectively. We do not assume $F^\star$ to be strongly convex.

[b] Zhang et al. [2022] assume that the functions $G$ and $F^\star$ are $L$-smooth (i.e., $L = \max\{L_x, L_y, L_{xy}\}$), and that $G$ is $\mu_x$-strongly-convex.

[c] Kovalev et al. [2021a] assume that the functions $G$ and $F^\star$ are $L_x$ and $L_y$-smooth, respectively, and that $G$ is $\mu_x$-strongly-convex.

[1] For brevity, we let $\kappa_{xy} \overset{\text{def}}{=} \frac{L_{xy}}{\mu_{xy}}$, $\kappa_x \overset{\text{def}}{=} \frac{L_x}{\mu_x}$, and $A_{xy} \overset{\text{def}}{=} \max\left\{\kappa_x^{1/2} \kappa_{xy}, \kappa_{xy}^2\right\}$. We omit constant factors and a $\log \frac{1}{\varepsilon}$ factor in all expressions, for brevity. So, for example, the expression $A_{xy}$ in the case of the method of our methods should be interpreted as $\mathcal{O}\left(A_{xy} \log \frac{1}{\varepsilon}\right)$.

[2] # outer iterations = # evaluations of the prox of $F^\star$.

[3] There is no prox operator and hence no inner iterations.

[4] The iterative method $\mathcal{M}$ evaluating the prox of $G$ inexactly in this case is: $\mathcal{M} = $ GD (see Lemma 1).

[5] The iterative method $\mathcal{M}$ evaluating the prox of $G$ inexactly in this case is: $\mathcal{M} = $ FGD (Fast Gradient Descent) + GD. See Lemma 1.

[6] The iterative method $\mathcal{M}$ evaluating the prox of $G$ inexactly in this case is: $\mathcal{M} = $ FSFOM + FGD (Fast Gradient Descent). See Lemma 1.

where $d_x \overset{\text{def}}{=} nd$, $x \overset{\text{def}}{=} (x_1, \ldots, x_n) \in \mathbb{R}^{nd}$, $G(x) \overset{\text{def}}{=} \sum_{i=1}^n f_i(x_i)$, and $F \overset{\text{def}}{=} \psi$ is the indicator function of the constraint $x_1 = \cdots = x_n$, i.e.,

$$\psi(x_1, \ldots, x_n) \overset{\text{def}}{=} \begin{cases} 0 & \text{if } x_1 = \cdots = x_n \\ +\infty & \text{otherwise} \end{cases}.$$

The evaluation of the proximity operator of $F$ is equivalent to averaging of the vectors $x_1, \ldots, x_n$, which necessitates communication. Therefore, if $p$ is small, ProxSkip communicates very rarely. Since $G$ is block separable, the gradient steps involving $G$, taken in between two communications, correspond to gradient steps with respect to the local loss functions $\{f_i\}$ taken by the clients. See [Mishchenko et al., 2022] for the details; we will elaborate on this as well in Section 6.

ProxSkip inexactly solves problem (1) in $\mathcal{O}\left(\kappa \log 1/\varepsilon\right)$ iterations, out of which only $\mathcal{O}\left(\sqrt{\kappa\chi} \log \frac{1}{\varepsilon}\right)$ involve communication, where $\chi$ is a condition number measuring the connectivity of the graph (the standard setup in FL corresponds to a fully connected graph, in which case $\chi = 1$; see (26) for definition).

## 2 Summary of Contributions

Inspired by the results of Mishchenko et al. [2022], we propose an alternative and substantially different algorithm which obtains the same guarantees for the number of prox evaluations (wrt $F$) as ProxSkip, but has better guarantees for the number of gradient steps (wrt $G$) in between the prox evaluations. Below we summarize the main contributions:

**Saddle-point formulation.** Unlike Mishchenko et al. [2022], we consider the saddle-point reformulation of (1)

$$\min_{x \in \mathbb{R}^{d_x}} \max_{y \in \mathbb{R}^{d_y}} \left\{G(x) + \langle y, Kx \rangle - F^\star(y)\right\}, \tag{7}$$

where $F^\star(y) \overset{\text{def}}{=} \sup_{y' \in \mathbb{R}^{d_y}} \left\{\langle y, y' \rangle - F(y')\right\}$ is the convex conjugate of $F$. Since $F$ is proper, closed and convex, so is $F^\star$. We assume throughout that (7) is solvable, i.e., there exists at least one solution

$(x^\star, y^\star)$. Such a solution then satisfies the first-oder optimality conditions[6]

$$0 \in \partial G(x^\star) + K^\top y^\star, \qquad 0 \in \partial F^\star(y^\star) - Kx^\star, \tag{8}$$

where $\partial$ denotes the subdifferential. By working with this reformulation, we can tap into the rich and powerful philosophical and technical toolbox offered by proximal-point theory, fixed point theory, and primal-dual methods, which facilitates the algorithm development and analysis. This ultimately enables us to shed new light on the nature of local gradient-type steps as inexact computations of the prox operator of $G$ in a new appropriately designed Accelerated Primal-Dual Algorithm (APDA; see Algorithm 1).

**Modifications of Chambolle-Pock** Our Algorithms 1 and 2 are inspired by the celebrated Chambolle-Pock method [Chambolle and Pock, 2011], but with several important and nontrivial modifications. While Chambolle-Pock achieves linear convergence when both $G$ and $F^\star$ are strongly convex, $F^\star$ is merely convex in our setting. Our modifications are:

  i) Inspired by the ideas of Kovalev et al. [2021a], we perform a careful modification of the dual update step (update of $y$) in order to retain linear convergence despite lack of strong convexity in $F^\star$. On the other hand, in contrast to the method of Kovalev et al. [2021a], we do not assume $F^\star$ to be smooth. This modification leads to a new method, which we call APDA (Algorithm 1). APDA relies on the evaluation of the prox operators of both $G$ and $F^\star$.

  ii) Next, we remove the reliance on the prox operator of $G$, and instead allow for its inexact evaluation via a suitable user-defined gradient-based method, which we call $\mathcal{M}$ (see (2) and Lemma 1). We call the resulting method "APDA with Inexact Prox" (Algorithm 2). The choice of method $\mathcal{M}$ will have a strong impact on the number of inexact/local steps, and this is one of the places in which we improve upon the results of Mishchenko et al. [2022].

**General theory** Our general complexity results for Algorithms 1 and 2, covered in Theorems 7 and 2, respectively, contrasted with the key baselines, are summarized in Table 1). While the method of [Chambolle and Pock, 2011] needs $F^\star$ to be strongly convex to obtain a linear rate, we only need convexity. While the AltGDA [Zhang et al., 2022] and APDG [Kovalev et al., 2021a] methods enjoy linear rates without strong convexity of $F^\star$, both require $F^\star$ to be $L_y$-smooth. In contrast, we do not need this assumption (i.e., we allow $L_y = \infty$). Our methods are the first to obtain linear convergence rates in the regime when $G$ is $L_x$-smooth and $\mu_x$-strongly-convex, and $F^\star$ is merely proper, closed and convex, without requiring it to be $L_y$-smooth, nor $\mu_y$-strongly-convex. This is important in some applications. Our two methods offer two alternative ways of dealing with this regime: while APDA (Algorithm 1) relies on the evaluation of the prox of $G$, APDA with Inexact Prox (Algorithm 2) does not. As we shall see, the latter method has an important application in federated learning.

**Federated learning and a third method** When applied to the distributed/federated problem (5) (see Section 6), APDA with Inexact Prox (Algorithm 2) turns out to be a theoretically better alternative to ProxSkip [Mishchenko et al., 2022]. In the centralized case, our method requires the same optimal number of communication rounds ($\widetilde{\mathcal{O}}(\sqrt{\kappa})$, where $\kappa = {}^{L_x}/_{\mu_x}$) as ProxSkip, but requires fewer local gradient-type steps ($\mathcal{O}(\kappa^{1/3})$ or $\mathcal{O}(\kappa^{1/4})$, compared to $\mathcal{O}(\kappa^{1/2})$ of ProxSkip, depending on the choice of the inner method $\mathcal{M}$). Like ProxSkip, our method can be applied to optimization over a connected network, and we obtain theoretical improvements in this decentralized scenario as well. However, in the decentralized regime, neither ProxSkip nor Algorithm 2 obtain the optimal bound for the number of communication rounds. For this reason, we propose a third method (Algorithm 5) which employs an accelerated gossip routine to remedy this situation. It is also notable that while ProxSkip uses a random number of local steps, all our methods perform a deterministic number of local steps. Our complexity results are summarized in Table 2.

## 3 From Proximal Point Method to Chambolle-Pock

In this section, we briefly motivate the development of the celebrated Chambolle-Pock method which acts as a starting point of our algorithm design.

---

[6]Whenever we invoke Assumption 2 ($L_x$-smoothness of $G$), we have $\partial G(x) = \{\nabla G(x)\}$, and hence the first condition can be replaced by $0 = \nabla G(x^\star) + K^\top y^\star$.

Table 2: Summary of our general convergence results provided by Theorem 2 for Algorithm 2 (APDA with Inexact Prox) and Theorem 3 for Algorithm 5 (APDA with Inexact Prox and Accelerated Gossip) for solving the saddle-point reformulation (25) of the federated learning problem (5).

| Algorithm | Method $\mathcal{M}^{(2)}$ for Inexact Prox | Deter-ministic # comm. rounds | Centralized case | | Decentralized case | |
|---|---|---|---|---|---|---|
| | | | Optimal #comm. rounds? | #Local steps per round | Optimal #comm. rounds? | #Local steps per round |
| ProxSkip [Mishchenko et al., 2022] | GD | ✗ | ✓ | $\mathcal{O}\left(\sqrt{\kappa}\right)$ | ✓[1] | $\widetilde{\mathcal{O}}\left(\sqrt{\kappa}\right)$[1] |
| Alg 2; Thm 2 | GD | ✓ | ✓ | $\widetilde{\mathcal{O}}\left(\sqrt{\kappa}\right)$ | ✗ | $\widetilde{\mathcal{O}}\left(\sqrt{\kappa}\right)$ |
| | FGD+GD | ✓ | ✓ | $\widetilde{\mathcal{O}}\left(\sqrt[3]{\kappa}\right)$ | ✗ | $\widetilde{\mathcal{O}}\left(\sqrt[3]{\kappa}\right)$ |
| | FGD+FSFOM | ✓ | ✓ | $\widetilde{\mathcal{O}}\left(\sqrt[4]{\kappa}\right)$ | ✗ | $\widetilde{\mathcal{O}}\left(\sqrt[4]{\kappa}\right)$ |
| Alg 5; Thm 3 | GD | ✓ | ✓ | $\widetilde{\mathcal{O}}\left(\sqrt{\kappa}\right)$ | ✓ | $\widetilde{\mathcal{O}}\left(\sqrt{\kappa}\right)$ |
| | FGD+GD | ✓ | ✓ | $\widetilde{\mathcal{O}}\left(\sqrt[3]{\kappa}\right)$ | ✓ | $\widetilde{\mathcal{O}}\left(\sqrt[3]{\kappa}\right)$ |
| | FGD+FSFOM | ✓ | ✓ | $\widetilde{\mathcal{O}}\left(\sqrt[4]{\kappa}\right)$ | ✓ | $\widetilde{\mathcal{O}}\left(\sqrt[4]{\kappa}\right)$ |

[1] This is true only when $\kappa \leq \chi$.
[1] GD = Gradient Descent; FGD = Fast Gradient Descent (i.e., Nesterov's accelerated GD); FSFOM = a fixed-step first-order method from [Kim and Fessler, 2021].

## 3.1 Proximal-Point Method for finding zeros of monotone operators

Our starting point is the general problem of finding a zero of an (set-valued) operator $A : \mathcal{H} \to 2^{\mathcal{H}}$, where $\mathcal{H}$ is a Hilbert space, i.e., find $z \in \mathcal{H}$ such that

$$0 \in A(z). \tag{9}$$

If $A$ is maximally monotone, its resolvent $(\mathrm{Id} + \eta A)^{-1}$ is single valued, nonexpansive, and has full domain. Moreover, $0 \in A(z)$ iff $z = (\mathrm{Id} + \eta A)^{-1}(z)$. The corresponding fixed point iteration, i.e., $z^{k+1} = (\mathrm{Id} + \eta A)^{-1}(z^k)$, is called the proximal point method (PPM) [Rockafellar, 1976]. This can be equivalently written as $z^k \in (\mathrm{Id} + \eta A)(z^{k+1})$, and subsequently as

$$z^{k+1} \in z^k - \eta A(z^{k+1}).$$

From now on, for simplicity only, we will ignore the fact that in general, $A(z^{k+1})$ is a set, and will write $z^{k+1} = z^k - \eta A(z^{k+1})$ instead to mean the same thing, i.e., that there exists $u \in A(z^{k+1})$ such that $z^{k+1} = z^k - \eta u$.

## 3.2 PPM applied to the saddle-point problem

The optimality conditions (8) of the saddle point problem (7) can be written in the form (9) with $z = (x; y) \in \mathbb{R}^{d_x} \times \mathbb{R}^{d_y}$ as follows[7]:

$$\begin{pmatrix} 0 \\ 0 \end{pmatrix} \in A \begin{pmatrix} x \\ y \end{pmatrix} \stackrel{\text{def}}{=} \begin{pmatrix} \partial G(x) + K^\top y \\ \partial F^\star(y) - Kx \end{pmatrix}. \tag{10}$$

Allowing for different stepsizes $\eta_x, \eta_y$ for each block of the vector $z = (x; y)$, PPM applied to (10) takes the form

$$\begin{aligned} x^{k+1} &\in x^k - \eta_x \left( \partial G(x^{k+1}) + K^\top y^{k+1} \right) \\ y^{k+1} &\in y^k - \eta_y \left( \partial F^\star(y^{k+1}) - Kx^{k+1} \right). \end{aligned}$$

The main advantage of this method is its unboundedly fast convergence rate under weak assumptions. According to Theorem 4, the proof of which we provide in the appendix for completeness, if $G$ and $F^\star$ are proper and closed, $G$ is $\mu_x$ strongly convex and $F^\star$ is $\mu_y$ strongly convex, then any choice of stepsizes $\eta_x > 0$ and $\eta_y > 0$ (yes, without an upper bound!), PPM find an $\varepsilon$-accurate solution in

$$\mathcal{O}\left( \left( 1 + \frac{1}{\min\{\eta_x \mu_x, \eta_y \mu_y\}} \right) \log \frac{1}{\varepsilon} \right) \tag{11}$$

iterations. Unfortunately, PPM is not implementable since in order to compute $x^{k+1}$, we need to know $y^{k+1}$, and vice versa.

---

[7]We replaced $\nabla G$ by $\partial G$ here as the beginning of our story does not require $G$ to be smooth.

---
**Algorithm 1** APDA
---
1: **Input**: Initial point $(x^0, y^0) \in \mathbb{R}^{d_x} \times \mathbb{R}^{d_y}$, $\bar{y}^0 = y^0$; Step sizes $\eta_x, \eta_y, \beta_y > 0, \theta \in [0, 1]$
2: **for** $k = 0, 1, \ldots$ **do**
3: $\quad x^{k+1} = x^k - \eta_x \left( \nabla G(x^{k+1}) + K^\top \bar{y}^k \right)$
4: $\quad y^{k+1} \in y^k - \eta_y \left( \partial F^\star(y^{k+1}) - Kx^{k+1} \right) - \eta_y \beta_y K \left( K^\top y^k + \nabla G(x^{k+1}) \right)$
5: $\quad \bar{y}^{k+1} = y^{k+1} + \theta \left( y^{k+1} - y^k \right)$
6: **end for**
---

## 3.3 Chambolle-Pock: Making PPM implementable, and fast

In order to overcome the above problem, Chambolle and Pock [2011] proposed to replace $y^{k+1}$ with $y^k$ (see Algorithm 1 in [Chambolle and Pock, 2011]), which leads to

$$
\begin{aligned}
x^{k+1} &\in x^k - \eta_x \left( \partial G(x^{k+1}) + K^\top y^k \right) \\
y^{k+1} &\in y^k - \eta_y \left( \partial F^\star(y^{k+1}) - Kx^{k+1} \right).
\end{aligned}
$$

Although this method is implementable, it has its own disadvantages, one of which is its weak iteration complexity bound

$$
\mathcal{O} \left( \frac{L_{xy}^2}{\mu_x \mu_y} \log \frac{1}{\varepsilon} \right). \tag{12}
$$

Chambolle and Pock [2011] proposed to fix this problem via an extrapolation step of the dual variable (see Algorithm 3 in [Chambolle and Pock, 2011]):

$$
\begin{aligned}
x^{k+1} &\in x^k - \eta_x \left( \partial G(x^{k+1}) + K^\top \bar{y}^k \right) \\
y^{k+1} &\in y^k - \eta_y \left( \partial F^\star(y^{k+1}) - Kx^{k+1} \right) \\
\bar{y}^{k+1} &= y^{k+1} + \theta(y^{k+1} - y^k).
\end{aligned}
$$

This new method enjoys the much better iteration complexity bound

$$
\mathcal{O} \left( \frac{L_{xy}}{\sqrt{\mu_x \mu_y}} \log \frac{1}{\varepsilon} \right). \tag{13}
$$

## 4 Accelerated Primal-Dual Algorithm (Algorithm 1)

Recall that the Chambolle-Pock method requires $F^\star$ to be strongly-convex to obtain a linear convergence rate. However, in our setting, $F^\star$ is not strongly convex[8] (see Assumption 3), and Chambolle-Pock method does not converge linearly in this scenario.

### 4.1 Modifying Chambolle-Pock to preserve linear rate without strong convexity of $F^\star$

To obtain a linear rate, we modify the dual update step of the algorithm using a trick proposed by Kovalev et al. [2021a] that was shown to work in the regime when $F^\star$ is smooth; the innovation here is that we do not need this assumption (see Table 1). From this point onwards, we will also need to assume $G$ to be $L_x$-smooth (see Assumption 2). In particular, we propose to modify the update step for $y^{k+1}$ in the Chambolle-Pock method as follows:

$$
\begin{aligned}
x^{k+1} &= x^k - \eta_x \left( \nabla G(x^{k+1}) + K^\top \bar{y}^k \right) \\
y^{k+1} &\in y^k - \eta_y \left( \partial F^\star(y^{k+1}) - Kx^{k+1} \right) - \eta_y \beta_y K \left( K^\top y^k + \nabla G(x^{k+1}) \right) \\
\bar{y}^{k+1} &= y^{k+1} + \theta(y^{k+1} - y^k).
\end{aligned}
$$

This is a new method, which we call APDA (formalized as Algorithm 1).

---

[8]We would need to assume $F$ to be smooth to ensure that $F^\star$ is strongly convex. However, we do not want to do this as this is not satisfied in many scenarios, in particular, in our key application to federated learning.

## 4.2 APDA converges linearly

Our first result shows that APDA indeed converges linearly, without the need for $F^\star$ to be strongly convex.

**Theorem 1** (Convergence of APDA; informal). *Let Assumptions 1, 2,3 and 4 hold. Then, with a suitable selection of stepsizes, APDA (Algorithm 1) solves problem* (7) *in*

$$\mathcal{O}\left(\max\left\{\sqrt{\frac{L_x}{\mu_x}}\frac{L_{xy}}{\mu_{xy}}, \frac{L_{xy}^2}{\mu_{xy}^2}\right\}\log\frac{1}{\varepsilon}\right) \tag{14}$$

*iterations.*

The formal statement and proof can be found in the appendix (see Theorem 7).

## 5   Accelerated Primal-Dual Algorithm with Inexact Prox (Algorithm 2)

The key disadvantage of APDA is that it requires the evaluations of the proximity operator of $G$, which can be very expensive in some applications. To remedy the situation, we first notice that Line 3 of APDA can be equivalently written in the form

$$x^{k+1} = \arg\min_{x\in\mathbb{R}^{d_x}}\left\{\Psi^k(x) \stackrel{\text{def}}{=} G(x) + \frac{1}{2\eta_x}\left\|x - \left(x^k - \eta_x K^\top \bar{y}^k\right)\right\|^2\right\}; \tag{15}$$

that is, this step involves the evaluation of the prox of $G$. The key idea of this section is to replace this by an inexact prox computation via a suitably selected iterative method $\mathcal{M}$ (this is the method performing the inner iterations in Table 1). This leads to our next method: APDA with Inexact Prox (Algorithm 2).

---

**Algorithm 2** APDA with Inexact Prox

---

1: **Input**: Initial point $(x^0, y^0) \in \mathbb{R}^{d_x} \times \mathbb{R}^{d_y}$, $\bar{y}^0 = y^0$; Step sizes $\eta_x, \eta_y, \beta_y > 0$, $\theta \in [0,1]$; # inner iterations $T$
2: **for** $k = 0, 1, \ldots$ **do**
3:     Find $\hat{x}^k$ as a final point of $T$ iteration of some method $\mathcal{M}$ for following problem:

$$\hat{x}^k \approx \arg\min_{x\in\mathbb{R}^{d_x}}\left\{\Psi^k(x) \stackrel{\text{def}}{=} G(x) + \frac{1}{2\eta_x}\left\|x - \left(x^k - \eta_x K^\top \bar{y}^k\right)\right\|^2\right\}$$

4:     $x^{k+1} = x^k - \eta_x\left(\nabla G(\hat{x}^k) + K^\top\bar{y}^k\right)$
5:     $y^{k+1} \in y^k - \eta_y\left(\partial F^\star(y^{k+1}) - K\hat{x}^k\right) - \eta_y\beta_y K\left(K^\top y^k + \nabla G(\hat{x}^k)\right)$
6:     $\bar{y}^{k+1} = y^{k+1} + \theta\left(y^{k+1} - y^k\right)$
7: **end for**

---

### 5.1   Gradient methods for finding a stationary point of convex functions

A key feature of Algorithm 2 is its reliance on a subroutine $\mathcal{M}$ for an inexact evaluation of the prox of $G$ via solving the auxiliary problem (15). Our theory requires the method $\mathcal{M}$ to be able to output, after $T$ iterations, a point $\hat{x}^k$ such that

$$\|\nabla\Psi^k(\hat{x}^k)\|^2 \leq \mathcal{O}\left(\frac{1}{T^\alpha}\right), \tag{16}$$

where $\alpha \geq 2$. In other words, we require a reduction of the squared norm of the gradient with a fast sublinear rate. In the next lemma, we present three examples of such methods.

**Lemma 1.** *Let $\Psi : \mathbb{R}^{d_x} \to \mathbb{R}$ be an $L$-smooth convex function, and let $w^\star$ be a minimizer of $\Psi$. Then there exists a gradient-based method $\mathcal{M}$ which after $T$ iterations outputs a point $w^T$ satisfying*

$$\|\nabla\Psi(w^T)\|^2 \leq \frac{AL^2\|w^0 - w^\star\|^2}{T^\alpha}, \tag{17}$$

*for all starting points $x^0 \in \mathbb{R}^{d_x}$ and some universal constant $A > 0$. In particular,*

*(i)* *if $\mathcal{M}$ is GD, then $\|\nabla\Psi(w^T)\|^2 \leq \frac{4L^2\|w^0-w^\star\|^2}{T^2}$,*

*(ii)* *if $\mathcal{M}$ is a combination[9] of Fast GD [Nesterov, 2004] and GD, then $\|\nabla\Psi(w^T)\|^2 \leq \frac{64L^2\|w^0-w^\star\|^2}{T^3}$,*

*(iii)* *if $\mathcal{M}$ is a combination[10] of Fast GD [Nesterov, 2004] and FSFOM [Kim and Fessler, 2021], then $\|\nabla\Psi(w^T)\|^2 \leq \frac{256L^2\|w^0-w^\star\|^2}{T^4}$.*

Let $w^{\star k} = \arg\min_{w\in\mathbb{R}^{d_x}} \Psi^k(w)$. Since $\Psi^k$ is $\left(L_x + \eta_x^{-1}\right)$-smooth, Lemma 1 implies that $T$ iterations of a method $\mathcal{M}$ satisfying (17) applied to the auxiliary problem (15) with starting point $w^0 = x^k$ yield point $w^T = \hat{x}^k$ for which

$$\|\nabla\Psi^k(\hat{x}^k)\|^2 \leq \frac{A\left(\eta_x^{-1} + L_x\right)^2\|x^k - w^{\star k}\|^2}{T^\alpha} = \frac{A\left(1 + \eta_x L_x\right)^2\|x^k - w^{\star k}\|^2}{\eta_x^2 T^\alpha}. \tag{18}$$

## 5.2 APDA with Inexact Prox converges linearly

Now we can provide the main theorem with the total complexity of gradient computation $\nabla G$ and proximity operator computation $\partial F^\star$.

**Theorem 2.** *Let Assumptions 1, 2, 3, 4 hold. Then there exist parameters of Algorithm 2 such that the total # of evaluations of prox $F^\star$ and the total number evaluations of the gradient of $\nabla G$ to find an $\varepsilon$ solution of problem* (7) *are*

$$\mathcal{O}\left(\max\left\{\sqrt{\frac{L_x}{\mu_x}}\frac{L_{xy}}{\mu_{xy}}, \frac{L_{xy}^2}{\mu_{xy}^2}\right\}\log\frac{1}{\varepsilon}\right), \ \mathcal{O}\left(\max\left\{\left(\frac{L_x}{\mu_x}\right)^{\frac{2+\alpha}{2\alpha}}\frac{L_{xy}}{\mu_{xy}}, \sqrt[\alpha]{\frac{L_x}{\mu_x}}\frac{L_{xy}^2}{\mu_{xy}^2}\right\}\log\frac{1}{\varepsilon}\right), \tag{19}$$

*respectively. In particular,*

*(i)* *if the inner method $\mathcal{M}$ is GD, then the total number of $\nabla G$ computations is equal to*

$$\mathcal{O}\left(\max\left\{\frac{L_x}{\mu_x}\frac{L_{xy}}{\mu_{xy}}, \sqrt{\frac{L_x}{\mu_x}}\frac{L_{xy}^2}{\mu_{xy}^2}\right\}\log\frac{1}{\varepsilon}\right), \tag{20}$$

*(ii)* *if the inner method $\mathcal{M}$ is combination of Fast GD and GD, then the total number of $\nabla G$ computations is equal to*

$$\mathcal{O}\left(\max\left\{\left(\frac{L_x}{\mu_x}\right)^{\frac{5}{6}}\frac{L_{xy}}{\mu_{xy}}, \sqrt[3]{\frac{L_x}{\mu_x}}\frac{L_{xy}^2}{\mu_{xy}^2}\right\}\log\frac{1}{\varepsilon}\right), \tag{21}$$

*(iii)* *if the inner method $\mathcal{M}$ is combination of Fast GD and FSFOM, then the total number of $\nabla G$ computations is equal to*

$$\mathcal{O}\left(\max\left\{\left(\frac{L_x}{\mu_x}\right)^{\frac{3}{4}}\frac{L_{xy}}{\mu_{xy}}, \sqrt[4]{\frac{L_x}{\mu_x}}\frac{L_{xy}^2}{\mu_{xy}^2}\right\}\log\frac{1}{\varepsilon}\right). \tag{22}$$

The proof relies on several lemmas; their statements and the proof of the theorem can be found in the appendix.

Note that our way of performing inexact computation of prox of $G$ allows us to keep the same complexity as APDA (Algorithm 1) in terms of the number of evaluations of the prox of $F^\star$. When

---

[9]The first half of the iterations is solved via the Fast Gradient Descent (FGD) method of Nesterov [2004], and the second half via Gradient Descent (GD).

[10]The first half of the iterations is solved via the Fast Gradient Descent (FGD) method of Nesterov [2004], and the second half via the fixed-step first-order method (FSFOM) of Kim and Fessler [2021].

$\mathcal{M}$ is chosen to be Gradient Decent ($\alpha = 2$), the number of computations of the gradient of $\nabla G$, given by (20), is larger than the number of computations of the prox of $G$ for APDA. Fortunately, we can reduce this if GD is replaced with a faster method. For example, if we choose $\mathcal{M}$ to be a simple combination of Fast GD (FGD) and GD, in which case $\alpha = 3$, Theorem 2 says that the number of computations of the gradient of $\nabla G$ is can be reduced to (21) (this choice is mentioned in the second-to-last row of Table 2). A further reduction is possible if we instead employ a combination of FSFOM and Fast GD; see (22) and the last row of Table 2.

# 6 Accelerated Primal Dual Algorithm with Inexact Prox and Accelerated Gossip (Algorithm 5)

In this section, we consider the most significant applications of Algorithm 2: decentralized optimization over a network $\mathcal{G} = (\mathcal{V}, \mathcal{E})$, and federated learning. In particular, we consider the finite-sum optimization problem

$$\min_{x \in \mathbb{R}^d} \sum_{i=1}^n f_i(x)$$

(see (5)) interpreted as follows: $n = |\mathcal{V}|$ is the number of clients/agents in the network. Communication can only happen between clients connected by an edge. The prevalent paradigm in federated learning, where a single server orchestrates communication in rounds, arises as a special case of this with $\mathcal{G}$ being the fully connected network.

If $\hat{W}$ is the Laplacian[11] of graph $\mathcal{G}$, and let $W = \hat{W} \otimes I_{dn}$. Then problem (5) can be rewritten in following equivalent way:

$$\min_{\sqrt{W}\mathbf{x}=0} P(\mathbf{x}) = \min_{\mathbf{x} \in \mathbb{R}^{dn}} P(\mathbf{x}) + \psi(\sqrt{W}\mathbf{x}), \tag{23}$$

where $\mathbf{x}^\top = (x_1^\top, x_2^\top, \dots, x_n^\top)$, $P(\mathbf{x}) = \sum_{i=1}^n f_i(x_i)$ and $\psi(\mathbf{x}) = 0$ iff $\mathbf{x} = 0$, and $\psi(\mathbf{x}) = +\infty$ otherwise. Problem (23) is a special case of (1). By dualizing the nonsmooth (but proper, closed and convex) penalty, we get the equivalent saddle point formulation

$$\min_{\sqrt{W}\mathbf{x}=0} P(\mathbf{x}) = \min_{\mathbf{x} \in \mathbb{R}^{dn}} \max_{\mathbf{y} \in \mathbb{R}^{dn}} \left\{ P(\mathbf{x}) + \langle \mathbf{y}, \sqrt{W}\mathbf{x} \rangle - \psi^\star(\mathbf{y}) \right\}, \tag{24}$$

As we can see, this problem (24) is the particular case of the problem (7). It means that we can solve it by Algorithm 2, for example. Moreover, we do not have to compute the prox of $\psi^\star$ due to the fact that $\psi^\star(\cdot) \equiv 0$, because $\psi(\cdot)$ is the indicator function of $\{0\}$. We thus arrive at the final formulation

$$\min_{\mathbf{x} \in \mathbb{R}^{dn}} \max_{\mathbf{y} \in \mathbb{R}^{dn}} \left\{ P(\mathbf{x}) + \langle \mathbf{y}, \sqrt{W}\mathbf{x} \rangle \right\}. \tag{25}$$

## 6.1 Application of Algorithm 2 to (25)

Before providing the complexity results related to the application of Algorithm 2 to problem (25), note that $L_{xy} = \sqrt{\lambda_{\max}(W)}$ and $\mu_{xy} = \sqrt{\lambda_{\min}^+(W)}$ and define

$$\chi \overset{\text{def}}{=} \frac{\lambda_{\max}(W)}{\lambda_{\min}^+(W)}. \tag{26}$$

According to Theorem 2, the total number of evaluations of the prox of $\psi^\star$, i.e., the communication complexity, and the total number of evaluations of $\nabla P$, i.e., computation complexity, are

$$\sharp\text{comm} = \widetilde{\mathcal{O}}\left(\max\left\{\sqrt{\kappa\chi}, \chi\right\}\right), \quad \sharp\text{comp} = \widetilde{\mathcal{O}}\left(\max\left\{\kappa^{\frac{2+\alpha}{2\alpha}}\sqrt{\chi}, \sqrt[\alpha]{\kappa\chi}\right\}\right), \tag{27}$$

respectively. For example, in centralized case, when $\mathcal{G}$ is the complete graph ($\chi = 1$) and $\mathcal{M}$ is chosen to be GD ($\alpha = 2$), we obtain the same complexities as ProxSkip [Mishchenko et al., 2022]:

$$\sharp\text{comm} = \widetilde{\mathcal{O}}\left(\sqrt{\kappa}\right), \qquad \sharp\text{comp} = \widetilde{\mathcal{O}}(\kappa). \tag{28}$$

---

[11] In fact, it is enough for $\hat{W}$ to satisfy the less restrictive Assumption 6.

However, this can be improved by using a more elaborate subroutine $\mathcal{M}$. If instead of GD we use either FGD + GD ($\alpha = 3$) or FGD + FSFOM ($\alpha = 4$) in place of $\mathcal{M}$, the number of communication rounds will be the same as in the case of ProxSkip (or Algorithm 2 used with $\mathcal{M} = $ GD). However, the total number of gradient computations gets improved to $\sharp\text{comp} = \widetilde{\mathcal{O}}\left(\kappa^{\frac{5}{6}}\right)$ in the first case, and to $\sharp\text{comp} = \widetilde{\mathcal{O}}\left(\kappa^{\frac{3}{4}}\right)$ in the second case.

## 6.2 Improvement on Algorithm 2 via accelerated gossip

Compared to the ProxSkip computation complexity $\widetilde{\mathcal{O}}\left(\sqrt{\kappa\chi}\right)$, in the general decentralized case (i.e., $\chi > 1$), complexity (27) of our Algorithm 2 is worse if $\kappa \leq \chi$. To tackle this problem, we propose to enhance Algorithm 2 using the accelerated gossip technique [Scaman et al., 2017]. Based on this approach, we propose one more (and final) method: Algorithm 5. For this method, we prove the following result.

**Theorem 3.** *Let Assumptions 1 and 2 hold for function $P$. Then there exist parameters of Algorithm 5 such that in order to find an $\varepsilon$-solution of problem* (7)*, the total number of communications and the total number of gradient computations can be bounded by*

$$\sharp comm = \widetilde{\mathcal{O}}\left(\sqrt{\kappa\chi}\right), \qquad \sharp comp = \widetilde{\mathcal{O}}\left(\kappa^{\frac{2+\alpha}{2\alpha}}\right), \tag{29}$$

*respectively. In particular, if the inner method $\mathcal{M}$ is*

(i) *GD, then the number of local steps is equal to $\widetilde{\mathcal{O}}\left(\kappa^{1/2}\right)$ and the total number of gradient computations is equal to $\widetilde{\mathcal{O}}\left(\kappa\right)$;*

(ii) *FGD + GD, then the number of local steps is equal to $\widetilde{\mathcal{O}}\left(\kappa^{1/3}\right)$ and the total number of gradient computations is equal to $\widetilde{\mathcal{O}}\left(\kappa^{5/6}\right)$;*

(iii) *FGD + FSFOM, then the number of local steps is equal to $\widetilde{\mathcal{O}}\left(\kappa^{1/4}\right)$ and the total number of gradient computations is equal to $\widetilde{\mathcal{O}}\left(\kappa^{3/4}\right)$.*

The communication complexities obtained this way are substantially better than those of decentralized ProxSkip; see Table 3 and the commentary in the next subsection.

Recall that in Table 2, we already compared ProxSkip and our Algorithm 2 in the centralized case. In Table 3 we add to this a comparison in the decentralized case, and include our Algorithm 5 as well. In particular, in Table 3 we compare the complexity results of our methods (Algorithms 2 and 5) for solving the decentralized optimization problem (5) to two selected benchmarks: D-SGD [Koloskova et al., 2020] and ProxSkip [Mishchenko et al., 2022].

First, observe that ProxSkip has vastly superior communication complexity to D-SGD, both in the centralized case (i.e., for fully-connected network; $\chi = 1$), where the improvement is from $\mathcal{O}(\kappa)$ to $\widetilde{\mathcal{O}}(\sqrt{\kappa})$, and the decentralized case ($\chi > 1$), where the improvement is from $\widetilde{\mathcal{O}}(\kappa\chi)$ to $\widetilde{\mathcal{O}}(\sqrt{\kappa\chi})$.

In the decentralized case, both our methods match the $\widetilde{\mathcal{O}}(\sqrt{\kappa\chi})$ communication complexity of ProxSkip. However, Algorithm 2 does so only when $\sqrt{\kappa\chi} \geq \chi$ (i.e., when $\kappa > \chi$). On the other hand, both our methods have an improved bound on the number of local gradient computations, depending on what subroutine $\mathcal{M}$ they employ. The improvement is from $\widetilde{\mathcal{O}}(\sqrt{\kappa})$ to $\widetilde{\mathcal{O}}\left(\sqrt[3]{\kappa}\right)$ (when $\mathcal{M} = $ FGD + GD) to $\widetilde{\mathcal{O}}\left(\sqrt[4]{\kappa}\right)$ (when $\mathcal{M} = $ FGD + FSFOM).

**Acknowledgments and Disclosure of Funding**

The work of all authors was supported by the KAUST Baseline Research Grant awarded to P. Richtárik. The work of A. Sadiev was supported by the Visiting Student Research Program (VSRP) at KAUST. The work of A. Sadiev was also partially supported by a grant for research centers in the field of artificial intelligence, provided by the Analytical Center for the Government of the Russian Federation in accordance with the subsidy agreement (agreement identifier 000000D730321P5Q0002 ) and the agreement with the Ivannikov Institute for System Programming of the Russian Academy of Sciences dated November 2, 2021 No. 70-2021-00142.

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
