# Appendix

## Contents

# A    Additional Related Work

An alternative approach to achieving improved communication efficiency is through the use of communication compression via (unbiased) quantization [Zhu et al., 2017, Alistarh et al., 2017, Horváth et al., 2019a, Mishchenko et al., 2021b], sketching [Hanzely et al., 2018, Safaryan et al., 2021] or sparsification [Wangni et al., 2017, Wang et al., 2018, Mishchenko et al., 2020]. Modern variants offering with variance reduction for the variance caused by compression Mishchenko et al. [2019], Horváth et al. [2019b], Gorbunov et al. [2020b], Safaryan et al. [2021], Mishchenko et al. [2021b], Wang et al. [2021a], Shulgin and Richtárik [2021], adaptivity [Mishchenko et al., 2021b], bidirectional compression [Horváth et al., 2019a, Tang et al., 2019, Philippenko and Dieuleveut, 2020] or acceleration Li et al. [2020c] enjoy better theoretical rates and practical performance. Variance-reduction for communication compression has been extended to work over arbitrary connected networks [Kovalev et al., 2021b], and to second-order methods [Islamov et al., 2021, Safaryan et al., 2022, Qian et al., 2022, Islamov et al., 2022]. The current state-of-the-art communication complexity in the smooth nonconvex regime is offered by the MARINA [Gorbunov et al., 2021, Szlendak et al., 2022] and DASHA [Tyurin and Richtárik, 2022] methods.

Greedy (biased) compressors, such as Top-K sparsification [Alistarh et al., 2018] or Rank-K approximation [Vogels et al., 2019], require a different approach via an error-feedback/compensation mechanism [Stich et al., 2018, Stich and Karimireddy, 2019, Tang et al., 2019, Gorbunov et al., 2020c]. For a more modern treatment of error feedback offering current state-of-the-art rates, we refer the reader to [Richtárik et al., 2021, Fatkhullin et al., 2021, Richtárik et al., 2022, Qian et al., 2020]. An alternative approach based on the transformation of a biased compressor into a related induced unbiased compressor was proposed in [Horváth and Richtárik, 2021], and a unified treatment of variance reduction and error-feedback was proposed in [Condat et al., 2022].

For a systems-oriented survey, we recommend the reader the work of Xu et al. [2020].

In this work we do not consider the communication compression approach to communication efficiency since this area is much more understood, and many methods already improve on the theoretical communication complexity of vanilla GD and SGD, often by significant data and dimension-dependent margins. Instead, we focus on the practice of delayed parameter synchronization via local training, and contribute to the theoretical foundations of this immensely popular yet poorly understood approach to achieving communication efficiency.

## A.1    Summary of complexity results for Algorithms 2 and 5 applied to decentralized optimization

In this section we provide the brief summery of our obtained results (comperison between Algorithms 2 , 5 applied to decentralized optimization) and previous ProxSkip of Mishchenko et al. [2022], D-SGD of Koloskova et al. [2020].

# B    Analysis of the Proximal-Point Method

In this section we justify the claims we made in Section 3.2 about the Proximal-Point Method (Algorithm 3). In particular, we prove the complexity result (11). The result is not new of course, but we could not find a source for the proof we include here.

## B.1    The Proximal-Point Method

We have described the Proximal-Point Method informally in Section 3.2. Here we state it formally as Algorithm 3.

## B.2    Bonding the distance of the primal iterates to the primal solution

In our first lemma, we will provide a bound on $\|x^{k+1} - x^\star\|^2$.

**Lemma 2.** *Let Assumption 1 hold and choose any* $\eta_x, \eta_y > 0$. *Then the iterates of the Proximal-Point Method (Algorithm 3) for all* $k \geq 0$ *satisfy*

$$(1 + 2\mu_x\eta_x)\frac{1}{\eta_x}\|x^{k+1} - x^\star\|^2 \leq \frac{1}{\eta_x}\|x^k - x^\star\|^2 - \frac{1}{\eta_x}\|x^{k+1} - x^k\|^2 - 2\langle K^\top y^{k+1} - K^\top y^\star, x^{k+1} - x^\star\rangle.$$

Table 3: Complexity results for Algorithms 2 and 5 applied to solving the decentralized optimization problem (5) formulated as the saddle-point problem (25). Our results improve upon those of ProxSkip, both in communication complexity (for Algorithm 5), and the number of local gradient steps (for both our methods, given a proper choice of $\mathcal{M}$).

| Alg. | Inner Method $\mathcal{M}$ | Deter-ministic #local steps | Centralized case | | Decentralized case | | |
|---|---|---|---|---|---|---|---|
| | | | #comm. rounds | #local steps per round | GA[1] | #comm. rounds | #local steps per round |
| D-SGD[3] | GD | ✓ | $\widetilde{\mathcal{O}}(\kappa)$ | $\tau$ | ✗ | $\widetilde{\mathcal{O}}(\kappa\chi)$ | $\tau$ |
| ProxSkip[4] | GD | ✗ | $\widetilde{\mathcal{O}}(\sqrt{\kappa})$ | $\mathcal{O}(\sqrt{\kappa})$ | ✗ | $\widetilde{\mathcal{O}}(\sqrt{\kappa\chi})^{(1)}$ | $\widetilde{\mathcal{O}}(\sqrt{\kappa})^{(2)}$ |
| Alg. 2 | GD | ✓ | $\widetilde{\mathcal{O}}(\sqrt{\kappa})$ | $\widetilde{\mathcal{O}}(\sqrt{\kappa})$ | ✗ | $\widetilde{\mathcal{O}}(\sqrt{\kappa\chi} \vee \chi)^{(5)}$ | $\widetilde{\mathcal{O}}(\sqrt{\kappa})$ |
| | FGD + GD | ✓ | | $\widetilde{\mathcal{O}}(\sqrt[3]{\kappa})$ | ✗ | | $\widetilde{\mathcal{O}}(\sqrt[3]{\kappa})$ |
| | FGD + FSFOM | ✓ | | $\widetilde{\mathcal{O}}(\sqrt[4]{\kappa})$ | ✗ | | $\widetilde{\mathcal{O}}(\sqrt[4]{\kappa})$ |
| Alg. 5 | GD | ✓ | $\widetilde{\mathcal{O}}(\sqrt{\kappa})$ | $\widetilde{\mathcal{O}}(\sqrt{\kappa})$ | ✓ | $\widetilde{\mathcal{O}}(\sqrt{\kappa\chi})$ | $\widetilde{\mathcal{O}}(\sqrt{\kappa})$ |
| | FGD + GD | ✓ | | $\widetilde{\mathcal{O}}(\sqrt[3]{\kappa})$ | ✓ | | $\widetilde{\mathcal{O}}(\sqrt[3]{\kappa})$ |
| | FGD + FSFOM | ✓ | | $\widetilde{\mathcal{O}}(\sqrt[4]{\kappa})$ | ✓ | | $\widetilde{\mathcal{O}}(\sqrt[4]{\kappa})$ |

[1] Does not use Accelerated Gossip technique (AG).
[2] Valid only when $\kappa \leq \chi$.
[3] This method was analyzed by Koloskova et al. [2020].
[4] This method was proposed and analyzed by Mishchenko et al. [2022].
[5] For $a, b \in \mathbb{R}$, we denote $a \vee b = \max\{a, b\}$

---

**Algorithm 3** Proximal-Point Method

1: **Input**: Initial point $(x^0, y^0) \in \mathbb{R}^{d_x} \times \mathbb{R}^{d_y}$; Stepsizes $\eta_x, \eta_y > 0$
2: **for** $k = 0, 1, \dots$ **do**
3: $\quad x^{k+1} = x^k - \eta_x \left( \partial G(x^{k+1}) + K^\top y^{k+1} \right)$
4: $\quad y^{k+1} = y^k - \eta_y \left( \partial F^\star(y^{k+1}) - K x^{k+1} \right)$
5: **end for**

---

*Proof.* By writing $x^{k+1}$ as $x^k + (x^{k+1} - x^k)$, and using line 3 of Algorithm 3, which reads $x^{k+1} = x^k - \eta_x \left( \partial G(x^{k+1}) + K^\top y^{k+1} \right)$, we get

$$
\begin{aligned}
\frac{1}{\eta_x}\|x^{k+1} - x^\star\|^2 &= \frac{1}{\eta_x}\|x^k - x^\star\|^2 + \frac{2}{\eta_x}\langle x^{k+1} - x^k, x^k - x^\star \rangle + \frac{1}{\eta_x}\|x^{k+1} - x^k\|^2 \\
&= \frac{1}{\eta_x}\|x^k - x^\star\|^2 + \frac{2}{\eta_x}\langle x^{k+1} - x^k, x^{k+1} - x^\star \rangle - \frac{1}{\eta_x}\|x^{k+1} - x^k\|^2 \\
&= \frac{1}{\eta_x}\|x^k - x^\star\|^2 - 2\langle \partial G(x^{k+1}) + K^\top y^{k+1}, x^{k+1} - x^\star \rangle - \frac{1}{\eta_x}\|x^{k+1} - x^k\|^2.
\end{aligned}
$$

We now split the inner product into two parts by applying the optimality condition (see (8))

$$
0 \in \partial G(x^\star) + K^\top y^\star,
$$

obtaining[12]

$$
\begin{aligned}
\frac{1}{\eta_x}\|x^{k+1} - x^\star\|^2 &= \frac{1}{\eta_x}\|x^k - x^\star\|^2 - 2\langle \partial G(x^{k+1}) - \partial G(x^\star), x^{k+1} - x^\star \rangle - \frac{1}{\eta_x}\|x^{k+1} - x^k\|^2 \\
&\quad - 2\langle K^\top y^{k+1} - K^\top y^\star, x^{k+1} - x^\star \rangle.
\end{aligned}
$$

Finally, this allows us to replace the first inner product using strong convexity of $G$ as follows:

$$
\begin{aligned}
\frac{1}{\eta_x}\|x^{k+1} - x^\star\|^2 &\leq \frac{1}{\eta_x}\|x^k - x^\star\|^2 - 2\mu_x\|x^{k+1} - x^\star\|^2 - \frac{1}{\eta_x}\|x^{k+1} - x^k\|^2 \\
&\quad - 2\langle K^\top y^{k+1} - K^\top y^\star, x^{k+1} - x^\star \rangle.
\end{aligned}
$$

---

[12]Abusing notation, here by $\partial G(x^\star)$ we refer to the subgradient $g_G(x^\star) \in \partial G(x^\star)$ for which $0 = g_G(x^\star) + K^\top y^\star$, i.e., $g_G(x^\star) = -K^\top y^\star$.

$\square$

## B.3 Bonding the distance of the dual iterates to the dual solution

In our second lemma, we will provide a bound on $\|y^{k+1} - y^\star\|^2$. For this, we will rely on an additional assumption (strong convexity of $F^\star$):

**Assumption 5.** *The function* $F^\star : \mathbb{R}^{d_y} \to \mathbb{R}$ *is* $\mu_y$*-strongly convex (but can be non-differentiable), i.e.,*

$$F^\star(y') - F^\star(y'') - \langle g_{F^\star}(y''), y' - y'' \rangle \geq \frac{\mu_y}{2} \|y' - y''\|^2, \qquad \forall y', y'' \in \mathbb{R}^{d_y}, \qquad (30)$$

*where* $g_{F^\star}(y'') \in \partial F^\star(y'')$ *is any subgradient of* $F^\star$ *at* $y'' \in \mathbb{R}^{d_y}$.

**Lemma 3.** *Let Assumptions 3 and 5 hold and choose any* $\eta_x, \eta_y > 0$. *Then the iterates of the Proximal-Point Method (Algorithm 3) for all* $k \geq 0$ *satisfy*

$$(1 + 2\mu_y\eta_y)\frac{1}{\eta_y}\|y^{k+1} - y^\star\|^2 \leq \frac{1}{\eta_y}\|y^k - y^\star\|^2 - \frac{1}{\eta_y}\|y^{k+1} - y^k\|^2 + 2\langle K^\top y^{k+1} - K^\top y^\star, x^{k+1} - x^\star \rangle.$$

*Proof.* By writing $y^{k+1}$ as $y^k + (y^{k+1} - y^k)$, and using line 4 of Algorithm 3, which reads $y^{k+1} = y^k - \eta_y \left( \partial F^\star(y^{k+1}) - Kx^{k+1} \right)$, we get

$$
\begin{aligned}
\frac{1}{\eta_y}\|y^{k+1} - y^\star\|^2 &= \frac{1}{\eta_y}\|y^k - y^\star\|^2 + \frac{2}{\eta_y}\langle y^{k+1} - y^k, y^k - y^\star \rangle + \frac{1}{\eta_y}\|y^{k+1} - y^k\|^2 \\
&= \frac{1}{\eta_y}\|y^k - y^\star\|^2 + \frac{2}{\eta_y}\langle y^{k+1} - y^k, y^{k+1} - y^\star \rangle - \frac{1}{\eta_y}\|y^{k+1} - y^k\|^2 \\
&= \frac{1}{\eta_y}\|y^k - y^\star\|^2 - 2\langle \partial F^\star(y^{k+1}) - Kx^{k+1}, y^{k+1} - y^\star \rangle - \frac{1}{\eta_y}\|y^{k+1} - y^k\|^2.
\end{aligned}
$$

We now split the inner product into two parts by applying the optimality condition (see (8))

$$0 \in \partial F^\star(y^\star) - Kx^\star,$$

obtaining[13]

$$
\begin{aligned}
\frac{1}{\eta_y}\|y^{k+1} - y^\star\|^2 &= \frac{1}{\eta_y}\|y^k - y^\star\|^2 - 2\langle \partial F^\star(y^{k+1}) - \partial F^\star(y^\star), y^{k+1} - y^\star \rangle - \frac{1}{\eta_y}\|y^{k+1} - y^k\|^2 \\
&\quad + 2\langle Kx^{k+1} - Kx^\star, y^{k+1} - y^\star \rangle.
\end{aligned}
$$

Finally, this allows us to replace the first inner product using strong convexity of $F^\star$ as follows:

$$
\begin{aligned}
\frac{1}{\eta_y}\|y^{k+1} - y^\star\|^2 &\leq \frac{1}{\eta_y}\|y^k - y^\star\|^2 - 2\mu_y\|y^{k+1} - y^\star\|^2 - \frac{1}{\eta_y}\|y^{k+1} - y^k\|^2 \\
&\quad + 2\langle K^\top y^{k+1} - K^\top y^\star, x^{k+1} - x^\star \rangle.
\end{aligned}
$$

$\square$

## B.4 Complexity of the Proximal-Point Method

We now formulate the main result describing the iteration complexity of the Proximal-Point Method. As we shall see, it follows by combining the above two lemmas. Note that the theorem postulates *arbitrarily fast linear convergence*, with the speed controlled via the primal and dual stepsizes $\eta_x$ and $\eta_y$, respectively. The larger the stepsizes, the faster the rate becomes. In the limit, as $\eta_x \to +\infty$ and $\eta_y \to +\infty$, the result obtained predicts convergence in a single iteration.

**Theorem 4.** *Let Assumptions 1 and 5 hold, and choose any* $\eta_x, \eta_y > 0$. *Then the iterates of the Proximal-Point Method (Algorithm 3) for all* $k \geq 0$ *satisfy*

$$\Delta^{k+1} \leq \frac{\Delta^k}{\min\{1 + 2\mu_x\eta_x, 1 + 2\mu_y\eta_y\}},$$

---

[13]Abusing notation, here by $\partial F^\star(y^\star)$ we refer to the subgradient $g_{F^\star}(y^\star) \in \partial F^\star(y^\star)$ for which $0 = g_{F^\star}(y^\star) - Kx^\star$, i.e., $g_{F^\star}(y^\star) = Kx^\star$.

*where the Lyapunov function is defined by*

$$\Delta^k \stackrel{def}{=} \frac{1}{\eta_x}\|x^k - x^\star\|^2 + \frac{1}{\eta_y}\|y^k - y^\star\|^2.$$

*This implies the following statement:*

$$k \geq \left(1 + \frac{1}{\min\{2\mu_x\eta_x, 2\mu_y\eta_y\}}\right)\log\frac{1}{\varepsilon} \qquad \Rightarrow \qquad \Delta^k \leq \varepsilon\Delta^0.$$

*Proof.* By adding the inequalities from Lemma 2 and Lemma 3, we obtain

$$(1 + 2\mu_x\eta_x)\frac{1}{\eta_x}\|x^{k+1} - x^\star\|^2 \quad + \quad (1 + 2\mu_y\eta_y)\frac{1}{\eta_y}\|y^{k+1} - y^\star\|^2$$

$$\leq \quad \frac{1}{\eta_x}\|x^k - x^\star\|^2 - \frac{1}{\eta_x}\|x^{k+1} - x^k\|^2 + \frac{1}{\eta_y}\|y^k - y^\star\|^2 - \frac{1}{\eta_y}\|y^{k+1} - y^k\|^2$$

$$-2\langle K^\top y^{k+1} - K^\top y^\star, x^{k+1} - x^\star\rangle + 2\langle K^\top y^{k+1} - K^\top y^\star, x^{k+1} - x^\star\rangle$$

$$\leq \quad \frac{1}{\eta_x}\|x^k - x^\star\|^2 + \frac{1}{\eta_y}\|y^k - y^\star\|^2.$$

Thus, if we denote $\Delta^k \stackrel{def}{=} \frac{1}{\eta_x}\|x^k - x^\star\|^2 + \frac{1}{\eta_y}\|y^k - y^\star\|^2$ and $m \stackrel{def}{=} \min\{1 + 2\mu_x\eta_x, 1 + 2\mu_y\eta_y\}$, the above inequality can be written in the following way:

$$\Delta^{k+1} \leq \frac{1}{m}\Delta^k = \left(1 - \frac{m-1}{m}\right)\Delta^k.$$

Using standard arguments, the above implies that

$$k \geq \frac{m}{m-1}\log\frac{1}{\varepsilon} \quad \Rightarrow \quad \Delta^k \leq \varepsilon\Delta^0.$$

$$\square$$

# C   Analysis of the Chambolle-Pock Method

In this section we justify the claims we made in Section 3.3 about the Chambolle-Pock Method (Algorithm 4). In particular, we provide formal statements and proofs of the complexity results (12) and (13) mentioned in Sections C.4 and C.5, respectively.

## C.1   The Chambolle-Pock Method

We have described the Chambolle-Pock Method informally in Section 3.2. Here we state it formally as Algorithm 4.

---

**Algorithm 4** Chambolle-Pock Method [Chambolle and Pock, 2011]

---

1: **Input**: Initial point $(x^0, y^0) \in \mathbb{R}^{d_x} \times \mathbb{R}^{d_y}$, $\bar{y}^0 = y^0$; Stepsizes $\eta_x, \eta_y > 0$, Extrapolation parameter $\theta \in [0, 1]$
2: **for** $k = 0, 1, \ldots$ **do**
3:     $x^{k+1} = x^k - \eta_x \left( \partial G(x^{k+1}) + K^\top \bar{y}^k \right)$
4:     $y^{k+1} = y^k - \eta_y \left( \partial F^\star(y^{k+1}) - K x^{k+1} \right)$
5:     $\bar{y}^{k+1} = y^{k+1} + \theta \left( y^{k+1} - y^k \right)$
6: **end for**

---

## C.2   Bonding the distance of the primal iterates to the primal solution

In our first lemma, we will provide a bound on $\|x^{k+1} - x^\star\|^2$. The result is identical to Lemma 2 with a single exception: instead of the fresh dual point $y^{k+1}$, we now have $\bar{y}^k$ on the right-hand side. This lemma applies both to the $\theta = 0$ and $\theta > 0$ cases.

**Lemma 4.** *Let Assumption 1 hold and choose any $\eta_x > 0$. Then the iterates of the Chambolle-Pock Method (Algorithm 4) for all $k \geq 0$ satisfy*

$$\left(1 + 2\mu_x \eta_x\right) \frac{1}{\eta_x} \|x^{k+1} - x^\star\|^2 \leq \frac{1}{\eta_x} \|x^k - x^\star\|^2 - \frac{1}{\eta_x} \|x^{k+1} - x^k\|^2 - 2\langle K^\top \bar{y}^k - K^\top y^\star, x^{k+1} - x^\star \rangle.$$

*Proof.* Identical to the proof of Lemma 2; one just needs to replace $y^{k+1}$ by $\bar{y}^k$ everywhere. □

## C.3   Bonding the distance of the dual iterates to dual solution

We do not need a new bound on $\|y^{k+1} - y^\star\|^2$ for Algorithm 4 since Lemma 3 we proved for the Proximal-Point Method applies here as well.

## C.4   Formal statement and proof of (12) (Chambolle-Pock in the $\theta = 0$ case)

We are now ready to state and prove the main complexity result for the Chambolle-Pock Method in the $\theta = 0$ case. This is the formal version of the informal complexity result (12).

**Theorem 5** (Complexity of the Chambolle-Pock Method in the $\theta = 0$ case)**.** *Let Assumptions 1, 5 hold. Consider the Chambolle-Pock Method (Algorithm 4) with the extrapolation parameter set as*

$$\theta = 0,$$

*and the primal and dual stepsizes set as*

$$\eta_x = \frac{\mu_y}{L_{xy}^2}, \qquad \eta_y = \frac{\mu_x}{L_{xy}^2}.$$

*Then for the Lyapunov function*

$$\Delta^k \stackrel{def}{=} \frac{1}{\eta_x} \|x^k - x^\star\|^2 + \frac{1}{\eta_y} \|y^k - y^\star\|^2$$

*and all $k \geq 0$ we have*

$$\Delta^{k+1} \leq \frac{\Delta^k}{\min\left\{1 + \mu_x \eta_x, 1 + 2\mu_y \eta_y\right\}} = \frac{\Delta^k}{1 + \frac{\mu_x \mu_y}{L_{xy}^2}}.$$

*This implies the following statement:*

$$k \geq \left(1 + \frac{1}{\min\left\{\mu_x\eta_x, 2\mu_y\eta_y\right\}}\right)\log\frac{1}{\varepsilon} = \left(1 + \frac{L_{xy}^2}{\mu_x\mu_y}\right)\log\frac{1}{\varepsilon} \qquad \Rightarrow \qquad \Delta^k \leq \varepsilon\Delta^0.$$

*Proof.* By adding the inequalities from Lemma 4 (and noting that $\bar{y}^k = y^k$ in the $\theta = 0$ case) and Lemma 3, we obtain

$$(1 + 2\mu_x\eta_x)\frac{1}{\eta_x}\|x^{k+1} - x^\star\|^2 \quad + \quad (1 + 2\mu_y\eta_y)\frac{1}{\eta_y}\|y^{k+1} - y^\star\|^2$$

$$\leq \quad \frac{1}{\eta_x}\|x^k - x^\star\|^2 - \frac{1}{\eta_x}\|x^{k+1} - x^k\|^2 + \frac{1}{\eta_y}\|y^k - y^\star\|^2 - \frac{1}{\eta_y}\|y^{k+1} - y^k\|^2$$

$$- \quad 2\langle K^\top y^k - K^\top y^\star, x^{k+1} - x^\star\rangle + 2\langle K^\top y^{k+1} - K^\top y^\star, x^{k+1} - x^\star\rangle \quad (31)$$

$$\leq \quad \frac{1}{\eta_x}\|x^k - x^\star\|^2 + \frac{1}{\eta_y}\|y^k - y^\star\|^2$$

$$- \quad \frac{1}{\eta_y}\|y^{k+1} - y^k\|^2 + 2\langle K^\top y^{k+1} - K^\top y^k, x^{k+1} - x^\star\rangle, \qquad (32)$$

where in the last step we have used the bound $-\frac{1}{\eta_x}\|x^{k+1} - x^k\|^2 \leq 0$.

Using the Cauchy-Schwarz inequality, the definition of $L_{xy}$ as the norm of $K$ (see (2)), and applying Young's inequality, we can bound the inner product by

$$2\langle K^\top y^{k+1} - K^\top y^k, x^{k+1} - x^\star\rangle \quad \leq \quad 2\|K^\top y^{k+1} - K^\top y^k\|\|x^{k+1} - x^\star\|$$

$$\overset{(2)}{\leq} \quad 2L_{xy}\|y^{k+1} - y^k\|\|x^{k+1} - x^\star\|$$

$$\leq \quad L_{xy}\left(C\|y^{k+1} - y^k\|^2 + \frac{1}{C}\|x^{k+1} - x^\star\|^2\right),$$

for any $C > 0$. Plugging this into (32), and rearranging the inequality so that all terms involving $\|x^{k+1} - x^\star\|^2$ appear on the left-hand side, we get

$$\left(1 + 2\mu_x\eta_x - \frac{L_{xy}\eta_x}{C}\right)\frac{1}{\eta_x}\|x^{k+1} - x^\star\|^2 \quad + \quad (1 + 2\mu_y\eta_y)\frac{1}{\eta_y}\|y^{k+1} - y^\star\|^2$$

$$\leq \quad \frac{1}{\eta_x}\|x^k - x^\star\|^2 + \frac{1}{\eta_y}\|y^k - y^\star\|^2$$

$$- (1 - CL_{xy}\eta_y)\frac{1}{\eta_y}\|y^{k+1} - y^k\|^2.$$

Taking $C = \frac{L_{xy}}{\mu_x}$ and $\eta_y = \frac{\mu_x}{L_{xy}^2}$, we obtain the simplified bound

$$(1 + \mu_x\eta_x)\frac{1}{\eta_x}\|x^{k+1} - x^\star\|^2 \quad + \quad (1 + 2\mu_y\eta_y)\frac{1}{\eta_y}\|y^{k+1} - y^\star\|^2$$

$$\leq \quad \frac{1}{\eta_x}\|x^k - x^\star\|^2 + \frac{1}{\eta_y}\|y^k - y^\star\|^2 - \left(1 - \frac{L_{xy}^2\eta_y}{\mu_x}\right)\frac{1}{\eta_y}\|y^{k+1} - y^k\|^2$$

$$\leq \quad \frac{1}{\eta_x}\|x^k - x^\star\|^2 + \frac{1}{\eta_y}\|y^k - y^\star\|^2. \qquad (33)$$

If we let

$$m \overset{\text{def}}{=} \min\left\{1 + \mu_x\eta_x, 1 + 2\mu_y\eta_y\right\} = \min\left\{1 + \frac{\mu_x\mu_y}{L_{xy}^2}, 1 + 2\frac{\mu_x\mu_y}{L_{xy}^2}\right\} = 1 + \frac{\mu_x\mu_y}{L_{xy}^2},$$

then inequality (33) implies

$$\Delta^{k+1} \leq \frac{1}{m}\Delta^k = \left(1 - \frac{m-1}{m}\right)\Delta^k \qquad \forall k \geq 0.$$

Using standard arguments, the above implies that

$$k \geq \frac{m}{m-1} \log \frac{1}{\varepsilon} = \left(1 + \frac{1}{m-1}\right) \log \frac{1}{\varepsilon} = \left(1 + \frac{L_{xy}^2}{\mu_x \mu_y}\right) \log \frac{1}{\varepsilon} \qquad \Rightarrow \qquad \Delta^k \leq \varepsilon \Delta^0.$$

$\square$

### C.5 Formal statement and proof of (13) (Chambolle-Pock in the $\theta > 0$ case)

We now study the case when the extrapolation parameter $\theta$ is set to a positive value, and show that this helps to get better rates.

**Lemma 5.** *Under Assumption 1, the iterates of the Chambolle-Pock Method (Algorithm 4) with $\theta > 0$ for any $C > 0$ and all $k \geq 1$ satisfy*

$$
\begin{aligned}
2\langle K^\top y^{k+1} - K^\top \bar{y}^k, x^{k+1} - x^\star \rangle \quad \leq \quad & 2\langle K^\top y^{k+1} - K^\top y^k, x^{k+1} - x^\star \rangle \\
& -2\theta \langle K^\top y^k - K^\top y^{k-1}, x^k - x^\star \rangle \\
& +\theta L_{xy} C \|x^{k+1} - x^k\|^2 + \frac{\theta L_{xy}}{C} \|y^k - y^{k-1}\|^2. \quad (34)
\end{aligned}
$$

*Proof.* Using line 5 from Algorithm 4, which reads $\bar{y}^k = y^k + \theta\left(y^k - y^{k-1}\right)$, we obtain

$$
\begin{aligned}
2\langle K^\top y^{k+1} - K^\top \bar{y}^k, x^{k+1} - x^\star \rangle \quad = \quad & 2\langle K^\top y^{k+1} - K^\top y^k, x^{k+1} - x^\star \rangle \\
& -2\theta \langle K^\top y^k - K^\top y^{k-1}, x^{k+1} - x^\star \rangle \\
= \quad & 2\langle K^\top y^{k+1} - K^\top y^k, x^{k+1} - x^\star \rangle \\
& -2\theta \langle K^\top y^k - K^\top y^{k-1}, x^k - x^\star \rangle \\
& -2\theta \langle K^\top y^k - K^\top y^{k-1}, x^{k+1} - x^k \rangle. \quad (35)
\end{aligned}
$$

Using the Cauchy-Schwarz inequality, the definition of $L_{xy}$ as the norm of $K$ (see (2)), and applying Young's inequality, we can bound the last inner product by

$$
\begin{aligned}
-2\theta \langle K^\top y^k - K^\top y^{k-1}, x^{k+1} - x^k \rangle \quad \leq \quad & 2\theta \| K^\top y^k - K^\top y^{k-1} \| \| x^{k+1} - x^k \| \\
\leq \quad & 2\theta L_{xy} \| y^k - y^{k-1} \| \| x^{k+1} - x^k \| \\
\leq \quad & \theta L_{xy} \left( C \| y^k - y^{k-1} \|^2 + \frac{1}{C} \| x^{k+1} - x^k \|^2 \right).
\end{aligned}
$$

It only remains to plug this inequality into (35).

$\square$

We are now ready to state and prove the general theorem.

**Theorem 6** (Complexity of the Chambolle-Pock Method in the $\theta > 0$ case). *Let Assumptions 1, 5 hold. Consider the Chambolle-Pock Method (Algorithm 4) with the extrapolation parameter set as*

$$\theta = \max\left\{ \frac{1}{1 + 2\mu_x \eta_x}, \frac{1}{1 + 2\mu_y \eta_y} \right\}, \quad (36)$$

*and primal and dual stepsizes set as*

$$\eta_x = \frac{1}{L_{xy}} \sqrt{\frac{\mu_y}{\mu_x}}, \qquad \eta_y = \frac{1}{L_{xy}} \sqrt{\frac{\mu_x}{\mu_y}}. \quad (37)$$

*Then for the Lyapunov function defined for $k \geq 0$ via*

$$
\begin{aligned}
\Delta^{k+1} \quad \overset{def}{=} \quad & (1 + 2\mu_x \eta_x) \frac{1}{\eta_x} \|x^{k+1} - x^\star\|^2 + (1 + 2\mu_y \eta_y) \frac{1}{\eta_y} \|y^{k+1} - y^\star\|^2 \\
& + \frac{1}{\eta_y} \|y^{k+1} - y^k\|^2 - 2\langle K^\top y^{k+1} - K^\top y^k, x^{k+1} - x^\star \rangle \quad (38)
\end{aligned}
$$

*and for $k = 0$ via*

$$\Delta^0 \overset{def}{=} (1 + 2\mu_x\eta_x)\frac{1}{\eta_x}\|x^0 - x^\star\|^2 + (1 + 2\mu_y\eta_y)\frac{1}{\eta_y}\|y^0 - y^\star\|^2, \tag{39}$$

*we have*

$$0 \le \Delta^k \le \theta^k\Delta^0 \qquad \forall k \ge 0. \tag{40}$$

*Proof.* We will proceed in several steps.

**Showing that $\Delta^k \ge 0$ for all $k \ge 0$.** First, let us show that $\Delta^k \ge 0$ for every $k$. This is clear for $k = 0$ from (39). Let us show that $\Delta^{k+1} \ge 0$ for $k \ge 0$. Using Young's inequality and (2), we obtain the inequality

$$
\begin{aligned}
\Delta^{k+1} \quad \overset{(38)}{=} \quad & (1 + 2\mu_x\eta_x)\frac{1}{\eta_x}\|x^{k+1} - x^\star\|^2 + (1 + 2\mu_y\eta_y)\frac{1}{\eta_y}\|y^{k+1} - y^\star\|^2 \\
& + \frac{1}{\eta_y}\|y^{k+1} - y^k\|^2 - 2\langle K^\top y^{k+1} - K^\top y^k, x^{k+1} - x^\star\rangle \\
\ge \quad & (1 + 2\mu_x\eta_x)\frac{1}{\eta_x}\|x^{k+1} - x^\star\|^2 + (1 + 2\mu_y\eta_y)\frac{1}{\eta_y}\|y^{k+1} - y^\star\|^2 \\
& + \frac{1}{\eta_y}\|y^{k+1} - y^k\|^2 - L_{xy}B\|y^{k+1} - y^k\|^2 - \frac{L_{xy}}{B}\|x^{k+1} - x^\star\|^2 \\
= \quad & \left(1 + 2\mu_x\eta_x - \frac{L_{xy}\eta_x}{B}\right)\frac{1}{\eta_x}\|x^{k+1} - x^\star\|^2 + (1 + 2\mu_y\eta_y)\frac{1}{\eta_y}\|y^{k+1} - y^\star\|^2 \\
& + (1 - L_{xy}B\eta_y)\frac{1}{\eta_y}\|y^{k+1} - y^k\|^2, \tag{41}
\end{aligned}
$$

which holds for all $B > 0$. Selecting $B = \frac{1}{L_{xy}\eta_y}$, the blue term is zeroed out, and using the primal and dual stepsizes (37), we get

$$
\begin{aligned}
\Delta^{k+1} \quad \overset{(41)}{\ge} \quad & \left(1 + 2\mu_x\eta_x - L_{xy}^2\eta_x\eta_y\right)\frac{1}{\eta_x}\|x^{k+1} - x^\star\|^2 + (1 + 2\mu_y\eta_y)\frac{1}{\eta_y}\|y^{k+1} - y^\star\|^2 \\
\overset{(37)}{=} \quad & \frac{2\sqrt{\mu_x\mu_y}}{L_{xy}}\|x^{k+1} - x^\star\|^2 + (1 + 2\mu_y\eta_y)\frac{1}{\eta_y}\|y^{k+1} - y^\star\|^2 \\
\ge \quad & 0.
\end{aligned}
$$

**Establishing technical bounds.** Denote

$$\mathcal{W}^{k+1} \overset{def}{=} (1 + 2\mu_x\eta_x)\frac{1}{\eta_x}\|x^{k+1} - x^\star\|^2 + (1 + 2\mu_y\eta_y)\frac{1}{\eta_y}\|y^{k+1} - y^\star\|^2. \tag{42}$$

Combining Lemma 4, which provides a bound on $\|x^{k+1} - x^\star\|^2$, and Lemma 3, which provides a bound on $\|y^{k+1} - y^\star\|^2$, we obtain

$$
\begin{aligned}
\mathcal{W}^{k+1} \quad \le \quad & \frac{1}{\eta_x}\|x^k - x^\star\|^2 - \frac{1}{\eta_x}\|x^{k+1} - x^k\|^2 - 2\langle K^\top\bar{y}^k - K^\top y^\star, x^{k+1} - x^\star\rangle \\
& + \frac{1}{\eta_y}\|y^k - y^\star\|^2 - \frac{1}{\eta_y}\|y^{k+1} - y^k\|^2 + 2\langle K^\top y^{k+1} - K^\top y^\star, x^{k+1} - x^\star\rangle \\
\le \quad & \frac{1}{\eta_x}\|x^k - x^\star\|^2 - \frac{1}{\eta_x}\|x^{k+1} - x^k\|^2 + \frac{1}{\eta_y}\|y^k - y^\star\|^2 - \frac{1}{\eta_y}\|y^{k+1} - y^k\|^2 \\
& + 2\langle K^\top y^{k+1} - K^\top\bar{y}^k, x^{k+1} - x^\star\rangle. \tag{43}
\end{aligned}
$$

This is the same inequality as (31) with the exception that $y^k$ was replaced by $\bar{y}^k$. Using Lemma 5 to bound the inner product in (43), we get

$$\mathcal{W}^{k+1} \overset{(43)+(34)}{\leq} \frac{1}{\eta_x}\|x^k - x^\star\|^2 - \frac{1}{\eta_x}\|x^{k+1} - x^k\|^2 + \frac{1}{\eta_y}\|y^k - y^\star\|^2 - \frac{1}{\eta_y}\|y^{k+1} - y^k\|^2$$
$$+2\langle K^\top y^{k+1} - K^\top y^k, x^{k+1} - x^\star\rangle - 2\theta\langle K^\top y^k - K^\top y^{k-1}, x^k - x^\star\rangle$$
$$+\theta L_{xy}C\|x^{k+1} - x^k\|^2 + \frac{\theta L_{xy}}{C}\|y^k - y^{k-1}\|^2$$

$$\leq \frac{1}{\eta_x}\|x^k - x^\star\|^2 - \left(\frac{1}{\eta_x} - \theta L_{xy}C\right)\|x^{k+1} - x^k\|^2 + \frac{1}{\eta_y}\|y^k - y^\star\|^2 - \frac{1}{\eta_y}\|y^{k+1} - y^k\|^2$$
$$+2\langle K^\top y^{k+1} - K^\top y^k, x^{k+1} - x^\star\rangle - 2\theta\langle K^\top y^k - K^\top y^{k-1}, x^k - x^\star\rangle$$
$$+\frac{\theta L_{xy}}{C}\|y^k - y^{k-1}\|^2$$

$$= \frac{1}{\eta_x}\|x^k - x^\star\|^2 + \frac{1}{\eta_y}\|y^k - y^\star\|^2 - \frac{1}{\eta_y}\|y^{k+1} - y^k\|^2$$
$$+2\langle K^\top y^{k+1} - K^\top y^k, x^{k+1} - x^\star\rangle - 2\theta\langle K^\top y^k - K^\top y^{k-1}, x^k - x^\star\rangle$$
$$+\theta^2 L_{xy}^2 \eta_x \eta_y \frac{1}{\eta_y}\|y^k - y^{k-1}\|^2, \tag{44}$$

where in the last step we have made the choice $C \overset{\text{def}}{=} (\theta\eta_x L_{xy})^{-1}$.

**Showing that $\Delta^{k+1} \leq \theta\Delta^k$ for $k \geq 1$.** By combining (38) and (42), and applying inequality (44), for $k \geq 1$ we get

$$\Delta^{k+1} \overset{(38)+(42)}{=} \mathcal{W}^{k+1} + \frac{1}{\eta_y}\|y^{k+1} - y^k\|^2 - 2\langle K^\top y^{k+1} - K^\top y^k, x^{k+1} - x^\star\rangle$$

$$\overset{(44)}{\leq} \frac{1}{\eta_x}\|x^k - x^\star\|^2 + \frac{1}{\eta_y}\|y^k - y^\star\|^2 - \frac{1}{\eta_y}\|y^{k+1} - y^k\|^2$$
$$\color{red}{+2\langle K^\top y^{k+1} - K^\top y^k, x^{k+1} - x^\star\rangle} - 2\theta\langle K^\top y^k - K^\top y^{k-1}, x^k - x^\star\rangle$$
$$+\theta^2 L_{xy}^2 \eta_x \eta_y \frac{1}{\eta_y}\|y^k - y^{k-1}\|^2$$
$$\color{blue}{+\frac{1}{\eta_y}\|y^{k+1} - y^k\|^2} \color{red}{-2\langle K^\top y^{k+1} - K^\top y^k, x^{k+1} - x^\star\rangle}$$

$$\overset{(A)}{\leq} \frac{1}{\eta_x}\|x^k - x^\star\|^2 + \frac{1}{\eta_y}\|y^k - y^\star\|^2 - 2\theta\langle K^\top y^k - K^\top y^{k-1}, x^k - x^\star\rangle + \theta^2 L_{xy}^2 \eta_x \eta_y \frac{1}{\eta_y}\|y^k - y^{k-1}\|^2$$

$$\overset{(B)}{\leq} \frac{1}{\eta_x}\|x^k - x^\star\|^2 + \frac{1}{\eta_y}\|y^k - y^\star\|^2 - 2\theta\langle K^\top y^k - K^\top y^{k-1}, x^k - x^\star\rangle + \theta^2 \frac{1}{\eta_y}\|y^k - y^{k-1}\|^2$$

$$\overset{(C)}{\leq} \theta\left((1 + 2\mu_x\eta_x)\frac{1}{\eta_x}\|x^k - x^\star\|^2 + (1 + 2\mu_y\eta_y)\frac{1}{\eta_y}\|y^k - y^\star\|^2\right)$$
$$+\theta\left(\frac{1}{\eta_y}\|y^k - y^{k-1}\|^2 - 2\langle K^\top y^k - K^\top y^{k-1}, x^k - x^\star\rangle\right)$$

$$\overset{(38)}{=} \theta\Delta^k, \tag{45}$$

where in step (A) we annihilated the red and blue terms as their sum is zero, in step (B) we used the fact that $L_{xy}^2 \eta_x \eta_y \leq 1$, which follows from the stepsize choice, and in (C) we used the inequalities $1 \leq \theta(1 + 2\mu_x\eta_x)$, $1 \leq \theta(1 + 2\mu_y\eta_y)$, and $\theta \leq 1$, which follow from (36).

**Showing that $\Delta^{k+1} \leq \theta\Delta^k$ for $k = 0$.** We start with inequality (43) for $k = 0$:

$$\mathcal{W}^1 \overset{(43)}{\leq} \frac{1}{\eta_x}\|x^0 - x^\star\|^2 - \frac{1}{\eta_x}\|x^1 - x^0\|^2 + \frac{1}{\eta_y}\|y^0 - y^\star\|^2 - \frac{1}{\eta_y}\|y^1 - y^0\|^2$$
$$+2\langle K^\top y^1 - K^\top y^0, x^1 - x^\star\rangle. \tag{46}$$

According to (38) and (36), we have

$$\Delta^1 \overset{(38)+(42)}{=} \mathcal{W}^1 + \frac{1}{\eta_y}\|y^1 - y^0\|^2 - 2\langle K^\top y^1 - K^\top y^0, x^1 - x^\star\rangle$$

$$\overset{(46)}{\leq} \frac{1}{\eta_x}\|x^0 - x^\star\|^2 - \frac{1}{\eta_x}\|x^1 - x^0\|^2 + \frac{1}{\eta_y}\|y^0 - y^\star\|^2$$

$$\leq \frac{1}{\eta_x}\|x^0 - x^\star\|^2 + \frac{1}{\eta_y}\|y^0 - y^\star\|^2$$

$$\leq \theta\left((1 + 2\mu_x\eta_x)\frac{1}{\eta_x}\|x^0 - x^\star\|^2 + (1 + 2\mu_y\eta_y)\frac{1}{\eta_y}\|y^0 - y^\star\|^2\right)$$

$$= \theta\Delta^0$$

where in the last inequality we used the inequalities $1 \leq \theta(1 + 2\mu_x\eta_x)$ and $1 \leq \theta(1 + 2\mu_y\eta_y)$, which follow from (36).

$\square$

The informal result (13) mentioned in Section C.5 is a simple corollary of the above theorem. Indeed, using the definition of $\theta$, we can obtain:

$$k \geq \frac{1}{1-\theta}\log\frac{1}{\varepsilon} \qquad \Rightarrow \qquad \Delta^k \leq \varepsilon\Delta^0.$$

It remains to remark that

$$\frac{1}{1-\theta}\log\frac{1}{\varepsilon} = \mathcal{O}\left(\left(1 + \max\left\{\frac{1}{2\mu_x\eta_x}, \frac{1}{2\mu_y\eta_y}\right\}\right)\log\frac{1}{\varepsilon}\right)$$

$$= \mathcal{O}\left(\left(1 + \frac{L_{xy}}{\sqrt{\mu_x\mu_y}}\right)\log\frac{1}{\varepsilon}\right),$$

which is the result from (13).

# D Analysis of the Accelerated Primal-Dual Algorithm (APDA; Algorithm 1)

In this section we perform convergence analysis for our new method APDA (Algorithm 1). We start by establishing three lemmas, followed by the proof of the main theorem.

## D.1 Three lemmas

The first result is a variant of Lemma 4, offering a bound on the distance of the primal iterates from the primal optimal solution. Compared to Lemma 4, this result is strengthened by the additional assumption of $L_x$-smoothness of $G$ (Assumption 2).

**Lemma 6.** *Let Assumptions 1 and 2 hold and choose any $\eta_x, \eta_y > 0$. Then the iterates of APDA (Algorithm 1) for all $k \geq 0$ satisfy*

$$(1 + \mu_x \eta_x) \frac{1}{\eta_x} \|x^{k+1} - x^\star\|^2 \;\leq\; \frac{1}{\eta_x} \|x^k - x^\star\|^2 - \frac{1}{\eta_x} \|x^{k+1} - x^k\|^2 - 2\langle K^\top \bar{y}^k - K^\top y^\star, x^{k+1} - x^\star \rangle$$
$$- \frac{1}{L_x} \|\nabla G(x^{k+1}) - \nabla G(x^\star)\|^2.$$

*Proof.* By writing $x^{k+1}$ as $x^k + (x^{k+1} - x^k)$, and using line 3 of Algorithm 1, which reads $x^{k+1} = x^k - \eta_x \left( \nabla G(x^{k+1}) + K^\top \bar{y}^k \right)$, we get

$$\frac{1}{\eta_x} \|x^{k+1} - x^\star\|^2 \;=\; \frac{1}{\eta_x} \|x^k - x^\star\|^2 + \frac{2}{\eta_x} \langle x^{k+1} - x^k, x^{k+1} - x^\star \rangle - \frac{1}{\eta_x} \|x^{k+1} - x^k\|^2$$
$$=\; \frac{1}{\eta_x} \|x^k - x^\star\|^2 - 2\langle \nabla G(x^{k+1}) + K^\top \bar{y}^k, x^{k+1} - x^\star \rangle - \frac{1}{\eta_x} \|x^{k+1} - x^k\|^2.$$

We now split the inner product into two parts by applying the optimality condition (see (8))

$$\textcolor{red}{\nabla G(x^\star) + K^\top y^\star = 0,}$$

obtaining

$$\frac{1}{\eta_x} \|x^{k+1} - x^\star\|^2 \;=\; \frac{1}{\eta_x} \|x^k - x^\star\|^2 - 2\langle \nabla G(x^{k+1}) \textcolor{red}{-\nabla G(x^\star)}, x^{k+1} - x^\star \rangle - \frac{1}{\eta_x} \|x^{k+1} - x^k\|^2$$
$$- 2\langle K^\top \bar{y}^k \textcolor{red}{- K^\top y^\star}, x^{k+1} - x^\star \rangle. \tag{47}$$

Since $G$ is $\mu_x$-strongly-convex and $L_x$-smooth, we can lower-bound [Nesterov, 2004] the inner product appearing in (47) as follows:

$$2\langle \nabla G(x^{k+1}) - \nabla G(x^\star), x^{k+1} - x^\star \rangle \geq \mu_x \|x^{k+1} - x^\star\|^2 + \frac{1}{L_x} \|\nabla G(x^{k+1}) - \nabla G(x^\star)\|^2. \tag{48}$$

Finally, by plugging (48) into (47), we get

$$\frac{1}{\eta_x} \|x^{k+1} - x^\star\|^2 \;\leq\; \frac{1}{\eta_x} \|x^k - x^\star\|^2 - \mu_x \|x^{k+1} - x^\star\|^2 - \frac{1}{\eta_x} \|x^{k+1} - x^k\|^2$$
$$- 2\langle K^\top \bar{y}^k - K^\top y^\star, x^{k+1} - x^\star \rangle - \frac{1}{L_x} \|\nabla G(x^{k+1}) - \nabla G(x^\star)\|^2.$$

$\square$

The second result is a technical lemma borrowed from [Kovalev et al., 2021a] for lower-bounding the term $\|K^\top y - K^\top y^\star\|$.

**Lemma 7** (See Kovalev et al. [2021a]). *Let Assumption 4 hold and let $(x^\star, y^\star)$ be a solution of (7). Then*

$$\|K^\top y - K^\top y^\star\| \geq \mu_{xy} \|y - y^\star\|, \qquad \forall y \in \mathbb{R}^{d_y}.$$

The third result offers a bound on the distance between the dual iterates and the dual optimal solution. When analyzing the Proximal Point method and the Chambolle-Pock method, in this step we relied on Lemma 3, in the proof of which we required $F^\star$ to be $\mu_x$-strongly convex. However, for APDA we explicitly wish to avoid using this assumption. As we shall see, in order to obtain linear convergence, it is enough will invoke Assumption 4. The next lemma is an analogue of Lemma 3 without the need to assume $\mu_x$-strong-convexity of $F^\star$.

**Lemma 8.** *Let Assumptions 2, 3, and 4 be satisfied, and choose*

$$\beta_y \leq \frac{1}{2L_{xy}^2 \eta_y}. \tag{49}$$

*Then the iterates of APDA (Algorithm 1) for all $k \geq 0$ satisfy*

$$\frac{1}{\eta_y}\|y^{k+1} - y^\star\|^2 \leq (1 - \mu_{xy}^2 \beta_y \eta_y)\frac{1}{\eta_y}\|y^k - y^\star\|^2 - \frac{1}{2\eta_y}\|y^{k+1} - y^k\|^2 + 2\langle K^\top y^{k+1} - K^\top y^\star, x^{k+1} - x^\star\rangle$$
$$+ \beta_y\|\nabla G(x^{k+1}) - \nabla G(x^\star)\|^2.$$

*Proof.* By writing $y^{k+1}$ as $y^k + (y^{k+1} - y^k)$, and using line 4 from Algorithm 1, which reads

$$y^{k+1} = y^k - \eta_y\left(\partial F^\star(y^{k+1}) - Kx^{k+1}\right) - \eta_y\beta_y K\left(K^\top y^k + \nabla G(x^{k+1})\right),$$

we get

$$
\begin{aligned}
\frac{1}{\eta_y}\|y^{k+1} - y^\star\|^2 &= \frac{1}{\eta_y}\|y^k - y^\star\|^2 + \frac{2}{\eta_y}\langle y^{k+1} - y^k, y^{k+1} - y^\star\rangle - \frac{1}{\eta_y}\|y^{k+1} - y^k\|^2 \\
&= \underbrace{\frac{1}{\eta_y}\|y^k - y^\star\|^2 - 2\langle\partial F^\star(y^{k+1}) - Kx^{k+1}, y^{k+1} - y^\star\rangle - \frac{1}{\eta_y}\|y^{k+1} - y^k\|^2}_{A^k} \\
&\quad \underbrace{- 2\beta_y\langle K^\top y^k + \nabla G(x^{k+1}), K^\top y^{k+1} - K^\top y^\star\rangle}_{B^k}.
\end{aligned}
\tag{50}
$$

We split the inner product appearing in $A^k$ into two parts by applying the optimality condition (see (8))

$$\color{red}{0 \in \partial F^\star(y^\star) - Kx^\star,}$$

obtaining

$$
\begin{aligned}
A^k &= \frac{1}{\eta_y}\|y^k - y^\star\|^2 - 2\langle\partial F^\star(y^{k+1}) \color{red}{- \partial F^\star(y^\star)}, y^{k+1} - y^\star\rangle - \frac{1}{\eta_y}\|y^{k+1} - y^k\|^2 \\
&\quad + 2\langle Kx^{k+1} \color{red}{- Kx^\star}, y^{k+1} - y^\star\rangle.
\end{aligned}
\tag{51}
$$

Applying the parallelogram identity[14] to $B^k$, and using the optimality condition (see (8)),

$$\color{blue}{\nabla G(x^\star) + K^\top y^\star = 0,} \tag{52}$$

we can write

$$
\begin{aligned}
B^k &= \beta_y\|\nabla G(x^{k+1}) + \color{blue}{K^\top y^\star}\|^2 - \beta_y\|K^\top y^k - K^\top y^\star\|^2 \\
&\quad + \beta_y\|K^\top y^k - K^\top y^{k+1}\|^2 - \beta_y\|\nabla G(x^{k+1}) + K^\top y^{k+1}\|^2 \\
&\stackrel{(52)}{=} \beta_y\|\nabla G(x^{k+1}) \color{blue}{- \nabla G(x^\star)}\|^2 - \beta_y\|K^\top y^k - K^\top y^\star\|^2 \\
&\quad + \beta_y\|K^\top y^k - K^\top y^{k+1}\|^2 - \beta_y\|\nabla G(x^{k+1}) + K^\top y^{k+1}\|^2.
\end{aligned}
\tag{53}
$$

---

[14]Here we refer to the identity $2\langle a + b, c - d\rangle = -\|b + d\|^2 + \|a - d\|^2 - \|a - c\|^2 + \|b + c\|^2$ which holds for all $a, b, c, d \in \mathbb{R}^{d_x}$.

Plugging (51) and (53) back into (50), and using convexity of $F^\star$, we get

$$\frac{1}{\eta_y}\|y^{k+1}-y^\star\|^2 \overset{(50)}{=} A^k + B^k$$

$$\overset{(51)+(53)}{=} \frac{1}{\eta_y}\|y^k-y^\star\|^2 \underbrace{-2\langle \partial F^\star(y^{k+1})-\partial F^\star(y^\star), y^{k+1}-y^\star\rangle}_{\leq 0} - \frac{1}{\eta_y}\|y^{k+1}-y^k\|^2$$

$$+2\langle Kx^{k+1} - Kx^\star, y^{k+1}-y^\star\rangle$$
$$+\beta_y\|\nabla G(x^{k+1})-\nabla G(x^\star)\|^2 - \beta_y\|K^\top y^k - K^\top y^\star\|^2$$
$$+\beta_y\|K^\top y^k - K^\top y^{k+1}\|^2 \underbrace{-\beta_y\|\nabla G(x^{k+1}) + K^\top y^{k+1}\|^2}_{\leq 0}$$

$$\leq \frac{1}{\eta_y}\|y^k-y^\star\|^2 - \frac{1}{\eta_y}\|y^{k+1}-y^k\|^2$$
$$+2\langle Kx^{k+1} - Kx^\star, y^{k+1}-y^\star\rangle$$
$$+\beta_y\|\nabla G(x^{k+1})-\nabla G(x^\star)\|^2 - \beta_y\|K^\top y^k - K^\top y^\star\|^2$$
$$+\beta_y\|K^\top y^k - K^\top y^{k+1}\|^2. \tag{54}$$

It now only remains to plug the bounds $\|K^\top y^k - K^\top y^\star\|^2 \geq \mu_{xy}^2\|y^k - y^\star\|^2$ (see Lemma 7) and $\|K^\top y^k - K^\top y^{k+1}\|^2 \leq L_{xy}^2\|y^k - y^{k+1}\|^2$ into (54), and apply the restriction (49) on $\beta_y$.

$\square$

## D.2 Main result

We are now ready to state the formal version of Theorem 1.

**Theorem 7** (Covergence of APDA; formal). *Let Assumptions 1, 2, 3, and 4 hold and choose the various parameters of the method as follows:*

$$\beta_y = \min\left\{\frac{1}{L_x}, \frac{1}{2L_{xy}^2\eta_y}\right\}, \tag{55}$$

$$\eta_x = \frac{1}{2\sqrt{L_x\mu_x}}\frac{\mu_{xy}}{L_{xy}}, \tag{56}$$

$$\eta_y = \frac{\sqrt{L_x\mu_x}}{L_{xy}\mu_{xy}}, \tag{57}$$

$$\theta = \max\left\{\frac{1}{1+\mu_x\eta_x}, 1 - \mu_{xy}^2\beta_y\eta_y\right\}. \tag{58}$$

*Then for the Lyapunov function defined for $k \geq 0$ via*

$$\Xi^{k+1} \overset{def}{=} (1+\mu_x\eta_x)\frac{1}{\eta_x}\|x^{k+1}-x^\star\|^2 + \frac{1}{\eta_y}\|y^{k+1}-y^\star\|^2$$
$$+\frac{1}{2\eta_y}\|y^{k+1}-y^k\|^2 - 2\langle K^\top y^{k+1} - K^\top y^k, x^{k+1}-x^\star\rangle \tag{59}$$

*and for $k = 0$ via*

$$\Xi^0 \overset{def}{=} (1+\mu_x\eta_x)\frac{1}{\eta_x}\|x^0-x^\star\|^2 + \frac{1}{\eta_y}\|y^0-y^\star\|^2,$$

*we have*

$$0 \leq \mu_x\|x^{k+1}-x^\star\|^2 + \frac{1}{\eta_y}\|y^{k+1}-y^\star\|^2 \leq \Xi^k \leq \theta^k\Delta^0 \qquad \forall k \geq 0. \tag{60}$$

*Proof.* We will proceed in several steps.

**Showing that $\Xi^k \geq 0$ for all $k \geq 0$.** First, we will show that $\Xi^k \geq 0$ for every $k$. This is clear for $k = 0$. Let us show that $\Xi^{k+1} \geq 0$ for every $k \geq 0$. Using Young's inequality with any parameter $B > 0$, and the definition of $L_{xy}$ from (2), we obtain

$$
\begin{aligned}
\Xi^{k+1} &\overset{(59)}{=} (1 + \mu_x\eta_x)\frac{1}{\eta_x}\|x^{k+1} - x^\star\|^2 + \frac{1}{\eta_y}\|y^{k+1} - y^\star\|^2 + \frac{1}{2\eta_y}\|y^{k+1} - y^k\|^2 \\
&\quad -2\langle K^\top y^{k+1} - K^\top y^k, x^{k+1} - x^\star\rangle \\
&\geq (1 + \mu_x\eta_x)\frac{1}{\eta_x}\|x^{k+1} - x^\star\|^2 + \frac{1}{\eta_y}\|y^{k+1} - y^\star\|^2 + \frac{1}{2\eta_y}\|y^{k+1} - y^k\|^2 \\
&\quad -L_{xy}B\|y^{k+1} - y^k\|^2 - \frac{L_{xy}}{B}\|x^{k+1} - x^\star\|^2 \\
&= (1 + \mu_x\eta_x)\frac{1}{\eta_x}\|x^{k+1} - x^\star\|^2 + \frac{1}{\eta_y}\|y^{k+1} - y^\star\|^2 + \left(\frac{1}{2} - L_{xy}B\eta_y\right)\frac{1}{\eta_y}\|y^{k+1} - y^k\|^2 \\
&\quad -\frac{L_{xy}}{B}\|x^{k+1} - x^\star\|^2. \tag{61}
\end{aligned}
$$

Using (61) with $B = \frac{1}{2L_{xy}\eta_y}$, and noticing that $2L_{xy}^2\eta_x\eta_y = 1$ (this follows from (56) and (57)), we get

$$
\begin{aligned}
\Xi^{k+1} &\geq (1 + \mu_x\eta_x)\frac{1}{\eta_x}\|x^{k+1} - x^\star\|^2 + \frac{1}{\eta_y}\|y^{k+1} - y^\star\|^2 - \frac{2L_{xy}^2\eta_x\eta_y}{\eta_x}\|x^{k+1} - x^\star\|^2 \\
&= \mu_x\|x^{k+1} - x^\star\|^2 + \frac{1}{\eta_y}\|y^{k+1} - y^\star\|^2 \\
&\geq 0.
\end{aligned}
$$

**Establishing technical bounds.** Denote

$$
\mathcal{W}^{k+1} \overset{\text{def}}{=} (1 + \mu_x\eta_x)\frac{1}{\eta_x}\|x^{k+1} - x^\star\|^2 + \frac{1}{\eta_y}\|y^{k+1} - y^\star\|^2. \tag{62}
$$

By adding the inequalities from Lemmas 6 and 8, we obtain

$$
\begin{aligned}
\mathcal{W}^{k+1} &\leq \frac{1}{\eta_x}\|x^k - x^\star\|^2 - \frac{1}{\eta_x}\|x^{k+1} - x^k\|^2 - \frac{1}{L_x}\|\nabla G(x^{k+1}) - \nabla G(x^\star)\|^2 \\
&\quad +(1 - \mu_{xy}^2\beta_y\eta_y)\frac{1}{\eta_y}\|y^k - y^\star\|^2 - \frac{1}{2\eta_y}\|y^{k+1} - y^k\|^2 + \beta_y\|\nabla G(x^{k+1}) - \nabla G(x^\star)\|^2 \\
&\quad +2\langle K^\top y^{k+1} - K^\top y^\star, x^{k+1} - x^\star\rangle - 2\langle K^\top \bar{y}^k - K^\top y^\star, x^{k+1} - x^\star\rangle \\
&\leq \frac{1}{\eta_x}\|x^k - x^\star\|^2 - \frac{1}{\eta_x}\|x^{k+1} - x^k\|^2 + (1 - \mu_{xy}^2\beta_y\eta_y)\frac{1}{\eta_y}\|y^k - y^\star\|^2 - \frac{1}{2\eta_y}\|y^{k+1} - y^k\|^2 \\
&\quad +2\langle K^\top y^{k+1} - K^\top \bar{y}^k, x^{k+1} - x^\star\rangle, \tag{63}
\end{aligned}
$$

where in the last step we used the bound

$$
\beta_y\|\nabla G(x^{k+1}) - \nabla G(x^\star)\|^2 - \frac{1}{L_x}\|\nabla G(x^{k+1}) - \nabla G(x^\star)\|^2 \leq 0,
$$

which follows from the restriction $\beta_y \leq \frac{1}{L_x}$; see (55). Using line 5 from Algorithm 1, which says

$$
\bar{y}^{k+1} = y^{k+1} + \theta(y^{k+1} - y^k), \tag{64}
$$

applying Cauchy-Schwarz inequality, and subsequently using Young's inequality with constant $C > 0$, the inner product from (63) can for $k \geq 1$ be further bounded as follows

$$
\begin{aligned}
2\langle K^\top y^{k+1} - K^\top \bar{y}^k, x^{k+1} - x^\star\rangle \overset{(64)}{=} \quad & 2\langle K^\top y^{k+1} - K^\top y^k, x^{k+1} - x^\star\rangle \\
& -2\theta\langle K^\top y^k - K^\top y^{k-1}, x^k - x^\star\rangle \\
& +2\theta\langle K^\top y^k - K^\top y^{k-1}, x^{k+1} - x^k\rangle \\
\leq \quad & 2\langle K^\top y^{k+1} - K^\top y^k, x^{k+1} - x^\star\rangle \\
& -2\theta\langle K^\top y^k - K^\top y^{k-1}, x^k - x^\star\rangle \\
& +2\theta L_{xy}\|y^k - y^{k-1}\|\|x^{k+1} - x^k\| \\
\leq \quad & +2\langle K^\top y^{k+1} - K^\top y^k, x^{k+1} - x^\star\rangle \\
& -2\theta\langle K^\top y^k - K^\top y^{k-1}, x^k - x^\star\rangle \\
& +\theta L_{xy}C\|y^k - y^{k-1}\|^2 + \frac{\theta L_{xy}}{C}\|x^{k+1} - x^k\|^2. \quad (65)
\end{aligned}
$$

**Showing that $\Xi^{k+1} \leq \theta\Xi^k$ for all $k \geq 1$.** Plugging the bounds (62) and (65) into the Lyapunov function (59), for any $k \geq 1$ we get

$$
\begin{aligned}
\Xi^{k+1} \overset{(59)+(62)+(65)}{\leq} \quad & \frac{1}{\eta_x}\|x^k - x^\star\|^2 - \left(\frac{1}{\eta_x} - \frac{\theta L_{xy}}{C}\right)\|x^{k+1} - x^k\|^2 + (1 - \mu_{xy}^2\beta_y\eta_y)\frac{1}{\eta_y}\|y^k - y^\star\|^2 \\
& -2\theta\langle K^\top y^k - K^\top y^{k-1}, x^k - x^\star\rangle + \theta L_{xy}C\|y^k - y^{k-1}\|^2 \\
\overset{C=\theta\eta_x L_{xy}}{\leq} \quad & \frac{1}{\eta_x}\|x^k - x^\star\|^2 + (1 - \mu_{xy}^2\beta_y\eta_y)\frac{1}{\eta_y}\|y^k - y^\star\|^2 \\
& -2\theta\langle K^\top y^k - K^\top y^{k-1}, x^k - x^\star\rangle + \theta^2 L_{xy}^2\eta_x\eta_y\frac{1}{\eta_y}\|y^k - y^{k-1}\|^2 \\
\overset{(56)+(57) \;\&\; \theta\leq 1}{\leq} \quad & \frac{1}{\eta_x}\|x^k - x^\star\|^2 + (1 - \mu_{xy}^2\beta_y\eta_y)\frac{1}{\eta_y}\|y^k - y^\star\|^2 \\
& -2\theta\langle K^\top y^k - K^\top y^{k-1}, x^k - x^\star\rangle + \theta\frac{1}{2\eta_y}\|y^k - y^{k-1}\|^2 \\
\leq \quad & \max\left\{\frac{1}{1 + \mu_x\eta_x}, 1 - \mu_{xy}^2\beta_y\eta_y\right\}\left((1 + \mu_x\eta_x)\frac{1}{\eta_x}\|x^k - x^\star\|^2 + \frac{1}{\eta_y}\|y^k - y^\star\|^2\right) \\
& +\theta\left(\frac{1}{2\eta_y}\|y^k - y^{k-1}\|^2 - 2\langle K^\top y^k - K^\top y^{k-1}, x^k - x^\star\rangle\right) \\
\overset{(58)+(59)}{=} \quad & \theta\Xi^k.
\end{aligned}
$$

**Showing that $\Xi^{k+1} \leq \theta\Xi^k$ for $k = 0$.** This can be done using similar arguments as those used in the proof of Theorem 6.

$\square$

### D.3 Proof of Theorem 1 (informal)

We now provide the iteration complexity of Algorithm 1 as a corollary of Theorem 7. Note that in view of (60), we get

$$
k \geq \mathcal{O}\left(\frac{1}{1 - \theta}\log\frac{1}{\varepsilon}\right) \qquad \Rightarrow \qquad \Xi^k \leq \varepsilon\Xi^0.
$$

Using the definition of $\theta$ given in (58), we have

$$
\begin{aligned}
\mathcal{O}\left(\frac{1}{1-\theta}\log\frac{1}{\varepsilon}\right) &= \mathcal{O}\left(\max\left\{1+\frac{1}{\mu_x\eta_x},\frac{1}{\mu_{xy}^2\beta_y\eta_y}\right\}\log\frac{1}{\varepsilon}\right) \\
&= \mathcal{O}\left(\max\left\{1+\frac{1}{\mu_x\eta_x},\frac{L_x}{\mu_{xy}^2\eta_y},\frac{L_{xy}^2}{\mu_{xy}^2}\right\}\log\frac{1}{\varepsilon}\right) \\
&= \mathcal{O}\left(\max\left\{1+\sqrt{\frac{L_x}{\mu_x}}\frac{L_{xy}}{\mu_{xy}},\frac{L_{xy}^2}{\mu_{xy}^2}\right\}\log\frac{1}{\varepsilon}\right).
\end{aligned}
$$

This proves the statement of Theorem 1 mentioned in Section 4 in the main body of the paper.

# E   Analysis of the Accelerated Primal-Dual Algorithm with Inexact Prox (Algorithm 2)

In this section we provide convergence analysis for our second new method, APDA with Inexact Prox (Algorithm 2). We start with statements of three lemmas, followed by the proof of the main theorem.

## E.1   Three Lemmas

The first result is essentially a modification of Lemma 6, providing a bound on the distance of the primal iterates from the primal optimal solution. Compare to Lemma 6, these changes consist in using the definition of function $\Psi^k$ (see the problem (15)) to prove this key fact.

**Lemma 9.** *Let $w^{\star k}$ be an exact solution to the problem* (15). *Then under Assumptions 1, 2, we have*

$$\left(1 + \frac{\mu_x \eta_x}{2}\right) \frac{1}{\eta_x} \|x^{k+1} - x^\star\|^2 \leq \frac{1}{\eta_x} \|x^k - x^\star\|^2 + (2\eta_x + \mu_x \eta_x^2) \|\nabla \Psi^k(\hat{x}^k)\|^2 - \frac{1}{2\eta_x} \|x^k - w^{\star k}\|^2$$
$$- 2\langle K^\top \bar{y}^k - K^\top y^\star, \hat{x}^k - x^\star \rangle - \frac{1}{L_x} \|\nabla G(\hat{x}^k) - \nabla G(x^\star)\|^2.$$

*Proof.* In view of line 4 of Algorithm 2, which reads

$$x^{k+1} = x^k - \eta_x \left(\nabla G(\hat{x}^k) + K^\top \bar{y}^k\right), \tag{66}$$

and writing $x^{k+1}$ as $x^k + (x^{k+1} - x^k)$, we get

$$\frac{1}{\eta_x} \|x^{k+1} - x^\star\|^2 = \frac{1}{\eta_x} \|x^k - x^\star\|^2 + \frac{2}{\eta_x} \langle x^{k+1} - x^k, x^{k+1} - x^\star \rangle - \frac{1}{\eta_x} \|x^{k+1} - x^k\|^2$$

$$\overset{(66)}{=} \frac{1}{\eta_x} \|x^k - x^\star\|^2 - \frac{2}{\eta_x} \langle \eta_x \left(\nabla G(\hat{x}^k) + K^\top \bar{y}^k\right), x^{k+1} - x^\star \rangle - \frac{1}{\eta_x} \left\|\eta_x \left(\nabla G(\hat{x}^k) + K^\top \bar{y}^k\right)\right\|^2$$

$$= \frac{1}{\eta_x} \|x^k - x^\star\|^2 - 2\langle \nabla G(\hat{x}^k) + K^\top \bar{y}^k, \hat{x}^k - x^\star \rangle - \eta_x \left\|\nabla G(\hat{x}^k) + K^\top \bar{y}^k\right\|^2$$

$$- \frac{2}{\eta_x} \left\langle \eta_x \left(\nabla G(\hat{x}^k) + K^\top \bar{y}^k\right), x^{k+1} - \hat{x}^k \right\rangle.$$

Using the identity $-2\langle a, b \rangle = \|a\|^2 + \|b\|^2 - \|a + b\|^2$ to rewrite the second inner product from the previous equation, we get

$$\frac{1}{\eta_x} \|x^{k+1} - x^\star\|^2 = \frac{1}{\eta_x} \|x^k - x^\star\|^2 - 2\langle \nabla G(\hat{x}^k) + K^\top \bar{y}^k, \hat{x}^k - x^\star \rangle - \eta_x \|\nabla G(\hat{x}^k) + K^\top \bar{y}^k\|^2$$

$$+ \frac{1}{\eta_x} \left(\left\|\eta_x \left(\nabla G(\hat{x}^k) + K^\top \bar{y}^k\right)\right\|^2 + \|x^{k+1} - \hat{x}^k\|^2\right)$$

$$- \frac{1}{\eta_x} \left\|\eta_x \left(\nabla G(\hat{x}^k) + K^\top \bar{y}^k\right) + x^{k+1} - \hat{x}^k\right\|^2$$

$$\overset{(66)}{=} \frac{1}{\eta_x} \|x^k - x^\star\|^2 - 2\langle \nabla G(\hat{x}^k) - \nabla G(x^\star), \hat{x}^k - x^\star \rangle$$

$$- 2\langle K^\top \bar{y}^k - K^\top y^\star, \hat{x}^k - x^\star \rangle + \frac{1}{\eta_x} \|x^{k+1} - \hat{x}^k\|^2 - \frac{1}{\eta_x} \|x^k - \hat{x}^k\|^2, \tag{67}$$

where in the last equation we applied the optimality condition (see (8))

$$\nabla G(x^\star) + K^\top y^\star = 0.$$

Due to the fact that $G$ is $\mu_x$-strongly-convex and $L_x$-smooth, we can estimate the inner product $2\langle \nabla G(\hat{x}^k) - \nabla G(x^\star), \hat{x}^k - x^\star \rangle$ from below similarly as in the proof of Lemma 6:

$$2\langle \nabla G(\hat{x}^k) - \nabla G(x^\star), \hat{x}^k - x^\star \rangle \geq \mu_x \|\hat{x}^k - x^\star\|^2 + \frac{1}{L_x} \|\nabla G(\hat{x}^k) - \nabla G(x^\star)\|^2. \tag{68}$$

Now, by plugging (68) into (67), we obtain

$$
\frac{1}{\eta_x}\|x^{k+1}-x^\star\|^2 \leq \frac{1}{\eta_x}\|x^k-x^\star\|^2 - \mu_x\|\hat{x}^k-x^\star\|^2 - \frac{1}{L_x}\|\nabla G(\hat{x}^k)+\nabla G(x^\star)\|^2
$$
$$
+\frac{1}{\eta_x}\|x^{k+1}-\hat{x}^k\|^2 - \frac{1}{\eta_x}\|x^k-\hat{x}^k\|^2 - 2\langle K^\top \bar{y}^k - K^\top y^\star, \hat{x}^k - x^\star\rangle.
$$

Applying the inequality $\|a-b\|^2 \geq \frac{1}{2}\|a-c\|^2 - \|b-c\|^2$ to $\|\hat{x}^k - x^\star\|^2$ and taking $c = x^{k+1}$, we obtain

$$
\frac{1}{\eta_x}\|x^{k+1}-x^\star\|^2 \leq \frac{1}{\eta_x}\|x^k-x^\star\|^2 - \frac{\mu_x}{2}\|x^{k+1}-x^\star\|^2 - \frac{1}{L_x}\|\nabla G(\hat{x}^k)-\nabla G(x^\star)\|^2
$$
$$
+(1+\mu_x\eta_x)\frac{1}{\eta_x}\|x^{k+1}-\hat{x}^k\|^2 - \frac{1}{\eta_x}\|x^k-\hat{x}^k\|^2
$$
$$
-2\langle K^\top \bar{y}^k - K^\top y^\star, \hat{x}^k - x^\star\rangle
$$
$$
\leq \frac{1}{\eta_x}\|x^k-x^\star\|^2 - \frac{\mu_x}{2}\|x^{k+1}-x^\star\|^2 - \frac{1}{L_x}\|\nabla G(\hat{x}^k)-\nabla G(x^\star)\|^2
$$
$$
+(1+\mu_x\eta_x)\frac{1}{\eta_x}\|x^{k+1}-\hat{x}^k\|^2 - \frac{1}{2\eta_x}\|x^k-w^{\star k}\|^2 + \frac{1}{\eta_x}\|\hat{x}^k-w^{\star k}\|^2
$$
$$
-2\langle K^\top \bar{y}^k - K^\top y^\star, \hat{x}^k - x^\star\rangle, \tag{69}
$$

where in the last inequality we used the bound $\|a-b\|^2 \geq \frac{1}{2}\|a-c\|^2 - \|b-c\|^2$ to estimate $\frac{1}{\eta_x}\|x^k-\hat{x}^k\|^2$. From line 3 of Algorithm 2, according to the definition of function $\Psi^k(x)$, we get

$$
\nabla\Psi^k(\hat{x}^k) = \nabla G(\hat{x}^k) + K^\top \bar{y}^k + \frac{1}{\eta_x}\left(\hat{x}^k - x^k\right)
$$
$$
= \nabla G(\hat{x}^k) + K^\top \bar{y}^k + \frac{1}{\eta_x}\left(\hat{x}^k - x^{k+1}\right) + \frac{1}{\eta_x}\left(x^{k+1} - x^k\right)
$$
$$
\overset{(66)}{=} \frac{1}{\eta_x}\left(\hat{x}^k - x^{k+1}\right). \tag{70}
$$

Finally, substituting (70) into (69), we gain

$$
\frac{1}{\eta_x}\|x^{k+1}-x^\star\|^2 = \frac{1}{\eta_x}\|x^k-x^\star\|^2 - \frac{\mu_x}{2}\|x^{k+1}-x^\star\|^2 - \frac{1}{L_x}\|\nabla G(\hat{x}^k)-\nabla G(x^\star)\|^2
$$
$$
+\left(2\eta_x+\mu_x\eta_x^2\right)\|\nabla\Psi^k(\hat{x}^k)\|^2 - \frac{1}{2\eta_x}\|x^k-w^{\star k}\|^2
$$
$$
-2\langle K^\top \bar{y}^k - K^\top y^\star, \hat{x}^k - x^\star\rangle,
$$

where we also use $\frac{1}{\eta_x}$-strong convexity of function $\Psi^k(x)$ (see problem (15)). $\qquad\square$

The second result offers a bound on the distance between the dual itarates and the dual optimal solution as Lemma 8. Since the line 4 of Algorithm 1 coincides with the line 5 of Algorithm 2, the statement and proof of the following lemma coincides with Lemma 8 and its proof with the only change of replacing $x^{k+1}$ with $\hat{x}^k$.

**Lemma 10.** *Then under Assumptions 2, 3, and 4 be satisfied, and choose*

$$
\beta_y \leq \frac{1}{2L_{xy}^2\eta_y}. \tag{71}
$$

*Then the iterates of APDA with Inexact Prox (Algorithm 2) for all $k \geq 0$ satisfy*

$$
\frac{1}{\eta_y}\|y^{k+1}-y^\star\|^2 \leq (1-\mu_{xy}^2\beta_y\eta_y)\frac{1}{\eta_y}\|y^k-y^\star\|^2 - \frac{1}{2\eta_y}\|y^{k+1}-y^k\|^2 + 2\langle K^\top y^{k+1} - K^\top y^\star, \hat{x}^k - x^\star\rangle
$$
$$
+ \beta_y\|\nabla G(\hat{x}^k)-\nabla G(x^\star)\|^2.
$$

*Proof.* The proof is identical to the proof of Lemma 8, with the sole distinction that $x^{k+1}$ should be replaced everywhere by $\hat{x}^k$. $\qquad\square$

The third result is a technical lemma, which offers a bound on an inner product .

**Lemma 11.** *Let $w^{\star k}$ be an exact solution to the problem* (15). *Then the following inequality holds*

$$-2\theta\langle K^\top y^k - K^\top y^{k-1}, \hat{x}^k - \hat{x}^{k-1}\rangle \leq 16L_{xy}^2\theta\eta_x\|y^k - y^{k-1}\|^2 + \frac{\theta}{4\eta_x}\|\hat{x}^k - w^{\star k}\|^2$$
$$+ \frac{\theta}{4\eta_x}\|x^k - w^{\star k}\|^2 + \frac{\theta\eta_x}{8}\|\nabla\Psi^{k-1}(\hat{x}^{k-1})\|^2.$$

*Proof.* Using the Cauchy-Schwarz inequality, the definition of $L_{xy}$ as the norm of $K$ (see (2)), and applying Young's inequality, we can bound the inner product by

$$-2\theta\langle K^\top y^k - K^\top y^{k-1}, \hat{x}^k - \hat{x}^{k-1}\rangle \leq 2\theta\|K^\top y^k - K^\top y^{k-1}\|\|\hat{x}^k - \hat{x}^{k-1}\|$$
$$\leq L_{xy}\theta C\|y^k - y^{k-1}\|^2 + \frac{L_{xy}\theta}{C}\|\hat{x}^k - \hat{x}^{k-1}\|^2$$
$$\leq L_{xy}\theta C\|y^k - y^{k-1}\|^2 + \frac{2L_{xy}\theta}{C}\|\hat{x}^k - x^k\|^2$$
$$+ \frac{2L_{xy}\theta}{C}\|x^k - \hat{x}^{k-1}\|^2$$

for any $C > 0$. Taking $C = 16L_{xy}\eta_x$, we obtain

$$-2\theta\langle K^\top y^k - K^\top y^{k-1}, \hat{x}^k - \hat{x}^{k-1}\rangle \leq L_{xy}\theta C\|y^k - y^{k-1}\|^2 + \frac{4L_{xy}\theta}{C}\|\hat{x}^k - w^{\star k}\|^2$$
$$\frac{4L_{xy}\theta}{C}\|\hat{x}^k - w^{\star k}\|^2 + \frac{2L_{xy}\theta}{C}\|x^k - \hat{x}^{k-1}\|^2$$
$$= 16L_{xy}^2\theta\eta_x\|y^k - y^{k-1}\|^2 + \frac{\theta}{4\eta_x}\|\hat{x}^k - w^{\star k}\|^2$$
$$+ \frac{\theta}{4\eta_x}\|x^k - w^{\star k}\|^2 + \frac{\theta\eta_x}{8}\|\nabla\Psi^{k-1}(\hat{x}^{k-1})\|^2,$$

where in the last equation we use (70) for $k-1$, which reads $\nabla\Psi^k(\hat{x}^{k-1}) = \frac{1}{\eta_x}(\hat{x}^{k-1} - x^k)$. $\qquad\square$

## E.2   Detailed theorem

**Theorem 8** (Convergence of APDA with Inexact Prox; formal)**.** *Let Assumptions 1, 2, 3 and 4 hold and select the various parameters of the method as follows:*

$$\beta_y = \min\left\{\frac{1}{L_x}, \frac{1}{2L_{xy}^2\eta_y}\right\}; \tag{72}$$

$$T = \sqrt[\alpha]{20A}\left(1 + \sqrt{\frac{L_x}{\mu_x}}\right)^{2/\alpha}; \tag{73}$$

$$\eta_x = \frac{1}{4\sqrt{L_x\mu_x}}\frac{\mu_{xy}}{L_{xy}}; \tag{74}$$

$$\eta_y = \frac{\sqrt{L_x\mu_x}}{8L_{xy}\mu_{xy}}; \tag{75}$$

$$\theta = \max\left\{\frac{2}{2 + \mu_x\eta_x}, 1 - \mu_{xy}^2\beta_y\eta_y\right\}, \tag{76}$$

*Then for the Lyapunov function defined for $k \geq 0$ via*

$$\Delta^{k+1} \stackrel{def}{=} \left(1 + \frac{\mu_x\eta_x}{2}\right)\frac{1}{\eta_x}\|x^{k+1} - x^\star\|^2 + \frac{1}{\eta_y}\|y^{k+1} - y^\star\|^2 + \frac{1}{2\eta_y}\|y^{k+1} - y^k\|^2$$
$$+ \frac{1}{8\eta_x}\|x^k - w^{\star k}\|^2 - 2\langle K^\top y^{k+1} - K^\top y^k, \hat{x}^k - x^\star\rangle, \tag{77}$$

*and for $k = 0$ via*

$$\Delta^0 \stackrel{def}{=} \left(1 + \frac{\mu_x \eta_x}{2}\right) \frac{1}{\eta_x} \|x^0 - x^\star\|^2 + \frac{1}{\eta_y} \|y^0 - y^\star\|^2, \tag{78}$$

*we have*

$$\frac{1}{2\eta_x} \|x^{k+1} - x^\star\|^2 + \frac{1}{\eta_y} \|y^{k+1} - y^\star\|^2 \leq \Delta^k \leq \theta^k \Delta^0, \quad \forall\, k \geq 0. \tag{79}$$

*Proof.* We denote $\mathcal{V}^{k+1}$ as follows:

$$\mathcal{V}^{k+1} = \left(1 + \frac{\mu_x \eta_x}{2}\right) \frac{1}{\eta_x} \|x^{k+1} - x^\star\|^2 + \frac{1}{\eta_y} \|y^{k+1} - y^\star\|^2 + \frac{1}{2\eta_y} \|y^{k+1} - y^k\|^2. \tag{80}$$

Combining Lemmas 9, 10 and (72), we obtain

$$
\begin{aligned}
\mathcal{V}^{k+1} \leq\ & \frac{1}{\eta_x} \|x^k - x^\star\|^2 + (1 - \mu_{xy}^2 \beta_y \eta_y) \frac{1}{\eta_y} \|y^k - y^\star\|^2 + (2\eta_x + \mu_x \eta_x^2) \|\nabla \Psi^k(\hat{x}^k)\|^2 \\
& - \frac{1}{2\eta_x} \|x^k - w^{\star k}\|^2 + \left(\beta_y - \frac{1}{L_x}\right) \|\nabla G(\hat{x}^k) - \nabla G(x^\star)\|^2 \\
& - 2\langle K^\top \bar{y}^k - K^\top y^\star, \hat{x}^k - x^\star \rangle + 2\langle K^\top y^{k+1} - K^\top y^\star, \hat{x}^k - x^\star \rangle \\
\leq\ & \frac{1}{\eta_x} \|x^k - x^\star\|^2 + (1 - \mu_{xy}^2 \beta_y \eta_y) \frac{1}{\eta_y} \|y^k - y^\star\|^2 + (2\eta_x + \mu_x \eta_x^2) \|\nabla \Psi^k(\hat{x}^k)\|^2 \\
& - \frac{1}{2\eta_x} \|x^k - w^{\star k}\|^2 + 2\langle K^\top y^{k+1} - K^\top y^k, \hat{x}^k - x^\star \rangle \\
& - 2\theta\langle K^\top y^k - K^\top y^{k-1}, \hat{x}^{k-1} - x^\star \rangle - 2\theta\langle K^\top y^k - K^\top y^{k-1}, \hat{x}^k - \hat{x}^{k-1} \rangle,
\end{aligned}
\tag{81}
$$

where in last inequality the line 6 from Algorithm 2 is used. Next, according to Lemma 11, we gain

$$
\begin{aligned}
\mathcal{V}^{k+1} \leq\ & \frac{1}{\eta_x} \|x^k - x^\star\|^2 + (1 - \mu_{xy}^2 \beta_y \eta_y) \frac{1}{\eta_y} \|y^k - y^\star\|^2 + (2\eta_x + \mu_x \eta_x^2) \|\nabla \Psi^k(\hat{x}^k)\|^2 - \frac{1}{2\eta_x} \|x^k - w^{\star k}\|^2 \\
& + 2\langle K^\top y^{k+1} - K^\top y^k, \hat{x}^k - x^\star \rangle - 2\theta\langle K^\top y^k - K^\top y^{k-1}, \hat{x}^{k-1} - x^\star \rangle \\
& + 16\theta L_{xy}^2 \eta_x \|y^k - y^{k-1}\|^2 + \frac{\theta}{4\eta_x} \|\hat{x}^k - w^{\star k}\|^2 + \frac{\theta}{4\eta_x} \|x^k - w^{\star k}\|^2 + \frac{\theta \eta_x}{8} \|\nabla \Psi^{k-1}_{\eta_x}(\hat{x}^{k-1})\|^2 \\
\leq\ & \frac{1}{\eta_x} \|x^k - x^\star\|^2 + (1 - \mu_{xy}^2 \beta_y \eta_y) \frac{1}{\eta_y} \|y^k - y^\star\|^2 + 16\theta L_{xy}^2 \eta_x \eta_y \frac{1}{\eta_y} \|y^k - y^{k-1}\|^2 \\
& + (2\eta_x + \mu_x \eta_x^2) \|\nabla \Psi^k(\hat{x}^k)\|^2 - \frac{1}{4\eta_x} \|x^k - w^{\star k}\|^2 + \frac{\theta}{4\eta_x} \|\hat{x}^k - w^{\star k}\|^2 + \frac{\theta \eta_x}{8} \|\nabla \Psi^{k-1}_{\eta_x}(\hat{x}^{k-1})\|^2 \\
& + 2\langle K^\top y^{k+1} - K^\top y^k, \hat{x}^k - x^\star \rangle - 2\theta\langle K^\top y^k - K^\top y^{k-1}, \hat{x}^{k-1} - x^\star \rangle.
\end{aligned}
$$

Using definition of function $\Psi^k$ and its $\frac{1}{\eta_x}$-strong convexity in the following form

$$\|\nabla \Psi^k(\hat{x}^k)\|^2 \geq \frac{1}{\eta_x^2} \|\hat{x}^k - w^{\star k}\|^2,$$

and assuming that $32 L_{xy}^2 \eta_x \eta_y \leq 1$ and $0 < \theta \leq 1$, we get

$$
\begin{aligned}
\mathcal{V}^{k+1} \leq\ & \frac{1}{\eta_x} \|x^k - x^\star\|^2 + (1 - \mu_{xy}^2 \beta_y \eta_y) \frac{1}{\eta_y} \|y^k - y^\star\|^2 + \frac{\theta}{2\eta_y} \|y^k - y^{k-1}\|^2 \\
& + (2\eta_x + \mu_x \eta_x^2) \|\nabla \Psi^k(\hat{x}^k)\|^2 - \frac{1}{4\eta_x} \|x^k - w^{\star k}\|^2 + \frac{\theta \eta_x}{4} \|\nabla \Psi^k(\hat{x}^k)\|^2 + \frac{\theta \eta_x}{8} \|\nabla \Psi^{k-1}_{\eta_x}(\hat{x}^{k-1})\|^2 \\
& + 2\langle K^\top y^{k+1} - K^\top y^k, \hat{x}^k - x^\star \rangle - 2\theta\langle K^\top y^k - K^\top y^{k-1}, \hat{x}^{k-1} - x^\star \rangle \\
\stackrel{0 < \theta \leq 1}{\leq}\ & \frac{1}{\eta_x} \|x^k - x^\star\|^2 + (1 - \mu_{xy}^2 \beta_y \eta_y) \frac{1}{\eta_y} \|y^k - y^\star\|^2 + \frac{\theta}{2\eta_y} \|y^k - y^{k-1}\|^2 \\
& + \left(\frac{9}{4}\eta_x + \mu_x \eta_x^2\right) \|\nabla \Psi^k(\hat{x}^k)\|^2 - \frac{1}{4\eta_x} \|x^k - w^{\star k}\|^2 + \frac{\theta \eta_x}{8} \|\nabla \Psi^{k-1}_{\eta_x}(\hat{x}^{k-1})\|^2 \\
& + 2\langle K^\top y^{k+1} - K^\top y^k, \hat{x}^k - x^\star \rangle - 2\theta\langle K^\top y^k - K^\top y^{k-1}, \hat{x}^{k-1} - x^\star \rangle.
\end{aligned}
$$

To estimate $\|\nabla\Psi^k(\hat{x}^k)\|^2$ we apply Lemma 17 for the problem (15) as follows (see (18)):

$$\|\nabla\Psi^k(\hat{x}^k)\|^2 \leq \frac{A\left(1+\eta_x L_x\right)^2\|x^k - w^{\star k}\|^2}{\eta_x^2 T^\alpha}.$$

According to (18), (73) and (74), we obtain

$$
\mathcal{V}^{k+1} \overset{(18)}{\leq} \frac{1}{\eta_x}\|x^k - x^\star\|^2 + (1 - \mu_{xy}^2\beta_y\eta_y)\frac{1}{\eta_y}\|y^k - y^\star\|^2 + \frac{\theta}{2\eta_y}\|y^k - y^{k-1}\|^2
$$
$$
+ \left(\frac{9}{4}\eta_x + \mu_x\eta_x^2\right)\frac{A\left(1+\eta_x L_x\right)^2}{\eta_x^2 T^\alpha}\|x^k - w^{\star k}\|^2 - \frac{1}{4\eta_x}\|x^k - w^{\star k}\|^2
$$
$$
+ \frac{\theta\eta_x}{8}\frac{A\left(1+\eta_x L_x\right)^2}{\eta_x^2 T^\alpha}\|x^{k-1} - w^{\star k-1}\|^2
$$
$$
+ 2\langle K^\top y^{k+1} - K^\top y^k, \hat{x}^k - x^\star\rangle - 2\theta\langle K^\top y^k - K^\top y^{k-1}, \hat{x}^{k-1} - x^\star\rangle
$$
$$
\overset{(73)+(74)}{\leq} \frac{1}{\eta_x}\|x^k - x^\star\|^2 + (1 - \mu_{xy}^2\beta_y\eta_y)\frac{1}{\eta_y}\|y^k - y^\star\|^2 + \frac{\theta}{2\eta_y}\|y^k - y^{k-1}\|^2
$$
$$
+ \frac{\theta}{8\eta_x}\|x^{k-1} - w^{\star k-1}\|^2 - \frac{1}{8\eta_x}\|x^k - w^{\star k}\|^2
$$
$$
+ 2\langle K^\top y^{k+1} - K^\top y^k, \hat{x}^k - x^\star\rangle - 2\theta\langle K^\top y^k - K^\top y^{k-1}, \hat{x}^{k-1} - x^\star\rangle.
$$

**Showing that $\Delta^{k+1} \leq \theta\Delta^k$ for $k \geq 1$.** Using the definition of $\Delta^{k+1}$ (see (77)), we have

$$
\Delta^{k+1} \leq \frac{1}{\eta_x}\|x^k - x^\star\|^2 + (1 - \mu_{xy}^2\beta_y\eta_y)\frac{1}{\eta_y}\|y^k - y^\star\|^2 + \frac{\theta}{2\eta_y}\|y^k - y^{k-1}\|^2
$$
$$
+ \frac{\theta}{8\eta_x}\|x^{k-1} - w^{\star k-1}\|^2 - 2\theta\langle K^\top y^k - K^\top y^{k-1}, \hat{x}^{k-1} - x^\star\rangle
$$
$$
\leq \max\left\{\frac{2}{2+\mu_x\eta_x}, 1 - \mu_{xy}^2\beta_y\eta_y\right\}\left(\left(1 + \frac{\mu_x\eta_x}{2}\right)\frac{1}{\eta_x}\|x^k - x^\star\|^2 + \frac{1}{\eta_y}\|y^k - y^\star\|^2\right)
$$
$$
+ \theta\left(\frac{1}{2\eta_y}\|y^k - y^{k-1}\|^2 + \frac{1}{8\eta_x}\|x^{k-1} - w^{\star k-1}\|^2 - 2\langle K^\top y^k - K^\top y^{k-1}, \hat{x}^{k-1} - x^\star\rangle\right)
$$
$$
\overset{(76)}{\leq} \theta\Delta^k,
$$

where in the last inequality we take $\theta = \max\left\{\frac{2}{2+\mu_x\eta_x}, 1 - \mu_{xy}^2\beta_y\eta_y\right\}$ (see (76)).

**Showing that $\Delta^{k+1} \leq \theta\Delta^k$ for $k = 0$.** We start with inequality (81) for $k = 0$:

$$
\mathcal{V}^1 \overset{(81)}{\leq} \frac{1}{\eta_x}\|x^0 - x^\star\|^2 + (1 - \mu_{xy}^2\beta_y\eta_y)\frac{1}{\eta_y}\|y^0 - y^\star\|^2 + (2\eta_x + \mu_x\eta_x^2)\|\nabla\Psi^k(\hat{x}^0)\|^2
$$
$$
- \frac{1}{2\eta_x}\|x^0 - w^{\star 0}\|^2 + \left(\beta_y - \frac{1}{L_x}\right)\|\nabla G(\hat{x}^0) - \nabla G(x^\star)\|^2
$$
$$
- 2\langle K^\top \bar{y}^0 - K^\top y^\star, \hat{x}^0 - x^\star\rangle + 2\langle K^\top y^1 - K^\top y^\star, \hat{x}^0 - x^\star\rangle
$$
$$
\leq \frac{1}{\eta_x}\|x^0 - x^\star\|^2 + (1 - \mu_{xy}^2\beta_y\eta_y)\frac{1}{\eta_y}\|y^0 - y^\star\|^2 - \frac{1}{2\eta_x}\|x^0 - w^{\star 0}\|^2
$$
$$
+ (2\eta_x + \mu_x\eta_x^2)\|\nabla\Psi^k(\hat{x}^0)\|^2 + 2\langle K^\top y^1 - K^\top y^0, \hat{x}^0 - x^\star\rangle, \tag{82}
$$

where in the last inequality we take $\beta_y \leq \frac{1}{L_x}$ (see (72)). Now, we apply Lemma 1 to the problem (15) for $k = 0$ (see (18)):

$$\|\nabla\Psi^k(\hat{x}^0)\|^2 \leq \frac{A(1+\eta_x L_x)^2\|x^0 - w^{\star 0}\|^2}{\eta_x T^\alpha}. \tag{83}$$

Substituting (83) into (82), we can get

$$
\begin{aligned}
\mathcal{V}^1 \quad \leq \quad & \frac{1}{\eta_x}\|x^0 - x^\star\|^2 + (1 - \mu_{xy}^2\beta_y\eta_y)\frac{1}{\eta_y}\|y^0 - y^\star\|^2 - \frac{1}{2\eta_x}\|x^0 - w^{\star 0}\|^2 \\
& + (2\eta_x + \mu_x\eta_x^2)\frac{A(1 + \eta_x L_x)^2}{\eta_x T^\alpha}\|x^0 - w^{\star 0}\|^2 + 2\langle K^\top y^1 - K^\top y^0, \hat{x}^0 - x^\star\rangle \\
\overset{(73)}{\leq} \quad & \frac{1}{\eta_x}\|x^0 - x^\star\|^2 + (1 - \mu_{xy}^2\beta_y\eta_y)\frac{1}{\eta_y}\|y^0 - y^\star\|^2 - \frac{1}{8\eta_x}\|x^0 - w^{\star 0}\|^2 \\
& + 2\langle K^\top y^1 - K^\top y^0, \hat{x}^0 - x^\star\rangle,
\end{aligned} \tag{84}
$$

where in the last inequality we take $T$ according to (73). According to (77) and (78), we have

$$
\begin{aligned}
\Delta^1 \quad \overset{(77)}{=} \quad & \mathcal{V}^1 + \frac{1}{8\eta_x}\|x^0 - w^{\star 0}\|^2 - 2\langle K^\top y^1 - K^\top y^0, \hat{x}^0 - x^\star\rangle \\
\overset{(84)}{\leq} \quad & \frac{1}{\eta_x}\|x^0 - x^\star\|^2 + (1 - \mu_{xy}^2\beta_y\eta_y)\frac{1}{\eta_y}\|y^0 - y^\star\|^2 \\
\leq \quad & \theta\left(\left(1 + \frac{\mu_x\eta_x}{2}\right)\frac{1}{\eta_x}\|x^0 - x^\star\|^2 + \frac{1}{\eta_y}\|y^0 - y^\star\|^2\right) \\
= \quad & \theta\Delta^0,
\end{aligned}
$$

where in the last inequality we used the inequalities $1 \leq \theta\left(1 + \frac{\mu_x\eta_x}{2}\right)$ and $1 - \mu_{xy}^2\beta_y\eta_y \leq \theta$, which follow from (76).

**Showing that $\Delta^{k+1} \geq 0$ for $k \geq 0$.** Finally, we need to show that $\Delta^k \geq 0$ for every $k$. This is obvious for $k = 0$ from . Using the Cauchy-Schwarz inequality, the definition of $L_{xy}$ as the norm of $K$ (see (2)), and applying Young's inequality, we get

$$
\begin{aligned}
\Delta^{k+1} \quad \overset{(77)}{=} \quad & \left(1 + \frac{\mu_x\eta_x}{2}\right)\frac{1}{\eta_x}\|x^{k+1} - x^\star\|^2 + \frac{1}{\eta_y}\|y^{k+1} - y^\star\|^2 + \frac{1}{2\eta_y}\|y^{k+1} - y^k\|^2 \\
& + \frac{1}{8\eta_x}\|x^k - w^{\star k}\|^2 - 2\langle K^\top y^{k+1} - K^\top y^k, \hat{x}^k - x^\star\rangle \\
\overset{(2)}{\geq} \quad & \left(1 + \frac{\mu_x\eta_x}{2}\right)\frac{1}{\eta_x}\|x^{k+1} - x^\star\|^2 + \frac{1}{\eta_y}\|y^{k+1} - y^\star\|^2 + \frac{1}{2\eta_y}\|y^{k+1} - y^k\|^2 \\
& + \frac{1}{8\eta_x}\|x^k - w^{\star k}\|^2 - 2L_{xy}\|y^{k+1} - y^k\|\|\hat{x}^k - x^\star\| \\
\geq \quad & \left(1 + \frac{\mu_x\eta_x}{2}\right)\frac{1}{\eta_x}\|x^{k+1} - x^\star\|^2 + \frac{1}{\eta_y}\|y^{k+1} - y^\star\|^2 + \left(\frac{1}{2} - L_{xy}B\eta_y\right)\frac{1}{\eta_y}\|y^{k+1} - y^k\|^2 \\
& + \frac{1}{8\eta_x}\|x^k - w^{\star k}\|^2 - \frac{L_{xy}}{B}\|\hat{x}^k - x^\star\|^2 \\
\geq \quad & \left(1 + \frac{\mu_x\eta_x}{2}\right)\frac{1}{\eta_x}\|x^{k+1} - x^\star\|^2 + \frac{1}{\eta_y}\|y^{k+1} - y^\star\|^2 + \left(\frac{1}{2} - L_{xy}B\eta_y\right)\frac{1}{\eta_y}\|y^{k+1} - y^k\|^2 \\
& + \frac{1}{8\eta_x}\|x^k - w^{\star k}\|^2 - \frac{2L_xy}{B}\|\hat{x}^k - x^{k+1}\|^2 - \frac{2L_xy}{B}\|x^{k+1} - x^\star\|^2.
\end{aligned}
$$

for any $B > 0$. According to the definition of function $\Psi^k$ (see 15), selecting $B = \frac{1}{2L_{xy}\eta_y}$, we get

$$
\begin{aligned}
\Delta^{k+1} \quad \geq \quad & \left(1 + \frac{\mu_x\eta_x}{2}\right)\frac{1}{\eta_x}\|x^{k+1} - x^\star\|^2 + \frac{1}{\eta_y}\|y^{k+1} - y^\star\|^2 + \frac{1}{8\eta_x}\|x^k - w^{\star k}\|^2 \\
& - 4L_{xy}^2\eta_y\eta_x\frac{1}{\eta_x}\|\eta_x\nabla\Psi_{\eta_x}^k(\hat{x}^k)\|^2 - 4L_{xy}^2\eta_y\eta_x\frac{1}{\eta_x}\|x^{k+1} - x^\star\|^2 \\
= \quad & \left(1 - 4L_{xy}^2\eta_y\eta_x + \frac{\mu_x\eta_x}{2}\right)\frac{1}{\eta_x}\|x^{k+1} - x^\star\|^2 \\
& + \frac{1}{\eta_y}\|y^{k+1} - y^\star\|^2 + \frac{1}{8\eta_x}\|x^k - w^{\star k}\|^2 - 4L_{xy}^2\eta_y\eta_x\frac{1}{\eta_x}\|\eta_x\nabla\Psi_{\eta_x}^k(\hat{x}^k)\|^2. \quad (85)
\end{aligned}
$$

Choosing stepsizes $\eta_x$, $\eta_y$ according to (74) and (75), we can derive the following inequality:

$$32 L_{xy}^2 \eta_x \eta_y \leq 1. \tag{86}$$

Now, plugging (86) and (18) into (85), we obtain

$$
\begin{aligned}
\Delta^{k+1} \overset{(86)}{\geq} \ & \left( \frac{7}{8} + \frac{\mu_x \eta_x}{2} \right) \frac{1}{\eta_x} \| x^{k+1} - x^\star \|^2 + \frac{1}{\eta_y} \| y^{k+1} - y^\star \|^2 \\
& + \frac{1}{8\eta_x} \| x^k - w^{\star k} \|^2 - \frac{1}{8\eta_x} \| \eta_x \nabla \Psi_{\eta_x}^k (\hat{x}^k) \|^2 \\
\overset{(18)}{\geq} \ & \left( \frac{7}{8} + \frac{\mu_x \eta_x}{2} \right) \frac{1}{\eta_x} \| x^{k+1} - x^\star \|^2 + \frac{1}{\eta_y} \| y^{k+1} - y^\star \|^2 \\
& + \frac{1}{8\eta_x} \| x^k - w^{\star k} \|^2 - \frac{1}{8\eta_x} \frac{C \left( 1 + \eta_x L_x \right)^2}{T^\alpha} \| x^k - w^{\star k} \|^2 \\
\overset{(73)}{\geq} \ & \frac{1}{2\eta_x} \| x^{k+1} - x^\star \|^2 + \frac{1}{\eta_y} \| y^{k+1} - y^\star \|^2 ,
\end{aligned}
$$

where in last inequality we use (73) to evaluate two last terms from below by zero. $\qquad\square$

### E.3 Proof of Theorem 2

To prove the informal result (19), we simply apply the above theorem. Using the definition of $\theta$, the theorem implies that

$$k \geq \frac{1}{1 - \theta} \log \frac{1}{\varepsilon} \quad \Rightarrow \quad \Delta^k \leq \varepsilon \Delta^0.$$

It remains to remark that the number of outer iterations to find $\varepsilon$-solution is equal to

$$
\begin{aligned}
\mathcal{O} \left( \frac{1}{1 - \theta} \log \frac{1}{\varepsilon} \right) \ &= \ \mathcal{O} \left( \max \left\{ 1 + \frac{2}{\mu_x \eta_x}, \frac{1}{\mu_{xy}^2 \beta_y \eta_y} \right\} \log \frac{1}{\varepsilon} \right) \\
&= \ \mathcal{O} \left( \max \left\{ 1 + \frac{1}{\mu_x \eta_x}, \frac{L_x}{\mu_{xy}^2 \eta_y}, \frac{L_{xy}^2}{\mu_{xy}^2} \right\} \log \frac{1}{\varepsilon} \right) \\
&= \ \mathcal{O} \left( \max \left\{ 1 + \sqrt{\frac{L_x}{\mu_x}} \frac{L_{xy}}{\mu_{xy}}, \frac{L_{xy}^2}{\mu_{xy}^2} \right\} \log \frac{1}{\varepsilon} \right) .
\end{aligned}
$$

In other words, the number of computations of prox is equal to

$$\sharp\mathrm{prox} = \mathcal{O} \left( \max \left\{ 1 + \sqrt{\frac{L_x}{\mu_x}} \frac{L_{xy}}{\mu_{xy}}, \frac{L_{xy}^2}{\mu_{xy}^2} \right\} \log \frac{1}{\varepsilon} \right) ,$$

and the number of gradient evaluations is

$$
\begin{aligned}
\sharp\nabla G \ &= \ k \cdot T \\
&= \ \mathcal{O} \left( \max \left\{ 1 + \sqrt{\frac{L_x}{\mu_x}} \frac{L_{xy}}{\mu_{xy}}, \frac{L_{xy}^2}{\mu_{xy}^2} \right\} \log \frac{1}{\varepsilon} \right) \cdot \mathcal{O} \left( \left( \frac{L_x}{\mu_x} \right)^{1/\alpha} \right) \\
&= \ \mathcal{O} \left( \max \left\{ \left( \frac{L_x}{\mu_x} \right)^{1/\alpha} + \left( \frac{L_x}{\mu_x} \right)^{\frac{2+\alpha}{2\alpha}} \frac{L_{xy}}{\mu_{xy}}, \left( \frac{L_x}{\mu_x} \right)^{1/\alpha} \frac{L_{xy}^2}{\mu_{xy}^2} \right\} \log \frac{1}{\varepsilon} \right) .
\end{aligned}
$$

---

**Algorithm 5** APDA with Inexact Prox and Accelerated Gossip

---

1: **Input**: Initial point $(\mathbf{x}^0, \mathbf{y}^0) \in \mathbb{R}^{d_x} \times \mathbb{R}^{d_y}$, $\bar{\mathbf{y}}^0 = \mathbf{y}^0$; Step sizes $\eta_x, \eta_y, \beta_y > 0$, $\theta \in [0, 1]$;
   Number of inner iterations $T$; Number of iterations of Accelerated Gossip $N$;
2: **for** $k = 0, 1, 2, \ldots$ **do**
3:    Find $\hat{\mathbf{x}}^k$ as a final point of $T$ iteration of method $\mathcal{M}$ for following problem:

$$\min_{\mathbf{x} \in \mathbb{R}^{d_x}} \left\{ P(\mathbf{x}) + \frac{1}{2\eta_x} \left\| \mathbf{x} - \left( \mathbf{x}^k - \eta_x \mathrm{AG}(W, \bar{\mathbf{y}}^k, N) \right) \right\|^2 \right\} \tag{87}$$

4:    $\mathbf{x}^{k+1} = \mathbf{x}^k - \eta_x \left( \nabla P(\hat{\mathbf{x}}^k) + \mathrm{AG}(W, \bar{\mathbf{y}}^k, N) \right)$
5:    $\mathbf{y}^{k+1} = \mathbf{y}^k + \eta_y \left( \mathrm{AG}(W, \hat{\mathbf{x}}^k, N) - \beta_y \mathrm{AG}(W, \mathrm{AG}(W, \mathbf{y}^k, N) + \nabla P(\hat{\mathbf{x}}^k), N) \right)$
6:    $\bar{\mathbf{y}}^{k+1} = \mathbf{y}^{k+1} + \theta \left( \mathbf{y}^{k+1} - \mathbf{y}^k \right)$
7: **end for**
8: **procedure** AG$(W, \mathbf{x}, N)$                                                               = Accelerated Gossip
9:    Set $a_0 = 1$, $a_1 = c_2$, $\mathbf{x}_0 = \mathbf{x}$, $\mathbf{x}_1 = c_2 \left( I - c_3 W \right) \mathbf{x}$
10:    **for** $i = 1, \ldots, N - 1$ **do**
11:        $a_{i+1} = 2c_2 a_i - a_{i-1}$, $\mathbf{x}_{i+1} = 2c_2 \left( I - c_2 W \right) \mathbf{x}_i - \mathbf{x}_{i-1}$
12:    **end for**
13:    **return** $\mathbf{x} - \frac{\mathbf{x}_N}{a_N}$
14: **end procedure**

---

# F  Analysis of the Accelerated Primal-Dual Algorithm with Inexact Prox and Accelerated Gossip (Algorithm 5)

## F.1  Gossip matrices

In Algorithm 5 we work with a more general notion of a gossip matrix (beyond graph Laplacians), defined next.

**Assumption 6** (see [Scaman et al., 2017]). *Let $\mathcal{G} = (\mathcal{V}, \mathcal{E})$ be a connected communication network. The communication process is represented via multiplication by a matrix $\hat{W} \in \mathbb{R}^{n \times n}$ which satisfies the following conditions:*

- *$\hat{W}$ is symmetric,*

- *$\hat{W}$ is positive semi-definite,*

- *the kernel of $\hat{W}$ satisfies $\mathrm{Ker}\, \hat{W} = \mathrm{span}\{(1, \ldots, 1)^\top \in \mathbb{R}^n\}$, and*

- *$\hat{W}_{i,j} \neq 0$ if and only if $i = j$ or $(i, j) \in \mathcal{E}$.*

It is easy to see that $\sigma(\hat{W}) \subset \sigma(W)$, where $\sigma(\cdot)$ denotes the spectrum, and $W = \hat{W} \otimes I_{dn}$.

## F.2  Theorem 3 follows from Theorem 2

Our main result for Algorithm 5, i.e., Theorem 3, follows from Theorem 2. Below we state it more formally.

**Theorem 9** (Formal version of Theorem 3; convergence for Algorithm 5). *Let us invoke the same assumptions as those made in Theorem 2. Additionally, let Assumption 6 hold. Assume that the auxiliary problem* (87) *is solved by one of the methods from Lemma 1. Set the parameters $N$, $c_1$, $c_2$, $c_3$ to*

$$N = \lfloor \sqrt{\chi} \rfloor, \quad c_1 = \frac{\sqrt{\chi} - 1}{\sqrt{\chi} + 1}, \quad c_2 = \frac{\sqrt{\chi} + 1}{\sqrt{\chi} - 1}, \quad c_3 = \frac{2\chi}{(1 + \chi) \lambda_{\max}(W)}$$

*and let*

$$\lambda_1 = 1 + \frac{2c_1^N}{1 + c_1^{2N}}, \quad \lambda_2 = 1 - \frac{2c_1^N}{1 + c_1^{2N}},$$

$$\beta_y = \min\left\{\frac{1}{L_x}, \frac{1}{2\lambda_1^2\eta_y}\right\}, \quad \eta_x = \frac{1}{2\sqrt{L_x\mu_x}}\frac{\lambda_2}{\lambda_1}, \quad \eta_y = \frac{\sqrt{L_x\mu_x}}{\sqrt{2}\lambda_1\lambda_2},$$

$$\theta = \max\left\{\frac{2}{2 + \mu_x\eta_x}, 1 - \lambda_2^2\beta_y\eta_y\right\},$$

$$T = \sqrt[\alpha]{20A}\left(1 + \sqrt{\frac{L_x}{\mu_x}}\right)^{2/\alpha}.$$

*Then for the Lyapunov function $\Delta^k$ from Theorem 8, we have*

$$0 \le \Delta^k \le \theta^k\Delta^0 \qquad \forall k \ge 0,$$

*Moreover, for every $\varepsilon > 0$, Algorithm 5 finds $(\mathbf{x}^k, \mathbf{y}^k)$ for which $\Delta^k \le \varepsilon\Delta^0$ in at most $\mathcal{O}\left(\kappa^{\frac{2+\alpha}{2\alpha}}\log\left(1/\varepsilon\right)\right)$ gradient computations and at most $\mathcal{O}\left(\sqrt{\kappa\chi}\log\left(1/\varepsilon\right)\right)$ communication rounds.*

*Proof.* The main idea of the proof is this methods is to show two following things: i) Theorem 2 holds true with some replacements, i.e. $L_{xy} \to \lambda_1$, $\mu_{xy} \to \lambda_2$, where $\lambda_1, \lambda_2$ is upper bound and lower bound of the spectrum of a matrix, which is defined below; ii) Estimate $\lambda_1$ and $\lambda_2$.

The proof of Theorem 9 is similar to the proof of Corollary 1 from [Kovalev et al., 2020]. According to the proof from [Kovalev et al., 2020], we need to replace

$$L_{xy} = \lambda_{\max}(\sqrt{W}) \to \lambda_1 = 1 + \frac{2c_1^N}{1 + c_1^{2N}}, \quad \mu_{xy} = \lambda_{\min}^+(\sqrt{W}) \to \lambda_2 = 1 - \frac{2c_1^N}{1 + c_1^{2N}}.$$

$\square$

# G Proof of Lemma 1

*Proof.* We consider the three options separately. We will use $f$ instead of $\Psi$ and $x$ instead of $w$ in the proof. Let $T = 2K$.

    (i) $K$ **iterations of GD followed by** $K$ **iterations GD.** We consider the GD method with stepsize $\gamma = \frac{1}{L}$, which performs the iterations

$$x^{k+1} = x^k - \gamma \nabla f(x^k) = x^k - \frac{1}{L} \nabla f(x^k),$$

where $L$ is the smoothness constant of $f$. We assume that $f$ is convex.

• **Gradient decreases monotonically.** According to the chain of inequalities

$$
\begin{aligned}
\|\nabla f(x^{k+1})\|^2 &= \|\nabla f(x^{k+1}) - \nabla f(x^k)\|^2 + 2\langle \nabla f(x^{k+1}) - \nabla f(x^k), \nabla f(x^k) \rangle + \|\nabla f(x^k)\|^2 \\
&= \|\nabla f(x^{k+1}) - \nabla f(x^k)\|^2 - \frac{2}{\gamma}\langle \nabla f(x^{k+1}) - \nabla f(x^k), x^{k+1} - x^k \rangle + \|\nabla f(x^k)\|^2 \\
&\leq \left(1 - \frac{2}{\gamma L}\right)\|\nabla f(x^{k+1}) - \nabla f(x^k)\|^2 + \|\nabla f(x^k)\|^2 \\
&\leq \|\nabla f(x^k)\|^2,
\end{aligned}
\tag{88}
$$

where the first inequality follows from convexity and $L$-smoothness, the gradient norm decreases monotonically.

• **Bound on best gradient norm.** Further, using $L$-smoothness of $f$, we obtain

$$
\begin{aligned}
f(x^{k+1}) &\leq f(x^k) + \langle \nabla f(x^k), x^{k+1} - x^k \rangle + \frac{L}{2}\|x^{k+1} - x^k\|^2 \\
&= f(x^k) - \gamma\|\nabla f(x^k)\|^2 + \frac{\gamma^2 L}{2}\|\nabla f(x^k)\|^2 \\
&= f(x^k) - \frac{1}{2L}\|\nabla f(x^k)\|^2.
\end{aligned}
$$

Summing up from $k = K$ to $k = 2K$, we get

$$\sum_{k=K}^{2K} \|\nabla f(x^k)\|^2 \leq 2L\left(f(x^K) - f(x^{2K})\right) \leq 2L\left(f(x^K) - f^\star\right),$$

which implies

$$\min_{k \in [K, 2K]} \|\nabla f(x^k)\|^2 \leq \frac{2L\left(f(x^K) - f^\star\right)}{K+1}. \tag{89}$$

• **Bound on function suboptimality.** It is known that for convex $L$-smooth functions, Gradient Descent (GD) with constant stepsize $\gamma = \frac{1}{L}$, i.e., the method

$$x^{k+1} = x^k - \gamma \nabla f(x^k) = x^k - \frac{1}{L} \nabla f(x^k),$$

satisfies [Nesterov, 2004]

$$f(x^K) - f^\star \leq \frac{L\|x^0 - x^\star\|^2}{2K}. \tag{90}$$

• **Bound on last gradient norm.** By combining (88), (90) and (89), we get

$$
\begin{aligned}
\|\nabla f(x^{2K})\|^2 &\overset{(88)}{=} \min_{k \in [K, 2K]} \|\nabla f(x^k)\|^2 \\
&\overset{(89)}{\leq} \frac{2L(f(x^K) - f^\star)}{K+1} \\
&\leq \frac{2L(f(x^K) - f^\star)}{K} \\
&\overset{(90)}{\leq} \frac{L^2\|x^0 - x^\star\|^2}{K^2}.
\end{aligned}
$$

(ii) $K$ **iterations of FGD followed by** $K$ **iterations GD.** A better rate can be obtained if we replace the first half of the iterative process with the Fast Gradient Descent (FGD) method [Nesterov, 2004].

• **Bound on function suboptimality.** If we employ the Fast Gradient Descent (FGD) method during the first $K$ iterations, we get (see, for example, [Nesterov, 2004]):

$$f(x^K) - f^\star \leq \frac{4L\|x^0 - x^\star\|^2}{K^2}. \tag{91}$$

• **Bound on last gradient norm.** Since in the last $K$ iterations we use GD, inequalities (88) and (89) still apply. It remains to combine (88), (89) and (91):

$$
\begin{aligned}
\|\nabla f(x^{2K})\|^2 &\overset{(88)}{=} \min_{k \in [K, 2K]} \|\nabla f(x^k)\|^2 \\
&\overset{(89)}{\leq} \frac{2L(f(x^K) - f^\star)}{K + 1} \\
&\leq \frac{2L(f(x^K) - f^\star)}{K} \\
&\overset{(91)}{\leq} \frac{8L^2\|x^0 - x^\star\|^2}{K^3}.
\end{aligned}
$$

(iii) $K$ **iterations of FGD followed by** $K$ **iterations FSFOM.** A better rate can be obtained if we further replace the second half of the iterative process with the FSFOM method of Kim and Fessler [2021].

• **Bound on the gradient norm of FSFOM.** The FSFOM method satisfies the following inequality (see Theorem 6.1 from [Kim and Fessler, 2021]):

$$\|\nabla f(x^{2K})\|^2 \leq \frac{4L(f(x^K) - f^\star)}{K^2}. \tag{92}$$

• **Bound on last gradient norm.** By combining (92) and (91), we obtain

$$\|\nabla f(x^{2K})\|^2 \overset{(92)}{\leq} \frac{4L(f(x^K) - f^\star)}{K^2} \overset{(91)}{\leq} \frac{16L^2\|x^0 - x^\star\|^2}{K^4}.$$

Results from parts (i), (ii) and (iii) proved above can be written in a unified form as

$$\|\nabla f(x^T)\|^2 \leq \frac{AL^2\|x^0 - x^\star\|^2}{T^\alpha},$$

where $T = 2K$, and $A$ is a constant which depends on the combination of the methods used during the first $K$ iterations and the last $K$ iterations.

$\square$