# OpenReview forum: "Communication Acceleration of Local Gradient Methods via an Accelerated Primal-Dual Algorithm with an Inexact Prox"
_NeurIPS.cc/2022/Conference — NeurIPS 2022 Accept_

### Official Review · Reviewer_qFrD · 2022-06-28

**Rating:** 7
**Confidence:** 4
**Soundness:** 3 good
**Presentation:** 4 excellent
**Contribution:** 3 good

**Summary:**

This paper presents three algorithms for finding approximate solutions of a strongly convex composition optimization problem with objective function $G(x) + F(Kx)$ for linear operator $K$. More specifically, each algorithm is based on the classic Chambolle-Pock primal-dual algorithm under different assumptions on the problem of interest. Specifically, the first one deals with the case where the resolvents of $F$ and $G$ are both simple, the second one is for the case where only the resolvent of $F$ is simple, and the third deals with an extension of the second one for federated learning. In particular, the third one makes use of a known technique (accelerated gossip) in order to improve some dependencies on certain universal constants. It is then shown that some versions of the proposed algorithms improve on existing communication efficient methods, such as ProxSkip, from various theoretical angles.

**Questions:**

1. Assumption 4 seems to imply that $K$ has full row rank in the case of general $F$. Perhaps a comment should be made about this to the reader?

2. Line 66 says that ProxSkip "solves" (1), which normally means that an *exact* solution is found. Since the tolerance $\varepsilon$ is considered in this work, the authors should change this to mean "inexactly solves" (1) and precisely define what an $\varepsilon$-inexact solution is.

3. Line 71 references the number of prox evaluations wrt $F$, but the algorithms use the prox of $F^*$. While these are equivalent, a remark should be made about this with appropriate references, e.g., using the Moreau Decomposition Theorem.

4. Theorem 2 should be slightly more self-contained. Specifically, both $\alpha$ and $\cal M$ should be described there so that the reader does not need to read Lemma 1 or the beginning of Section 5 in order to understand these quantities.

5. In line 228, the assertion of $\psi^{*}(\cdot)\equiv 0$ for every indicator function $\psi(\cdot)$ is false. For a closed convex set $S$, the conjugate of the indicator function of $S$ is called the support function of $S$, which is generally not the zero function. It just so happens to be the case that when $S=${$0$}, the support function is also identically zero.

6. The replacement of $\in$ with $=$ mentioned in lines 129-130 is confusing in this paper, as the notation $=$ in algorithms is typically interpreted as assignment. I would prefer (at least in Algorithms 1-3) that $=$ be replaced with $\in$ and appropriate adjustments, e.g., $x = A(x)$ becomes ${\bf \rm find\ } x \in A(x)$.

7. Some small typos:
* line 59: The part "Letting $d_x=$...$ is a sentence fragment.
* line 78: oder -> order

**Limitations:**

The authors have sufficiently addressed the possible limitations (assumptions) of their work.

**Strengths And Weaknesses:**

**Disclaimer**: I have not read the appendices of this work and, hence, I am limiting my review to the 9-page main body of this work.

*Strengths*:

1. The improved complexity bounds are novel and significant. I especially like the connections of the techniques to existing literature.

2. The literature review and comparisons in Table 1-2 are much appreciated. This gives clarity to how it fit into other state-of-art methods.

3.  The overall presentation of the material is well-organized and easy to read. I especially like the summary in Section 2 and the review in Section 3. The writing style is clean and there are very few grammatical issues.

*(Minor) Weaknesses*:

1. Some parts of the technical results lack clarity or require further discussion and elaboration (see the *Questions* section below).

2. There are a few minor technical mistakes (see the *Questions* section below).

As the strengths of this paper greatly outweigh its few minor weaknesses, I recommend its acceptance to NeurIPS, under the mild condition that the simple comments in the  *Questions* section below are sufficiently addressed.

---

> ### Author Response · Authors · 2022-08-01
> **Response to Reviewer qFrD**
>
> Thanks for the review and positive evaluation of our work!!!
>
> >1. Assumption 4 seems to imply that $K$ has full row rank in the case of general $F$. Perhaps a comment should be made about this to the reader?
>
> Please read our reply to Reviewer EGED where we handle the same question.
>
> >2. Line 66 says that ProxSkip "solves" (1), which normally means that an exact solution is found. Since the tolerance $\epsilon$ is considered in this work, the authors should change this to mean "inexactly solves" (1) and precisely define what an $\epsilon$-inexact solution is.
>
> Adding the definition of the $\epsilon$-inexact solution makes sense; we will do it. [However, the wording is used is ubiquitous when numerical iterative methods are considered, and we have used it in many papers ourselves before.]
>
> > 3. Line 71 references the number of prox evaluations wrt $F$, but the algorithms use the prox of $F^\star$. While these are equivalent, a remark should be made about this with appropriate references, e.g., using the Moreau Decomposition Theorem.
>
> Yes, we will state this known fact explicitly for better readability. According to Moreau decomposition theorem,
>
> $$prox_{F}(x) +prox_{F^{\star}}(x) = x,$$
>
> so evaluation of prox $F$ can be replaced with evaluation of prox $F^{\star}$, and vice versa.
>
> > 4. Theorem 2 should be slightly more self-contained. Specifically, both $\alpha$ and $\mathcal{M}$ should be described there so that the reader does not need to read Lemma 1 or the beginning of Section 5 in order to understand these quantities.
>
> This would be ideal. Unfortunately, due to the page limits we had to reduce the statements of some theorems. We will think more how to rewrite it to make this more clear.
>
> > 5. In line 228, the assertion of  $\psi^{\star}(\cdot) = 0$ for every indicator function  $\psi(\cdot)$ is false. For a closed convex set  $S$, the conjugate of the indicator function of  $S$ is called the support function of $S$, which is generally not the zero function. It just so happens to be the case that when  $S =\{0\}$, the support function is also identically zero.
>
> Unfortunately, we made a mistake, you are right. Thanks for catching this.
>
> > 6. The replacement of $\in$ with = mentioned in lines 129-130 is confusing in this paper, as the notation = in algorithms is typically interpreted as assignment. I would prefer (at least in Algorithms 1-3) that = be replaced with
> $\in$ and appropriate adjustments, e.g.,
> $x=A(x)$ becomes $\text{find}~x\in A(x)$.
>
> We will replace = with $\in$ in appropriate places.
>
> > 7. Some small typos
>
> We will fix them all.

---

> > ### Comment · Reviewer_qFrD · 2022-08-03
> > **Thank you for the responses**
> >
> > Thank you for your points of clarification and responses to my comments.
> >
> > I look forward to the proposed changes, and I am maintaining my positive impression of your work.

---

> > > ### Author Response · Authors · 2022-08-03
> > > **Thank you**
> > >
> > > Dear Reviewer qFrD,
> > >
> > > Thank you for letting us know, we very much appreciate that. And thanks one more time for donating your time to reading and evaluating our work.
> > >
> > > Kind regards,
> > >
> > > author(s) of Paper12515

---

### Official Review · Reviewer_zYvm · 2022-07-08

**Rating:** 6
**Confidence:** 4
**Soundness:** 4 excellent
**Presentation:** 2 fair
**Contribution:** 3 good

**Summary:**

Authors develop a new method for the problem G(x) + F(Kx), with G being strongly convex and smooth, F being convex and K being a matrix. Via a saddle point reformulation and accelerated minmax algorithms, they improve upon previous approaches and can remove some assumptions. They apply their framework to federated learning and improve on the number of local steps that are required between communication rounds.


**Questions:**

This work shows that one does not need the smoothness condition on F* if one has a prox access to F*. In L107 it is said that this is important for some applications (it also talks about \mu_x strongly convex, but that was an issue that was also addressed in previous work). My question is, what are some applications that do not require F* to be smooth. In the federated learning application, after the saddle point reformulation F* is 0, so this does not apply. Authors should provide such an example and reference it in L107


**Limitations:**

yes

**Strengths And Weaknesses:**

In general I am positive about this work. The paper obtains some improvements on solutions for problem (1) and also to federated learning, as a consequence of their framework plus some modifications. The paper is written in a self contained way that allows to follow the analysis well. But on the other hand this paper needs to state and cite clearly the theoretical analyses and ideas that are taken from previous works. It is hard to pinpoint the new ideas this paper introduces and I think the paper does not do a good job on explaining these. Everything is re-proved (which is not a bad thing per se) but without clear indications of what was already done and what is new, see below for examples, but for instance the results of lemma 1 were known and there is no direct reference to these, only to the subalgorithms that are used.

Let me elaborate on my interpretation of the work and how many of these ideas could be better attributed:

This method solves problem (1) by combining the structure of the accelerated algorithm in (Kovalev, Gasnikov, Richtárik 2021) and the method in (Chambolle, Pock, 2011) but substituting the gradient update step with a subgradient of F* by an approximate implicit step. The former work already considered a problem with the structure (1) among its applications and the optimization of the saddle point problem and how this applies to distributed setting. Such work already had the techniques to accelerate with linear rates when F* is not strongly convex, provided that the matrix K satisfies some regularity conditions, as it is assumed in this work as well. This *is* mentioned in the main paper.

In accelerated algorithms of this kind, it is known how to obtain a similar accelerated algorithm by means of computing an approximate proximal operator, which intuitively means that one substitutes an explicit (sub)gradient step that occurs in the algorithm by a sufficiently precise aproximate implicit step. Indeed, an inexact implicit step of this exact kind has been used previously for general convex smooth optimization in "Adaptive Catalyst for Smooth Convex Optimization, Ivanova et al" (effectively allowing a Catalyst-like reduction without incurring an extra logarithmic factor, as it happened with the original Catalyst paper) and has been improved in several works like in "Contracting proximal methods for smooth convex optimization" by allowing constrained problems, among other works. All these works require a subroutine to minimize the norm of the gradient of a function coming from the objective of the proximal subroutine. There is no comment on this. Also, L461 "the result is not new of course, but we could not find a source for the proof we include here". Citations to the closest works should be provided

The problem of minimizing gradient norm for solving the subroutine for a smooth and convex function is already known. Also, explaining all the possible different suboptimal variants seems redundant to me.

The accelerated gossip is already known as well. And how to apply it to this setting was already studied in (Kovalev et al 2020)

So one wonders what is left. There is certainly some value in having put together all of these things, and on combining the techniques of (Kovalev et al 2021) with Chambolle Pock. But it is hard to pinpoint any other contribution beyond this (and the work certainly does not explain this clearly) except for maybe not assuming that F* is smooth, although forgoing this assumption seems to come simply from the fact that one assumes access to the exact prox of F*.

I can see how the writing of this paper can make many people that are not so familiar with the related literature wonder what is the contribution of this work and what were already known things.

These are concrete things that I think would help to make the presentation more
+ There should be more citations about where to find the original analyses of several things that are re-proven, like the closest approximation to the results in section B
+ state clearly which parts of the analysis of Chambolle-Pock in appendix C are re-proven in that
+ it would be nice to have a comparison of appendix D with (Kovalev et al,
+ sections E and F seem fine to
+ Then again appendix G needs to state clearly that all these methods were known and provide references for all, only Kim and Fessler is being cited and it seems it is only cited when the paper refers to FSFOM, but Kim and Fessler 2021 also had essentially the same final result, like in (29) of the arxiv version of their paper.

I don't think the reduction in font size in the tables is allowed, according to the NeurIPS style guidelines: "do not change font sizes (except perhaps in the References section; see below)". It only happens in some (extensive) footnotes for the tables, but strictly speaking this can be an argument for desk rejection.

Minor:

L72 "for the number gradient steps" missing 'of'.
L171 "Step 3" it would be better if you write "Line 3" (since line 1 is not a step) and even better if you use \ref by having a label in that line in the algorithm
L208 "Gradient Decent"
footnote in page 8 "Assumtion". Also, the sentence should be rephrased, as it does not make a lot of sense, as is.
L131 "in line 8" The pseudocode of Algorithm does not have numbers. Please add them
L478 the text assumes that F^\star is convex but it should say it is \mu_y strongly convex
L607 "be a exact solution" -> "be an exact solution"
L677 Fix sentence "The main idea of the proof is this methods is to show two following things"
L689 typo "requared"

---

> ### Author Response · Authors · 2022-08-01
> **Response to Reviewer zYvm (Part 1)**
>
> Thanks for a thorough review!
>
> > In accelerated algorithms of this kind, it is known how to obtain a similar accelerated algorithm by means of computing an approximate proximal operator, which intuitively means that one substitutes an explicit (sub)gradient step that occurs in the algorithm by a sufficiently precise aproximate implicit step. Indeed, an inexact implicit step of this exact kind has been used previously for general convex smooth optimization in "Adaptive Catalyst for Smooth Convex Optimization, Ivanova et al" (effectively allowing a Catalyst-like reduction without incurring an extra logarithmic factor, as it happened with the original Catalyst paper) and has been improved in several works like in "Contracting proximal methods for smooth convex optimization" by allowing constrained problems, among other works. All these works require a subroutine to minimize the norm of the gradient of a function coming from the objective of the proximal subroutine.
>
> As far as we know, the statement *"all these works require a subroutine to minimize the norm of the gradient of a function coming from the objective of the proximal subroutine"* **is incorrect.** In order to compute the inexact proximal operator, the paper *Adaptive Catalyst for Smooth Convex Optimization* of Ivanova et al. does not use algorithms for gradient norm reduction like we do. **They use algorithms for function subptimality reduction which is clearly stated in Assumption 1 on the inner method $\mathcal{M}$.** This results in the additional factor in the complexity which is proportional to the logarithm of the condition number.
>
> Although the gradient norm is used for stopping criterion in the paper *Contracting Proximal Methods for Smooth convex optimization* of Doikov and Nesterov, they still have extra logarithmic factors in the complexity of the resulting algorithm. It is explicitly stated in the end of Section 4 of their paper: "However, at each step it uses a logarithmic number of steps of the basic method. It seems to be a reasonable price for the level of generality. Indeed, we are free to choose an arbitrary method as the basic one."
>
> We will add comments along these lines to the camera-ready version of this paper. Indeed, giving this broader context might be helpful to the reader interested in these other related developments.
>
> > There is no comment on this. Also, L461 "the result is not new of course, but we could not find a source for the proof we include here". Citations to the closest works should be provided
>
> It is a special case of the proximal point method for strongly monotone inclusion problem; a similar proof can be found in the following paper: R. Tyrrell Rockafellar. *Monotone Operators and the Proximal Point Algorithm*. We will add this and other citations wherever appropriate.
>
> > The problem of minimizing gradient norm for solving the subroutine for a smooth and convex function is already known.
>
> We agree with you that the problem of finding an algorithm that converges optimally according to the gradient norm is not new. However, the algorithms for gradient norm reduction play a key role in our work. **The fact that we use such algorithms to perform the inexact proximal operator computation allows us to obtain algorithms that do not have extra logarithmic factors in the complexity**. That is, the number of inner iterations is not proportional to any logarithmic factor. It is a clear advantage over the works such as *Contracting Proximal Methods for Smooth convex optimization* and *Adaptive Catalyst for Smooth Convex Optimization*. For instance, Nesterov and Doikov write: "However, at each step, it uses a logarithmic number of steps of the basic method. It seems to be a reasonable price for the level of generality. Indeed, we are free to choose an arbitrary method as the basic one."  **Our algorithm does not have such an issue.**
>
>
> > Also, explaining all the possible different suboptimal variants seems redundant to me.
>
> Suboptimal variants, such as GD, are important for practical applications, such as performing local gradient steps in federated learning.
>
> It may be redundant from a theoretical point of view since the other variants outperform GD. However, ProxSkip uses GD, and we are improving on ProxSkip. So, including GD is very important to us in order to explain where the difference in our results comes from. (This is not the only difference; but it is a key difference). So, excluding the GD case would make our paper much less clear -- we are not willing to do this.

---

> ### Author Response · Authors · 2022-08-01
> **Response to Reviewer zYvm (Part 2)**
>
> > The accelerated gossip is already known as well. And how to apply it to this setting was already studied in (Kovalev et al 2020). So one wonders what is left. There is certainly some value in having put together all of these things, and on combining the techniques of (Kovalev et al 2021) with Chambolle Pock. But it is hard to pinpoint any other contribution beyond this (and the work certainly does not explain this clearly) except for maybe not assuming that $F^{\star}$ is smooth, although forgoing this assumption seems to come simply from the fact that one assumes access to the exact prox of $F^{\star}$.
>
> **There are fundamentally two ways how one can look at the results of our paper: i) what innovations we bring to the federated learning literature, ii) what innovations we bring to the proximal splitting literature.** We claim that our key innovations belong to the first category.
>
> Indeed, we improve upon the current SOTA local gradient method ProxSkip (the first work that managed to show that local steps provably lead to communication acceleration), which is one of the foundational observations of the entire field of federated learning. However, this foundation was shaky until the ProxSkip paper breakthrough. Since we improve upon ProxSkip in several ways, which we believe is clear from the paper and Tables 2 and 3, our methods  automatically enjoy the current best theoretical communication (matching that of ProxSkip in the standard setting, and improving on it in the decentralized setting) and local computation complexity: from $\tilde{O}(\kappa^{1/2})$ to $\tilde{O}(\kappa^{1/3})$ and $\tilde{O}(\kappa^{1/4})$, depending on the method  $\mathcal{M}$ used to approximately solve the prox in Algorithm APDA-Inexact, for example. We have additional improvements in the decentralized case (please see Tables 2 and 3 for a summary). **We believe these are the *key* results, and this is how our work should be *primarily* judged.**
>
> **We found that the reviewer does not look at our work this way, but instead judges it from the viewpoint ii). This is fine, since we have some innovations here as well (summarized in Table 1), but we believe this is a *secondary* and not primary way of looking at our paper.** Let us explain what we mean using a hypothetical scenario: assume that there were no innovations in (ii) whatsoever, and that all we did was application of *known* results from field X (say, proximal splitting literature) to federated learning. By the logic of the reviewer, our work would not be innovative. However, we still advance on the current best theory for local methods in field Y (federated learning). Such a transfer between fields is very valuable. Having said that, the general results we obtain in category ii) were not known. But indeed, they were obtained using a creative and precisely executed combination and adaptation of several known ideas. The innovation is in the realization that this is all possible, and in the execution of this precise combination.
>
> Please look at the summary provided in Table 1. This table compares our results for Alg 1-2 to selected benchmarks. Competing methods (CP, AltGDA and APDG) either do not work with $L_y=\infty$, or do not have linear rate if $\mu_y=0$. Providing methods that do *not* have this issue is our key innovation in category ii).
>
> Also, these methods do not have any "inner iterations" (see the last column) and hence can't be interpreted as local gradient methods. None of the benchmarks have quite the propertieswe need in order to get the improvements for the federated learning application, which is central to our work.
>
> An orthogonal contribution of our work is to show how such a method can be constructed step-by-step, from very general ideas of the proximal point method, through the Chambolle-Pock method, with two important modifications that eventually lead to a local method with the desired theoretical properties. In so doing, we are introducing to the FL audience the beauty of proximal splitting schemes, and the elegance of a general approach to designing numerical methods.
>
> > There should be more citations about where to find the original analyses of several things that are re-proven, like the closest approximation to the results in section B
>
> Will add.
>
> > state clearly which parts of the analysis of Chambolle-Pock in appendix C are re-proven in that
>
> Will add.
>
> > it would be nice to have a comparison of appendix D with (Kovalev et al,
>
> Will add.
>
> > Then again appendix G needs to state clearly that all these methods were known and provide references for all, only Kim and Fessler is being cited and it seems it is only cited when the paper refers to FSFOM, but Kim and Fessler 2021 also had essentially the same final result, like in (29) of the arxiv version of their paper.
>
> Absolutely; we apologize for the oversight. We will refer to Nesterov et al. *Primal–dual accelerated gradient methods with small-dimensional relaxation oracle* (Remark 2.1).

---

> ### Author Response · Authors · 2022-08-07
> **Our rebuttal: Does it address all your concerns to your full satisfaction?**
>
> Dear Reviewer zYvm,
>
> We believe we addressed all the issues you raised and questions you asked in 3 messages:
> - Response to Reviewer zYvm (Part 1)
> - Response to Reviewer zYvm (Part 2)
> - Re: q
>
> Please can you let us know what you think? Do our responses indeed address all the concerns? If so, please let us know! If not, also please let us know, and also why, so that we can clarify further.
>
> Thank you!
>
> (Authors)

---

> > ### Comment · Reviewer_zYvm · 2022-08-09
> > **Re: Our rebuttal: Does it address all your concerns to your full satisfaction?**
> >
> > > As far as we know, the statement "all these works require a subroutine to minimize the norm of the gradient of a function coming from the objective of the proximal subroutine" is incorrect
> >
> > You are right, my bad. These works do not use algorithms for gradient minimization. It seems this paper https://arxiv.org/pdf/2205.09647.pdf does do this, but it was published on arxiv too close to neurips deadline. It would be nice to cite it for the final version, though.
> >
> >
> > Please, make sure to add to your paper the things discussed mostly regarding attributions of previously known results or highly related (specially in "Response to Reviewer zYvm (Part 2)").
> >
> > Thank you as well for the response to my question regarding a non-smooth F*

---

> > > ### Author Response · Authors · 2022-08-09
> > > **Re: Re: Our rebuttal: Does it address all your concerns to your full satisfaction?**
> > >
> > > > You are right, my bad. These works do not use algorithms for gradient minimization.
> > >
> > > Thanks for checking and acknowledging, appreciated.
> > >
> > > > It seems this paper https://arxiv.org/pdf/2205.09647.pdf does do this, but it was published on arxiv too close to neurips deadline. It would be nice to cite it for the final version, though.
> > >
> > > Indeed, we will cite it as concurrent work.
> > >
> > > > Please, make sure to add to your paper the things discussed mostly regarding attributions of previously known results or highly related (specially in "Response to Reviewer zYvm (Part 2)").
> > >
> > > Absolutely. This will make the paper more complete and more accurate in terms of attributions. Our bad for not having done that full extent in our original submission already.
> > >
> > > > Thank you as well for the response to my question regarding a non-smooth $F^*$.
> > >
> > > Our pleasure!

---

### Official Review · Reviewer_bWvi · 2022-07-10

**Rating:** 5
**Confidence:** 4
**Soundness:** 3 good
**Presentation:** 2 fair
**Contribution:** 2 fair

**Summary:**

The paper considers convex-concave saddle-point problems. The proposed methods are a combination of multiple techniques such as inexact proximal evaluation and acceleration. In particular, the method builds heavily on Chambolle-Pock splitting with the scheme studied in [Kovalev et. al, 2021].

**Questions:**

Since this is a theory paper, I would expect more elaborated statements in the main results. For example, many specific constants in the complexity formulas are ignored. It is important to explicitly provide the parameters required to achieve the rate in the theorems. For example, what is "a suitable selection of stepsizes" in Theorem 1 and "there exist parameters of Algorithm 2..." in Theorem 2? Do they depend on unknown parameters? While the linear rate is nice theoretically, it can be much slower than sublinear rates if the convergence factor is close to 1, unless the number of iterations tends to infinity. All these parameters are important for practitioners to implement the methods.

The author mentioned in Line 107 that relaxing the convexity condition on $F^\star$ is important in some applications. Can the authors specify some of such applications in which convexity of $F^\star$ is critical but not strong convexity?

Instead of Chambolle-Pock splitting, one can also use Douglas-Rachford splitting for such a structured problem. How would we compare the two in the current paper's setting?

Any recommendation on how to efficiently implement the proposed methods?

**Limitations:**

see above

**Strengths And Weaknesses:**

The main contribution is the relaxation of the strong convexity and smoothness assumptions on the conjugate function $F^\star$. The first relaxation has been achieved in several previous works such as [Kovalev et. al, 2021] and [Du and Hu, 2019]. The contribution of allowing inexactness in the proximal evaluation is quite incremental as such a feature is abundant in proximal-type methods.

It should also be emphasized that to relax such assumptions on $F$ (in the function $F(Kx)$), the paper requires an extra assumption on the linear map $K$ [Assumption 4]. So it is not quite clear if moving a more restricted assumption from $F$ to $K$ is such a large gain. This deserves more clarification in the paper.


References:
----------

[1] Du, S. S. and Hu, W. Linear convergence of the primal-dual gradient method for convex-concave saddle point problems without strong convexity. AISTATS, 2019.

---

> ### Author Response · Authors · 2022-08-01
> **Response to Reviewer bWvi (Part 1)**
>
> Thanks for the borderline accept. We hope to convince you that our paper brings much stronger contributions than your current  evaluation (score 5) suggests. We would be delighted if you could read and consider our response.
>
> > The contribution of allowing inexactness in the proximal evaluation is quite incremental as such a feature is abundant in proximal-type methods.
>
> Note that this feature is very important in the case of distributed optimization and federated learning. In this case, function $G(x)$ is constructed from the empirical risk losses computed across the datasets stored by the compute nodes. In this case, we typically have access to the gradient $\nabla G(x)$ only and don't have access to the proximal operator of $G$. See Section 6 for details.
>
> We do not claim that the general idea of inexact evaluation of the prox as such (in some method using a prox, and there are many such methods) is novel in general (we will add some citations - can you provide some links?). However, we have a particular and novel execution of this, and we apply this in a novel setting, with the aim to shed light on a certain task that remained unsolved in the distributed/federated learning literature (FL) for almost a decade, and was only recently resolved by Mishchenko et al (2022) in their ProxSkip paper. We show that a particular novel combination of otherwise existing ideas, such as i) Chambolle-Pock splitting, ii) a modification thereof which allows us to preserve linear rate without strong convexity of $F^\star$, and iii) an inexact prox evaluation, ultimately lead to a method that in several ways outperforms the theoretical bounds of ProxSkip. We also provide a decentralized extension, also with improved theoretical complexity. **We believe it to be beautiful that such a novel combination of ideas from the field of proximal splitting ethods can shed light on a long-standing problem in a different field (FL).** In fact, we believe the simpler the ideas that can do this are, the better - there is beauty in simplicity and simple explanations typically go closer to the heart of the problem.
>
> > It should also be emphasized that to relax such assumptions on $F$ (in the function $F(Kx)$), the paper requires an extra assumption on the linear map $K$ [Assumption 4]. So it is not quite clear if moving a more restricted assumption from $F$ to $K$ is such a large gain. This deserves more clarification in the paper.
>
> **In the light of what we set out to achieve in this paper (and we succeeded in this) --- improving upon the theoretical complexity of ProxSkip, which is the current SOTA in terms of the theoretical communication complexity of local gradient type methods in the FL literature, and its decentralized variant SplitSkip (from the same paper as ProxSkip) --- this is exactly the change we want to make!** That is, in these applications, we need more freedom in terms of $F$, and the *seemingly* restrictive assumption on $K$ is *satisfied for free* (see our "Response to Reviewer EGED (Part 1)")! This is precisely why we do this trade-off. So, this question and our ability to execute this trade-off is absolutely at the heart of our approach to obtaining the improvements in the theoretical SOTA communication and local computation rates of local gradient-type methods in federated learning. We believe it is remarkable that this all is possible given that thousands of papers were written on this topic (local methods for FL) in the last few years alone.
>
> **This is a very good question, and we apologize if this was not clear from our paper. We will certainly provide a clarification of this in the camera-ready version of the paper.**
>
> > Since this is a theory paper, I would expect more elaborated statements in the main results. For example, many specific constants in the complexity formulas are ignored.
>
> In our theorem statements in the main body of the paper we use $\mathcal{O}(\cdot)$ notation, which hides universal numerical constants and additive constants that do not depend on $\epsilon$, since this is all we want the reader to pay attention to at this point. Indeed, we believe that our readership will include a large portion of the federated learning community, and we felt these details needed to be suppressed in order for the main message not to get lost in the avalanche of constants, parameters and details present in the full theorems.  More importantly, **the detailed results, including all parameter settings and all constants, are given in the appendix.** We believe this was the right choice. If you believe you are right, please let us know and support your view with arguments. We are open to discussion on this point - perhaps you see what we do not see.
>
> [ Also, but this is much less important, please also note that it is a common practice within optimization community to write such convergence results in the form $\mathcal{O}(\rho \log (1/\epsilon))$, where $\rho$ is a speed of linear convergence. ]

---

> ### Author Response · Authors · 2022-08-01
> **Response to Reviewer bWvi (Part 2)**
>
> > It is important to explicitly provide the parameters required to achieve the rate in the theorems. For example, what is "a suitable selection of stepsizes" in Theorem 1 and "there exist parameters of Algorithm 2..." in Theorem 2? Do they depend on unknown parameters? While the linear rate is nice theoretically, it can be much slower than sublinear rates if the convergence factor is close to 1, unless the number of iterations tends to infinity. All these parameters are important for practitioners to implement the methods.
>
> Thank you for pointing to this. **All parameters of our algorithms have simple explicit formulas that are provided in the full/detailed versions of the convergence theorems that can be found in the appendix.** So, these details are already contained in the paper.
>
> The resulting complexities (the complexities after the various stepsize and other parameters are substituted into the general formulas for rates) are also **shown in the three tables of the paper**.
>
> **Having said that, we will make it all even more reader-friendly, and will  add a table with the formulas for the parameters of the algorithms in the final version of the paper.**
>
>
> > The author mentioned in Line 107 that relaxing the convexity condition on $F^*$ is important in some applications. Can the authors specify some of such applications in which convexity of $F^*$ is critical but not strong convexity?
>
> We are a bit confused by this question and are not sure what reviewer meant. On line 107 we mention that it can be important to relax strong convexity to mere convexity. **The most important example of such problem is centralized/decentralized distributed optimization and federated learning (see Section 6).** In this case $F^*(x) \equiv 0$, which is a convex function but not strongly convex. **This is the key  problem we are trying to solve in this paper, and this is exactly why we needed (and provided) such a relaxation.**
>
> There might be other applications of this, but in this paper we fully focus on improving the current SOTA of local gradient-type methods in federated learning, which we believe is important enough due to the enormous attention federated learning has gained (e.g., most big data companies now have dedicated FL research and engineering teams; FL was named by Forbes as one of top 6 future AI technologies; thousands of papers have been written on this topic ...)
>
> > Instead of Chambolle-Pock splitting, one can also use Douglas-Rachford splitting for such a structured problem. How would we compare the two in the current paper's setting?
>
> Indeed, Chambolle-Pock splitting is not the only splitting that can be used in our paper. As an alternative, we could use, for instance, Condat-Vu splitting or (in some cases) Loris-Verhoven splitting. **Since Douglas-Rachford is equivalent to Chambolle-Pock in the case $K = I$, we can still analyze this algorithm under Assumptions 1-4 (in other words, our theory applies to this as well).** However, the practical application of **Douglas-Rachford is questionable in many applications since it would require to compute the proximal operator of function $F(Kx)$.** For instance, this would make no sense in the case of decentralized distributed optimization, because the resulting algorithm would require to perform exact consensus over the network at each iteration of the algorithm, while Chambolle-Pock allows to perform gossip steps only.
>
> > Any recommendation on how to efficiently implement the proposed methods?
>
> Implementation of Algorithms 1 and 2 is more or less straightforward. The difficult part is to implement the inner algorithms. GD algorithm is straightforward, but algorithms FGM + GD and FSOM (Kim and Fessler, 2021) are less practical since they require to set the number of iterations in advance. In practice, one can use Accelerated Gradient Descent with the so-called Monteiro-Svaiter stopping condition which is a practical stopping criterion. One can prove that such an algorithm achieves the state-of-the-art complexity up to logarithmic factors. As promised before, we will include some experiments in the camera ready version of the paper.

---

> ### Author Response · Authors · 2022-08-06
> **Re: Official Review of Paper12515 by Reviewer bWvi**
>
> Dear Reviewer bWvi,
>
> We did not hear from you yet, but we would be delighted to learn whether our responses answered you questions.
>
> Regards,
>
> authors

---

### Official Review · Reviewer_EGED · 2022-07-12

**Rating:** 7
**Confidence:** 3
**Soundness:** 3 good
**Presentation:** 3 good
**Contribution:** 3 good

**Summary:**


This paper aims to develop a similar method as ProxSkip, but based on PDHG instead. The proposed algorithm allows inexact $Prox_G$, as well as the non-strongly convexity of $F$.

**Questions:**


Please see above.

**Limitations:**

Yes

**Strengths And Weaknesses:**


Strengths:

1. The theory for inexact evaluation of prox of G at each iteration is nice. The possible difficulty in $Prox_G$ is often an overlooked factor.
2. The avoidance of the strong convexity of $F$ is also novel. This makes the convergence proof non-trivial.

Major concerns:

1. My first concern is on Assumption 4, where a constant $\mu_{xy}$ is defined. Although it is from another work, the validity of this assumption should still be discussed in detail.

For example, why is this assumption reasonable? What specific problems satisfy this assumption? It would be much better if some examples of $K$ and $F$ arising from real-world applications can be provided.

How small can  $\mu_{xy}$ be? It appears in the denominator of the complexities. Although it is a constant, it would be better if we know its magnitude.

2. There are no numerical comparisons with other methods, especially ProxSkip. The claimed superiority in gradient complexities should be accompanied by numerical evidence.

Minor concerns:

1. In Assumption 4, the paper you cited uses $\mu^2_{xy}$ instead of $\mu_{xy}$, should it be  $\mu^2_{xy}$?

---

> ### Author Response · Authors · 2022-08-01
> **Response to Reviewer EGED (Part 1)**
>
> Thanks for the positive evaluation of our work.
>
> >My first concern is on Assumption 4, where a constant $\mu_{xy}$ is defined. Although it is from another work, the validity of this assumption should still be discussed in detail. For example, why is this assumption reasonable? What specific problems satisfy this assumption?
>
> 1. Assumption 4 is satisfied, for example, **when the matrix $K$ has full row rank**. Practical examples of such problems (for instance, reinforcement learning) were discussed in detail in the work of S. Du and W. Hu (2019): *Linear Convergence of the Primal-Dual Gradient Method for Convex-Concave Saddle Point Problems without Strong Convexity*. We will add this remark to our paper to make this point more clear and hence the assumption more accessible to a more general audience not familiar with these details. This is a good suggestion, albeit a very minor remark in our view.
>
> 2. Assumption 4 is also satisfied in the case of **decentralized distributed optimization** problems: here, matrix $K$ is equal to the square root of the Laplacian matrix and $F(x)$ is the indicator function of the set $\{0\}$. This case also covers centralized distributed opimization. We discuss such problems in Section 6.
>
> >  It would be much better if some examples of $K$ and  $F$ arising from real-world applications can be provided. How small can  $\mu_{xy}$ be? It appears in the denominator of the complexities. Although it is a constant, it would be better if we know its magnitude.
>
> 1. The assumption looks a bit complicated, but this is merely due to our efforts to make it as little restrictive as possible. Please notice that the assumption is implied by the stronger (but easier to parse) assumption that the smallest eigenvalue of $KK^\top$ is positive., in which case we can set $\mu^2_{xy}$ to be equal to the smallest eigenvalue of $KK^\top$. Note that since $KK^\top$ is symmetric positive semidefinite, its smallest eigenvalue is always nonnegative. So, asking for it to be strictly positive is not a very strong assumption: we just need $KK^\top$ to be positive definite as opposed to positive semidefinite. In ProxSkip, for example, $K$ is the identity matrix, and hence $\mu^2_{xy} = 1$. So, this value has no adverse effect on the complexity.
>
> 2. In the situation when $F$ and $K$ interact in a favorable way, described by the inclusion $\partial F^\star (y) \subseteq {\rm range} K$ for all $y$, we do not need to even assume $KK^\top$ to be positive definite, and set $\mu^2_{xy}$ to be the smallest positive eigenvalue of $KK^\top$, which is, by definition, always positive. In the decentralized optimization setting considered in , as described above, this favorable inclusion holds, and $\mu_{xy}^2$ is the smallest positive eigenvalue of the gossip matrix $W$ (which depends on the Laplacian of the graph defining the communication links). In this case, the value of $\mu^2_{xy}$ depend on the connectivity of the graph. Better connected graph will have larger values of $\mu^2_{xy}$. We can link to some work on gossip or spectral graph theory to help the reader not familiar with these notions and results, which are very well known.

---

> > ### Comment · Reviewer_EGED · 2022-08-03
> > **Thanks for the clarification.**
> >
> >
> > Please add the above discussions on Assumption 4 in the main text, preferably right Assumption 4.

---

> > > ### Author Response · Authors · 2022-08-05
> > > **Re: Thanks for the clarification.**
> > >
> > > Yes, we will do so. We do not think we can add it in this much detail in the main text due the page limitations, but we will add a shorter version of it in the main body of the paper, and a full version to the Appendix. We will provide a link from the main paper to the Appendix.

---

> ### Author Response · Authors · 2022-08-01
> **Response to Reviewer EGED (Part 2)**
>
> > There are no numerical comparisons with other methods, especially ProxSkip. The claimed superiority in gradient complexities should be accompanied by numerical evidence.
>
> 1. We believe that a strong theoretical paper does not require a numerical section. Please note that strong numerical papers also do not require a mathematical section. There are plenty of examples of immensely influential papers in both categories. Here are some (randomly obtained) examples of recent NeurIPS 2021 theory papers without any experiments ("Oracle Complexity in Nonsmooth Nonconvex Optimization", "Complexity Lower Bounds for Nonconvex-Strongly-Concave Min-Max Optimization",  "Optimal Rates for Random Order Online Optimization"), and empirical papers without any theory ("Bias Out-of-the-Box: An Empirical Analysis of Intersectional Occupational Biases in Popular Generative Language Models", "Learning to See by Looking at Noise", "Deep Self-Dissimilarities as Powerful Visual Fingerprints").
>
> 2. We would be therefore most happy if our paper could be judged on the merits of our algorithmic and theoretical complexity contributions, which we believe are very strong, as they lead to new best theoretical results for the literature of local gradient type methods, which have been studied in the distributed and federated learning literature for about a decade. We improve upon the current state-of-the-art local method ProxSkip, which is the work that managed to show that local steps provably lead to communication acceleration, which is one of the foundational observations of the entire field of federated learning. However, this foundation was shaky until the ProxSkip paper breakthrough. Since we improve upon ProxSkip, our method has automatically the current best theoretical communication (matching that of ProxSkip in the standard setting, and improving on it in the decentralized setting) and local computation complexity: from $O(\kappa^{1/2})$ to $O(\kappa^{1/3})$ and $O(\kappa^{1/4})$, depending omen the method $\cal M$ used to approximately solve the prox in our Algorithm APDA-Inexact, for example. We believe these results stand on their own. An orthogonal contribution of our work is to show how such a method can be constructed step-by-step, from very general ideas of the proximal point method, through the Chambolle-Pock method, with two important modifications that eventually lead to a local method with the desired theoretical properties. In so doing, we are introducing to the Federated Learning audience the beauty of proximal splitting schemes, and the elegance of a general approach to designing numerical methods.
>
> 2. Having said that, **we are happy to include in the camera-ready version of the paper some preliminary numerical experiments.** This is easy to do, but will take time since we want to do this properly. But we do not think this will really add much value to our work. We do agree though that the paper will perhaps become marginally better.
>
> > In Assumption 4, the paper you cited uses $\mu_{xy}^2$ instead of $\mu_{xy}$, should it be $\mu_{xy}^2$?
>
> It should be $\mu_{xy}^2$. This is a typo. Thank you. We heave already fixed this in our local version of the paper.

---

> > ### Comment · Reviewer_EGED · 2022-08-04
> > **Regarding the theoretical contributions**
> >
> >
> > I agree that the theoretical contributions are solid for the federated learning community. However, it would be better if detailed numerical comparisons and code can be provided, so that readers can have easy access to the proposed algorithms and can test them on their problems.
> >
> > I decided to raise my score and I am looking forward to the numerical comparisons.

---

> > > ### Author Response · Authors · 2022-08-04
> > > **Re: Regarding the theoretical contributions**
> > >
> > > Thank you, much appreciated. We will do so.

---

> ### Author Response · Authors · 2022-08-01
> **General comments**
>
> You rated our contribution as "fair" (score 2). We believe this is not a "fair" judgment of our actual contributions (please forgive the pun; we made it for comic effect to lighten up the discussion ;-). Please can you have another look at what we say about contributions in the paper, and also at the quick summary we provide in point 2 of "Response to Reviewer EGED (Part 2)"? We believe our contributions are substantial.
>
> Please also note that the major concerns you raise are objectively minor comments and questions which do not invalidate our theory, nor make the results weaker in any way. They are simply questions about certain quantities appearing in our theory and bounds, which we answered in our response. Please let us know if these answers is what you were looking for. We do not see how such (to us very simple) clarifying questions can be seen as major issues. Once these clarifications are provided, we do not see any more outstanding concerns. We do agree though that adding some comments about these will make the assumptions more accessible to a more general audience, and we are most happy to do so.
>
> Thank you very much again for an overall positive evaluation of our work.

---

### Author Response · Authors · 2022-08-02
**A Message to all Reviewers**

We wish to thank all reviewers for donating their free time to the NeurIPS community. Much appreciated. We are particularly thankful for the many comments, many of which are useful.

**In short, in this work we design new algorithms, by combining several recent techniques and tricks mostly from the proximal splitting literature in a very particular and, we believe, creative way, so as to obtain new theoretical SOTA results in terms of communication complexity and local gradient complexity in the field of federated learning, where local gradient methods have been studied for many years, but without much success in beating even simple gradient descent in terms of communication complexity. Inspired by recent breakthrough of Mishchenko et al (2022) with their ProxSkip method, which is the first work that breaks the ${\tilde O}(\kappa)$ communication complexity barrier for local gradient methods, and thus shows that local training steps indeed lead to communication acceleration, as has been observed by practitioners, we take a very different path, borrowing and adapting various tools and ideas from the proximal skipping and decentralized optimization literature (e.g., Chambolle Pock splitting, inexact solution of prox, fixing linear convergence, accelerated gossip), combining them in a creative way with the goal of designing new local gradient methods that  improve upon the SOTA results of Mishchenko et al. in several ways.**

We have received many positive comments in 4 reviews:

**Reviewer EGED (score 6)**
- "The theory for inexact evaluation of prox of $G$ at each iteration is nice. The possible difficulty in prox of $G$ is often an overlooked factor.
- "The avoidance of the strong convexity of $F$ is also novel. This makes the convergence proof non-trivial."

**Reviewer bWvi (score 5)**
- "The main contribution is the relaxation of the strong convexity and smoothness assumptions on the conjugate function $F^\star$."

**Reviewer zYvm (score 6)**
- "In general I am positive about this work. The paper obtains some improvements on solutions for problem (1) and also to federated learning, as a consequence of their framework plus some modifications. The paper is written in a self contained way that allows to follow the analysis well. "
- "There is certainly some value in having put together all of these things, and on combining the techniques of (Kovalev et al 2021) with Chambolle Pock. "

**Reviewer qFrD (score 7)**
- "The improved complexity bounds are novel and significant. I especially like the connections of the techniques to existing literature."
- "The literature review and comparisons in Table 1-2 are much appreciated. This gives clarity to how it fit into other state-of-art methods."
- "The overall presentation of the material is well-organized and easy to read. I especially like the summary in Section 2 and the review in Section 3. The writing style is clean and there are very few grammatical issues."
- "The authors have sufficiently addressed the possible limitations (assumptions) of their work."

We have written a detailed response to all criticism raised and questions asked. We have found several useful suggestions which we will incorporate in the paper; thanks for those! But we have also found some criticism based on misunderstanding of some content of the paper, and of the real importance of our results. We have tried to clarify all these points in our response.

Very brief summary:

**Reviewer EGED (score 6)** raised just two concerns. We believe they are minor since one is easily handled by a simple explanation which we provided (and we will add this to the paper as well), and the other concerns experiments, which are at most tangentially relevant to a theoretical piece of work in our view, and which we will nevertheless do as suggested.

**Reviewer bWvi (score 5)** made some interesting comments, but we do not find them to be major. We addressed them all. We believe the low score here is due to the reviewer not appreciating the import of our results for the federated learning community (please can you inspect Tables 2 and 3?). No comments regarding these many significant contributions have been made.

**Reviewer zYvm (score 6)** asked some clarifying questions which we believe answered well, and made numerous useful suggestion to improve the paper (such as adding some explanations and citations). We noticed here as well though that the reviewer focused on our contributions to the proximal splitting field rather than to federated learning, where our main contributions lie. We believe these contributions as significant and numerous. We explained this in detail.

**Reviewer qFrD (score 7)** made a number of useful suggestions which we will incorporate to improve the paper. All are minor.

We will be happy if you could engage with us during the reviewer-author discussion period. We would like to know whether our replies addressed your questions, and what else remains to be explained.

Thanks!

---

> ### Comment · Reviewer_zYvm · 2022-08-04
> **q**
>
> > Reviewer zYvm (score 6) asked some clarifying questions which we believe answered well
>
> Please note that you missed the question I formulated on the questions section

---

> > ### Author Response · Authors · 2022-08-05
> > **Re: q**
> >
> > Sorry, this was not intentional! We will answer soon by updating this post!
> >
> > *Update:*
> >
> > You are right that we did not comment on this more in the paper. We left this as a passing remark, without elaborating. The main reason why we did that at the time of writing was that we did not think it was important to elaborate, since we had a limited space, since we believed experts will know these applications, and since we wanted to focus the remaining part of the paper more on our contributions from category i) [i.e., to the theory of local gradient-type methods in federated learning; see Tables 2 and 3 for summary] instead of those from category ii) [i.e., to the proximal splitting literature; see Table 1 for summary]. Please recall that we explained in more detail what we mean by these two categories in "Response to Reviewer zYvm (Part 2)", where we also argued why we consider category i) much more important, and the heart of our work.
> >
> > However, we agree it would be useful to list some examples of problems where $F^*$ is nonsmooth. These include (not in order of importance):
> > - minimax reformulation of lasso (since $F(x)=\Vert x\Vert_1$ is not strongly convex, its Fenchel conjugate is not smooth)
> > - minimax reformulation of support vector machines (see, e.g., https://arxiv.org/pdf/1705.07252.pdf)
> > - many of the imaging applications mentioned in the literature related to the Chambolle-Pock method do not involve smooth $F^*$; some examples include
> >     - *total variation based image denoising* under the Rudin, Osher and Fatemi (1992) model or in the TV-L1 model (in both cases $F^*$ is the indicator function of a certain convex set);
> >     - *image deconvolution and digital zooming* ($F^*$ is an L2-regularized variant of the indicator function of a convex set);
> >     - *motion estimation*  ($F^*$ is the sum of two indicator functions of a convex sets);
> >     - *image inpainting* ($F^*$ is also the indicator function of a certain convex set here);
> >     - and many more.
> >
> > We will make a brief remark on this in the main body of the paper, and will also include a link to the appendix where we will add the abive examples (and perhaps some more) in full.
> >
> > *References:*
> >
> > A. Chambolle, T. Pock. A first-order primal-dual algorithm for convex problems with applications to imaging, Journal of Mathematical Imaging and Vision, volume 40, pages 120-145, 2011.
> >
> > L. Rudin, S. J. Osher, and E. Fatemi. Nonlinear total variation based noise removal algorithms. Physica D, 60:259–268, 1992.

---

### Meta-Review · Area_Chair_oUdk · 2022-08-22

**Recommendation:** Accept
**Confidence:** Less certain

**Metareview:**

The paper obtains new algorithms in the domain of federated learning that provide state of the art complexity guarantees in terms of communication and local gradient oracle queries. These results are obtained by combining ideas and techniques from the literature on proximal splitting algorithms with the specific setting of federated learning. While the reviewers generally appreciated the contributions of the paper and its clarity of presentation, the authors are advised to carefully consider the feedback provided by the reviews when preparing a revision of the paper. Most notably, the paper should be making a more careful comparison to existing work, as recommended by Reviewer zYvm, some of which is summarized here, while the rest can be found in the original review and the discussion.

* There should be clear pointers to the literature for the results that are essentially re-proven in this work. A specific example are the results from Appendix B. Similarly, it needs to be stated clearly which parts of the analysis of Chambolle-Pock are reproduced.

* Appendix D should compare to the results of Kovalev et al.

* Appendix G needs to state clearly that all of the methods mentioned there were already known and provide correct references for all. Further, there needs to be a more careful comparison to the results of Kim and Fessler already cited in the same section.


**Award:**

No

---

### Decision · Program_Chairs · 2022-09-14

Accept